# ZJU-AERO V0.5: An Accurate and Efficient Radar Operator Designed for CMA-GFS / MESO with Capability of Simulating Non-spherical Hydrometeors

Hejun Xie[1], Lei Bi[1], Wei Han[2,3]

[1]Key Laboratory of Geoscience Big Data and Deep Resource of Zhejiang Province, School of Earth Sciences, Zhejiang University, Hangzhou, 310027, China

[2]CMA Earth System Modeling and Prediction Centre (CEMC), China Meteorological Administration, Beijing 100081, China

[3]State Key Laboratory of Severe Weather (LaSW), Chinese Academy of Meteorological Sciences, China Meteorological Administration, Beijing, 100081, China

*Correspondence to*: Lei Bi (bilei@zju.edu.cn), Wei Han (hanwei@cma.gov.cn)

**Abstract.** In this study, we present a new forward polarimetric radar operator called the Accurate and Efficient Radar Operator designed by ZheJiang University (ZJU-AERO). This operator was designed to interface with the numerical weather prediction (NWP) model of the global forecast system/regional mesoscale model of the China Meteorology Administration (CMA-GFS/MESO). The main objective of developing this observation operator was to assimilate observations from the Precipitation Measurement Radar (PMR). It is also capable of simulating ground-based radar's polarimetric radar variables, excluding the Doppler variables such as radial velocity and spectrum width. To calculate the hydrometeor optical properties of ZJU-AERO, we utilize the invariant-imbedding T-matrix (IITM) method, which can handle non-spherical and inhomogeneous hydrometeor particles in the atmosphere. The optical database of ZJU-AERO was designed with a multi-layered architecture to ensure the flexibility in hydrometeor morphology and orientation specifications, while maintaining operational efficiency. Specifically, three levels of databases are created that store the single scattering properties for different shapes at discrete sizes for various fixed orientations, integrated single scattering properties over shapes and orientations, and bulk scattering properties incorporating the size average, respectively. In this work, we elaborate on the design concepts, physical basis, and hydrometeor specifications of ZJU-AERO. Additionally, we present a case study demonstrating the application of ZJU-AERO in simulating radar observations of Typhoon Haishen.

## 1 Introduction

The development of regional models with finer horizontal resolutions, such as the Chinese operational regional numerical weather prediction (NWP) model, known as the regional mesoscale model of China Meteorology Administration (CMA-MESO) (Chen et al., 2008; Shen et al., 2023), necessitates the acquisition of more convective-scale information about the

atmosphere to improve quantitative precipitation forecasts. Fortunately, the measurements from space-borne and ground-based weather radars provide valuable sources of three-dimensional kilometre-scale volume data with high temporal resolutions. However, weather radar can only observe the amplitude and phase of electromagnetic waves echoed from meteorological objects, specifically various types of hydrometeors. Except for the Doppler radar variables, such as radial velocity (beyond the

scope of this study), it is challenging to establish a connection between the prognostic variables simulated by the NWP model and the observable polarimetric radar variables, which are inferred from the statistical moments of voltage time series collected by the receiver of the weather radar electronics system (Zhang, 2016).

The software package introduced in this work is referred to as a "forward radar operator", designed to transform the model prognostic variables into the observation space, resulting in equivalent simulated synthetic radar variables. Utilizing a unified

forward radar operator for assimilations and retrievals is believed to be superior to employing an ensemble of retrieval relationships along with pre-processing procedures and corrections for different frequencies and platforms. In essence, using radar data in the observation space is preferred over the model space due to the highly non-linear and non-unique nature of the processes that observational operator of polarimetric radar describes.

Several radar operators have been developed and published over the past several decades. For instance, Jung et al. (2008)

implemented a polarimetric radar simulator known as the Polarimetric Radar data Simulator developed by the Center for Analysis and Prediction of Storms (CAPS-PRS) at the University of Oklahoma. This simulator uses spheroids to characterize hydrometeors and computes optical properties using online Rayleigh approximations or offline look-up tables (LUT) constructed by the extended boundary condition method (EBCM) as described in Mishchenko and Travis (1994). This simulator has been applied to low-frequency bands, such as S-, C- and X-band. Ryzhkov et al. (2011) described another radar

operator for research purposes, specifically tailored for spectra-resolving cloud microphysics models, although it is more computationally expensive. Zeng et al. (2016) described an efficient radar operator that is online-coupled to the Consortium for Small-scale Modelling (COSMO) and Icosahedral Nonhydrostatic Weather and Climate Model (ICON) model, making use of Mie / T-matrix scattering look-up table of solid, liquid and melting (mixed-phased) hydrometeors, named as efficient modular volume-scanning radar forward operator (EMVORADO). While early versions of EMVORADO focus on non-

polarimetric radar variables, later developments on EMVORADO have enabled its capability on simulating dual-polarization variables and conducted sufficient evaluations (Trömel et al., 2021; Shrestha et al., 2022). In addition to the above operators, Wolfensberger and Berne (2018) reported a cross-platform polarimetric radar operator termed POLarimetric forward radar operator for the COSMO model (COSMO-POL). This operator was designed for the COSMO–NWP model and can simulate melting particles. The optical database of the COSMO-POL was constructed also by using the EBCM, characterizing all

hydrometeor particles as homogeneous spheroids. However, in the COSMO-POL, the hydrometeor orientations and shape parameter settings are fixed at the level of the optical database, which limits sensitivity testing and fine-tuning. Wang and Liu (2019) reported a forward reflectivity observation operator (together with its tangent linear and adjoint operator) with simulation capability of frozen hydrometeors designed for data assimilation purpose of ground-based radar in Weather Research & Forecasting Model (WRF), in which the simulated reflectivity are parameterized as fast polynomial-relationships

with respect to mixing ratios of hydrometeors. More recently, Oue et al. (2020) developed the Cloud-resolving model Radar SIMulator (CR-SIM), which can simulate polarimetric Radar and light detection and ranging (Lidar) observations for various Cloud Resolving Models (CRM), including the WRF, ICON, Regional Atmospheric Modeling System (RAMS), and Advective Statistical Forecast Model (SAM). CRM-SIM has a unique capability of explicitly representing rimming procedure by coupling with the prognostic variable known as rimming ratio in the Predicted Particle Properties (P3) microphysics

package (Morrison and Milbrandt, 2015). However, CR-SIM is currently limited to ground-based platforms and offers no explicit treatment for melting particles. Moreover, the fast parameterized forward radar operator developed by Zhang et al. (2021) contains a melting scheme module, targeting data assimilation purposes.

This work aims to design a cross-band and cross-platform radar operator for research purposes, such as microphysics package validation, and operational data assimilation use in CMA–GFS / MESO. The software prototype of this radar operator,

hereafter referred to as the Accurate and Efficient Radar Operator designed by ZheJiang University (ZJU-AERO), which essentially addresses the scattering computation of hydrometeors and construction of optical properties database as the key aspects in the evolution of the radar operator. We utilize a semi-analytical scattering computation approach of T-matrix to ensure accuracy and features a multi-layered optical database that includes single scattering properties at discrete sizes and orientations, integrated single scattering properties over shapes and orientations, and bulk scattering properties incorporating

the size average. Additionally, ZJU-AERO allows for flexibility in particle orientation and shape probability distribution tuning. Notably, this software has also inherited established techniques, such as sub-beam sampling, used for simulating the effects of beam bending/broadening/shielding (Ryzhkov, 2007).

The development of ZJU-AERO was primarily motivated by the future data assimilation purpose of the precipitation measurement radar (PMR) onboard the FengYun-3 Rain Measurement satellite (FY3-RM) (Zhang et al., 2023). The parameters

of the instrument FY3-RM/PMR are comparable with those of the Global Precipitation Measurement/Dual-Frequency Precipitation Radar (GPM/DPR). The DPR onboard the GPM was designed to obtain the storm structure, rainfall rates, drop size distributions (DSD), path-integrated attenuation, and other useful information that are not available from passive sensor observations (Iguchi et al., 2003; Iguchi, 2020). Both PMR and DPR have two bands (the Ku- and Ka-bands), while PMR is designed with a swath of 303 km, horizontal resolution of 5 km at nadir, and radial resolution of 250 m.

This paper is organized as follows. Section 2 provides readers with an overview of the general concepts of ZJU-AERO. The variables (matrices) that describe the scattering properties of hydrometeors are presented here to clarify the notation convention used in this study and eliminate ambiguity in the context. Details of the implementation of hydrometeor settings, including dielectric models, aspect ratio models, orientation preference models, and particle size distributions (PSDs) are also listed in Section 2. Section 3 elaborates on the flexible architecture of the state-of-the-art non-spherical scattering properties database,

which distinguishes the ZJU-AERO from its predecessors. The characteristics of the multi-layered optical database are illustrated using a non-spheroid hydrometeor model, namely the Chebyshev-shaped raindrop. Moreover, Section 4 presents a case study of observations of a tropical cyclone given by the space-borne radar GPM/DPR, compared with simulations given

by the ZJU-AERO. Sensitivity tests of PSDs and non-sphericity are also described in this section. Section 5 summarizes this study and describes the development plans for the ZJU-AERO.

## 2 General Descriptions of ZJU-AERO

The ZJU-AERO was developed to simulate the observable polarimetric radar variables for radar systems aboard on various platforms, including ground-based, space-borne, and potentially airborne radars in the future. The conceptual graph of the multi-platform radar operator is shown in Figure 1. While the physical principles of the weather radar detection process are universal, certain factors, such as beam-broadening, beam-bending, and beam-blocking among others (as indiated along the beam trajectory in Figure 1), which are critical to ground-based radars, are not equally important across platforms. For example, the beam-bending effect is typically negligible for space-borne radar due to large absolute elevation angles (usually 70~90°, as illustrated in Figure 1 for spaceborne radar) and shorter beam trajectories (usually < 20 km) below the model top, as compared to the ground-based radar (the trajectory can reach up to about 250 km). However, the simulations of space-borne and ground-based radars share consistent hydrometeor setting entries for snow/graupel, melting snow/graupel, and rain, such as the dielectric model, particle morphology (distribution), size distribution, orientation preference, and others.

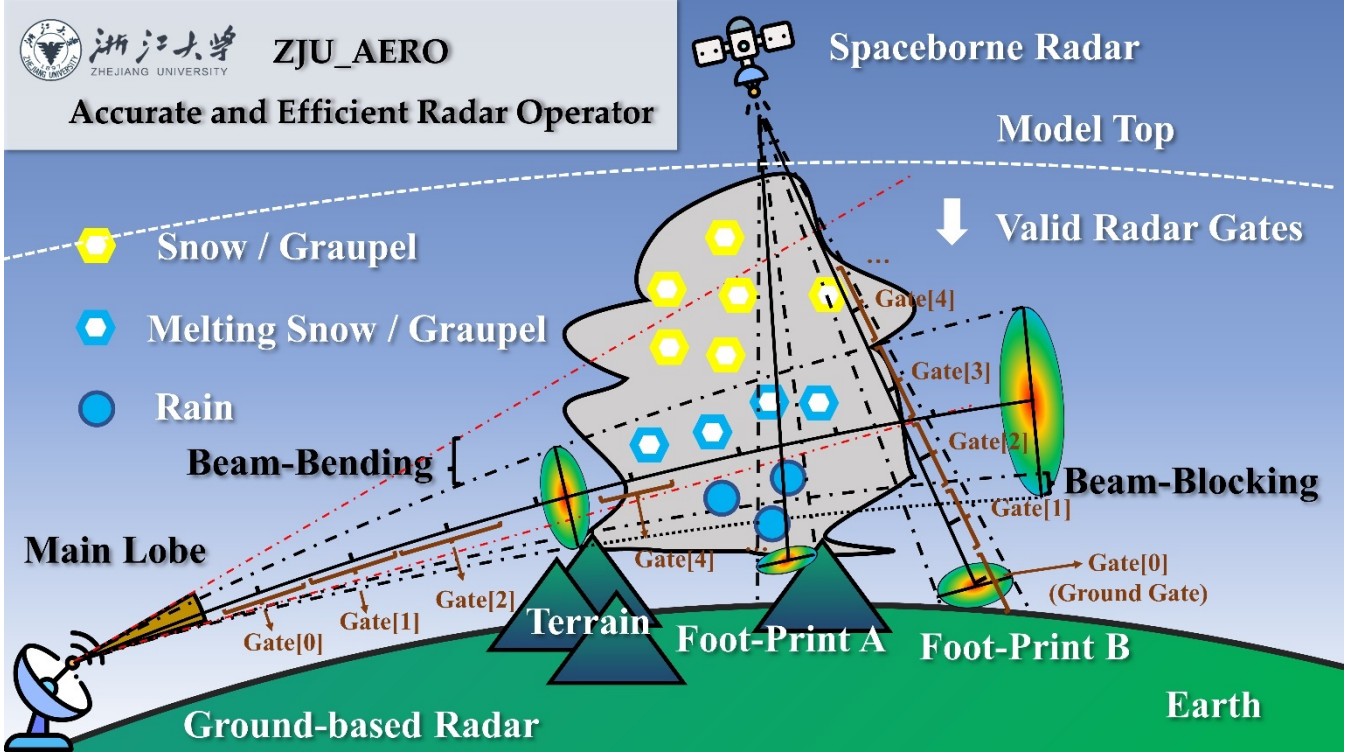

**Figure 1: A conceptual graph illustrating the types of simulations that the Accurate and Efficient Radar Operator designed by ZheJiang University (ZJU-AERO) can accommodate. This graph visualizes the beam-bending and beam-blocking effects, which are taken into considerations by many radar operators designed for ground-based radars. For these radars, the sampling volume within the main lobe of the radar antenna patterns increases with the range of detection, as indicated by the area between the two dashed–**

dotted black curves. The radar beam follows a 4/3$R_E$-radius curve under standard atmospheric profile conditions and is commonly referred to as beam-bending (in which $R_E$ is the radius of the Earth). Note that the sampling volume can be partially blocked by terrain as indicated by the area above the dotted black line. The space-borne radar scans, which have relatively small zenith angles (usually within 20°), are weakly affected by the beam-bending phenomenon. The radar gates of ground-based radar are recorded from the zero range-bin, while the gates of spaceborne radar are only stored within the data sampling range with respect to mean sea level, for example, GPM/DPR products only stores range-bins of altitude between –5 km and 19 km as spaceborne radars scan always cast a footprint on the earth. In ZJU-AERO, only the radar gates beneath the NWP model top, known as "valid radar gates", are represented and simulated in order to save on memory usage and computational resources.

## 2.1 Flow Chart and Concepts

Figure 2 provides an overview of the ZJU-AERO simulation procedure for a single radar scan. ZJU-AERO consists of five modules, represented by green boxes in Fig. 2. These modules are (A): NWP interface submodule, (B) Beam submodule (decomposed into B1 and B2 for ground-based and spaceborne Radars, respectively, (C) Interpolation submodules, (D) Hydrometeor sumbmodule, and (E) Core submodule. The workflow of ZJU-AERO can be summarized as follows:

1. The NWP interface submodule (A) reads NWP prognostic variables from external storage files and interpolates the data defined on the original model grid (such as horizontal Arakawa-C grid and vertical Charney-Phillips grid in CMA–GFS/MESO (Chen et al., 2008; Shen et al., 2023)) to a regular grid on which all variables are collocated and evenly spaced horizontally (in the space of projection). The prognostic variables include hydrometeor variables (Qc, Qi, Qr, Qs, and Qg representing mixing ratios of cloud water, cloud ice, rain, snow, and graupel, respectively) and dynamic variables (U, V, W, T, P, and Q representing zonal, meridional, vertical wind, temperature, pressure, and water vapor mixing ratio, resepectively). Addtionally, static model information such as orography data defined on model grids is required for simulating partial beam blocking. This step prepares for a quick and convenient second-round interpolation from the regular model grid to the radar beam trajectories (in step 3). It is worth to point it out that ZJU-AERO can also interface with the output of WRF NWP model (Skamarock et al., 2019). Enabling ZJU-AERO to interface with grid data from another NWP model involves only technical adjustments, requiring basic information about that NWP model's horizontal mesh (projections), vertical grid, and variable mapping table.

2. The beam trajectory submodule (B) calculates the propagation trajectories of the radar beam. For ground-based radar (B1), users have the option of using an online trajectory solver that uses temperature and humidity profiles above the radar site. Specifically, the atmosphere refractive index Na are determined from atmosphere temperature T, pressure P, and water vapor mixing ratio Q. The trajectory is determined by a ray-tracing ordinary differential equation (ODE) solver (Zeng et al., 2014). Additionally, ZJU-AERO offers an alternative option of using an offline 4/3$R_E$ solver for ground-based radar in ZJU-AERO. Multiple sub-trajectories (determined by horizontal quadrature number N × vertical quadrature number M) are sampled within the 3dB-beamwidth of the main lobe for the N×M sub-trajectories. The observable radar variables are calculated by integrating bulk scattering properties over the antenna patterns (as described in step 5) to obtain the final results (beam-broadening and beam-blocking are considered in this way). We applied the sub-beam sampling and averaging methods as described in Zeng et al. (2016). Nevertheless, for space-borne radar (B2), the beam trajectory is computed using the geometry shown in Figure A1, and sub-trajectory

averaging is not supported at this time. For more details on the beam trajectory module of space-borne radar (B2), please refer to Appendix A.

3. The interpolation submodule (C) uses a trilinear interpolation algorithm to interpolate the NWP prognostic variables to the gates of radar beam trajectories. The quick trilinear interpolations are performed in a two-step manner: (1) vertically interpolating the data defined on the eight vertices of the cube containing the radar gates to the four vertices of the horizontal box surrounding the radar gate and (2) performing bilinear interpolation; we adopted the approach described in Appendix A of Wolfensberger and Berne (2018) and reimplemented it as a Cython extension.

4. The hydrometeor submodule (D) specifies the properties of hydrometeors in each radar gate along each trajectory, usually by loading presets of microphysics-consistent constants of hydrometeors. This includes the orientation preference, probability distribution of particle morphology, particle size distribution (PSD), and other parameters. Those presets of hydrometeor properties can also be modified by users to perform "forced" (i.e., inconsistent with microphysics) simulations for research purposes. The PSD parameters of hydrometeors are solved in this step, from prognostic bulk hydrometeor variables in NWP models (mass concentrations and number concentrations). For more details, see Section 2.3.

5. The core module (E) finally calculates the polarimetric radar variables:

   5.1 The bulk scattering properties of particles in each radar gate are computed by integrating the single scattering properties across the size spectrum of each hydrometeor type and summing over hydrometeor types, which can be conducted either online (E1, research mode) or offline (E2, operational mode) in ZJU-AERO. The scattering properties LUTs of ZJU-AERO are consulted in this step, which is composed of three levels (Level A, Level B, and Level C). The multi-layered architecture will be introduced in detail in Section 3.1. The research mode is more flexible since it accesses the Level B database for single scattering property, while the operational mode is more efficient by accessing the Level C database for bulk scattering properties straightforward. We provide users with tool scripts for level A to B and Level B to C conversions (integration parameters are alterable in YAML configure files).

   5.2 Once the bulk scattering properties on each sub-trajectory gridpoint within each radar gate are available, the antenna pattern integration involves integrating the bulk scattering properties within the scanning volume of each radar gate.

   5.3 Then, the core module calculates the intrinsic polarimetric radar variables on each radar gate, based on the bulk scattering properties incorporating the antenna pattern integration presented in step 5.2. These radar variables include single-polarization reflectivities ($Z_H$ for horizontal reflectivities), dual-polarization variables ($Z_{DR}$, $K_{DP}$, $\delta_{hv}$, $\Phi_{DP}$ and $\rho_{hv}$ for differential reflectivity, differential phase shift, backscatter differential phase and co-polar correlation coefficient, respectively), and attenuation variables ($a_H$ and $a_V$ for horizontal and vertical attenuation coefficient, respectively). The definitions of these variables can be found in Appendix D. For a detailed explanation of the intrinsic radar variables, please refer to Zhang (2016).

5.4 In the final step of core module, the observable radar variables are obtained by taking into account the attenuation and phase shift accumulated along the beam trajectories.

The above procedures of steps 2-5 are independent for each single beam, which guarantees the top-level scalability of the forward operator. Therefore, we are using the shared-memory python parallel library (multiprocessing) to speed up the simulation.

In addition, for those computationally intensive tasks (such as the trilinear interpolation in step 3), we are employing the technique of mixed programming (building C/Fortran-extensions that interface with python scripts) to further accelerate the computation.

The performance of the forward operator is generally satisfactory: ZJU-AERO can complete a ground-base station volume scan with 9 Plan Position Indicator (PPI) sweeps in 2 minutes on a modern laptop CPU with a 6-core (i7-10750H) if online size distribution integration is performed and the operator takes Level-B single scattering property database as input. If Level-C bulk scattering property database is used, such a volume PPI scan can be completed even faster (in 30s). Such efficiency can support data assimilation purposes, while also preserving flexibility for research purposes.

In this paper, we do not elaborate on the algorithm details of certain issues, such as (a) trilinear interpolation, (b) sub-beam sampling and antenna pattern weighted averaging, and (c) ray-tracing trajectory solver. For trilinear interpolation, we follow the approach described in Wolfensberger and Berne (2018) to interpolate the model grid data to radar gates of each sub-trajectory gridpoint. Regarding sub-beam sampling and antenna pattern weighted averaging, we utilized Gauss-Hermite quadrature as outlined in Caumont et al. (2006). As the ray-tracing trajectory solver, we offer users both an offline beam trajectory solver (4/3RE) and an online beam trajectory solver (Zeng et al., 2014). All of these methods are reimplemented in Python, using either efficient numpy/scipy API or customized Cython extensions.

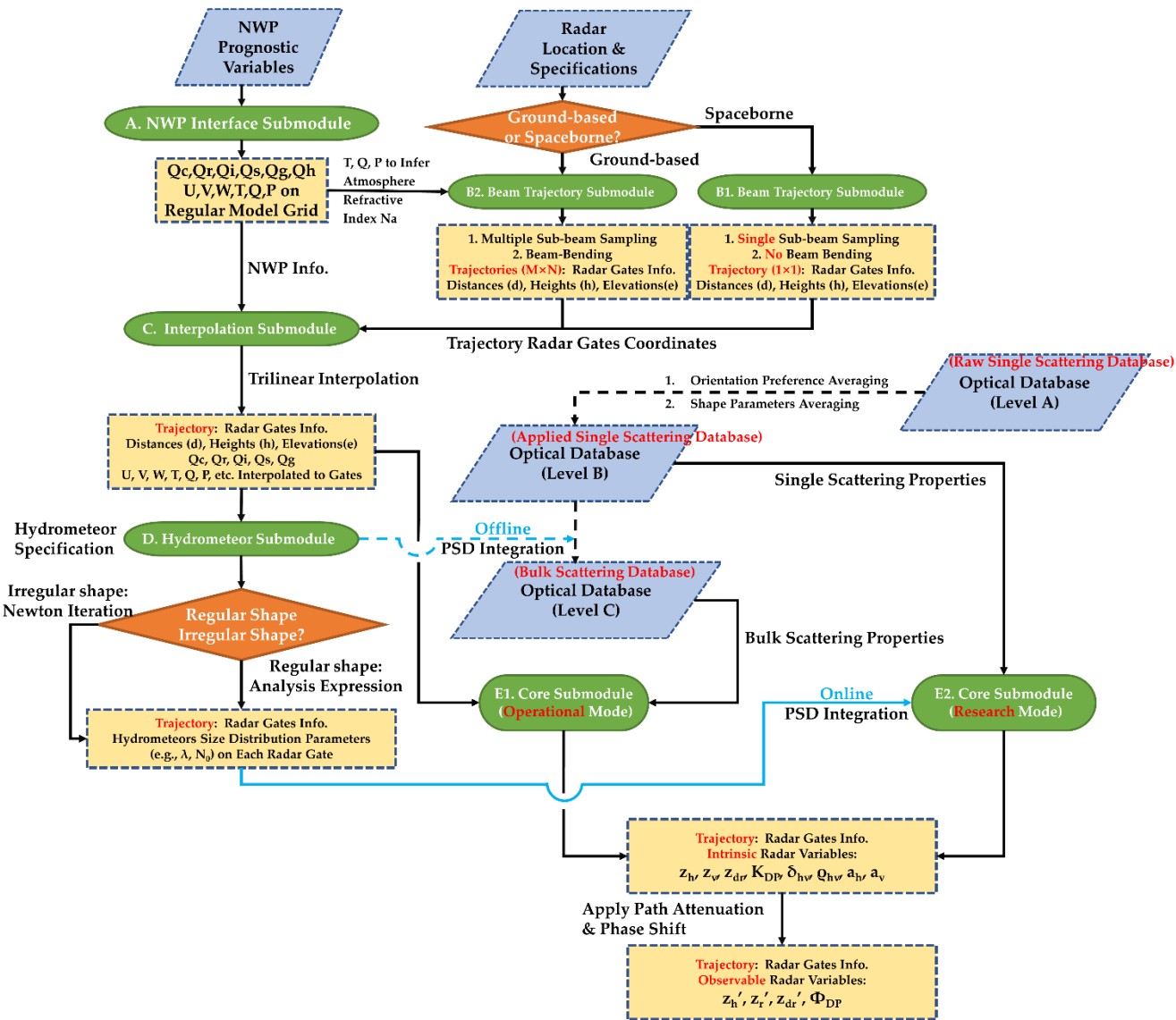

**Figure 2: A conceptual flow chart of ZJU-AERO. The parallelogram boxes represent input data (including numerical weather prediction (NWP) output, radar specifications, and optical properties look-up tables). The green round rectangles indicate the names of the key submodules of ZJU-AERO. The yellow dashed boxes indicate the key data structure used in the simulations. The diamonds represent the points at which crucial "if-else" judgements can be conducted during processing. Those dashed arrows in this flowchart represent external database generation steps carried out using released tool scripts of ZJU-AERO.**

## 2.2 Physical Basis

At this point, we can specify the formulation convention used in the radar operator and move on to the non-spherical optical database design of ZJU-AERO.

The amplitude scattering matrix $\mathbf{S_{FSA}}$ of a single particle is defined as follows:

$$\begin{bmatrix} E_h^{sca} \\ E_v^{sca} \end{bmatrix} = \frac{e^{-ik_0 r}}{r} \mathbf{S_{FSA}} \begin{bmatrix} E_h^{inc} \\ E_v^{inc} \end{bmatrix} = \frac{e^{-ik_0 r}}{r} \begin{bmatrix} S_{hh} & S_{hv} \\ S_{vh} & S_{vv} \end{bmatrix}_{FSA} \begin{bmatrix} E_h^{inc} \\ E_v^{inc} \end{bmatrix}, \tag{1}$$

In Eq. (1), $\mathbf{E}$ indicates the vector electric field, and the superscripts "sca" / "inc" represent scattering and incident waves, respectively. The subscripts "h" and "v" indicate two decomposed components of the vector electric field in the horizontal and vertical directions, respectively. Specifically, the horizontal and vertical unit vectors are defined as unit vectors $\hat{\phi}$ and $\hat{\theta}$ of the spherical coordinate system under the Forward Scattering Alignment (FSA) convention. The differences between FSA and Backward Scattering Alignment (BSA) are described in Appendix C. $k_0$ is the wave number in free space, and $r$ is the distance from the particle center.

The scattering matrix relates the incident electric field to the scattered electric field, and it must have the dimension of L (L is the dimension symbol of length). In practice, the amplitude scattering matrix is usually expressed in the units of mm. The amplitude scattering matrix of a single hydrometeor particle is obtained from scattering computations, specifically using the T-matrix method in this study.

Apart from the $2\times2$ complex amplitude scattering matrix $\mathbf{S}$ defined on complex electric field vector bases, one can define the $4\times4$ real matrix, known as the Mueller matrix, $\mathbf{Z}$, and extinction matrix, $\mathbf{K}$, which describe polarimetric light scattering and extinction properties of particles on the real Stokes vector bases, respectively. We kept the definitions of $\mathbf{Z}$ and $\mathbf{K}$ consistent with those mentioned in a study by Mishchenko (2014).

The Mueller and extinction matrices can be derived from the amplitude scattering matrix (Mishchenko, 2014). Specifically, the forward scattering amplitude matrix shows a linear relationship with $\mathbf{K}$. For example, the formulas of the matrix elements of $\mathbf{K}$ used in this study are shown below:

$$K_{11} = \frac{2\pi}{k_0} \Im(S_{hh} + S_{vv})$$

$$K_{12} = \frac{2\pi}{k_0} \Im(S_{hh} - S_{vv}), \tag{2}$$

$$K_{34} = \frac{2\pi}{k_0} \Re(S_{vv} - S_{hh})$$

Here, the notations of $\Im$ and $\Re$ indicate the real and imaginary parts of a complex number, respectively. On the other hand, the backscattering amplitude matrix can be used to calculate $\mathbf{Z}$ in backscattering geometry. For example, the formulas of the matrix elements of $\mathbf{Z}$ used in this study are shown below:

$$Z_{11} = \frac{1}{2}\left(|S_{hh}|^2 + |S_{hv}|^2 + |S_{vh}|^2 + |S_{vv}|^2\right)$$

$$Z_{12} = \frac{1}{2}\left(|S_{hh}|^2 - |S_{hv}|^2 + |S_{vh}|^2 + |S_{vv}|^2\right)$$

$$Z_{21} = \frac{1}{2}\left(|S_{hh}|^2 + |S_{hv}|^2 - |S_{vh}|^2 - |S_{vv}|^2\right)$$

$$Z_{12} = \frac{1}{2}\left(|S_{hh}|^2 + |S_{hv}|^2 + |S_{vh}|^2 + |S_{vv}|^2\right)$$

$$Z_{22} = \frac{1}{2}\left(|S_{hh}|^2 - |S_{hv}|^2 - |S_{vh}|^2 + |S_{vv}|^2\right),\tag{3}$$

$$Z_{33} = \Re\left(S_{hh}S_{vv}^* + S_{hv}S_{vh}^*\right)$$

$$Z_{34} = \Im\left(S_{hh}S_{vv}^* + S_{vh}S_{hv}^*\right)$$

$$Z_{43} = \Im\left(S_{vv}S_{hh}^* - S_{hv}S_{vh}^*\right)$$

$$Z_{44} = \Re\left(S_{vv}S_{hh}^* - S_{hv}S_{vh}^*\right)$$

The complete set of equations from the elements of amplitude matrix $\mathbf{S}$ to each element of the Mueller matrix $\mathbf{Z}$ or the extinction matrix $\mathbf{K}$ can be found in Mishchenko (2014). The elements of $\mathbf{Z}$ and $\mathbf{K}$ are both in the dimension of $L^2$ (namely, they are usually expressed in the units of $mm^2$).

We could compute the bulk-scattering properties $\langle \mathbf{X} \rangle$[1] used in the expression of several polarimetric radar variables by performing size distribution integrations over elements of $\mathbf{Z}$ and $\mathbf{K}$ matrices:

$$\langle \mathbf{X} \rangle = \int_0^\infty \mathbf{X}(D)\,\mathrm{N}(D)\mathrm{d}D, \quad \mathbf{X} = \mathbf{Z}, \mathbf{K},\tag{4}$$

Here, $\mathrm{N}(D)$ represents the number concentration distribution function in units of $mm^{-1}\cdot m^{-3}$ over the particle spectrum, and $D$ is the diameter[2] of the hydrometeor in units of mm. The elements of bulk matrices $\langle \mathbf{Z} \rangle$ and $\langle \mathbf{K} \rangle$ were usually expressed in units of $mm^2\cdot m^{-3}$.

---

[1] Angle brackets are only used to indicate integration over particle size distribution in this study. Integration over azimuthal orientation and shape distributions are indicated by overlines.

[2] In the weather radar community, the particle diameter D usually refers to equal-volume-sphere diameter for a liquid hydrometeor, while D is often regarded as maximum dimension when describing solid and melting types of hydrometeors.

Note that we can only apply the particle ensemble integration over the complex amplitude scattering matrix, $\mathbf{S}$, in the forward

scattering geometry. This is because integrating over extinction matrix elements is proportional to integrating corresponding forward amplitude scattering matrix elements. For example:

$$\left\langle K_{11} \right\rangle = \frac{2\pi}{k_0} \left\langle S_{hh} \right\rangle + \frac{2\pi}{k_0} \left\langle S_{vv} \right\rangle, \tag{5}$$

Eq. (5) is derived by performing ensemble mean on Eq. (2).

Therefore, we use Mueller and extinction matrices to represent radar variables because ensemble means can be performed

directly on them, as is not the case for amplitude matrix. Also, they have a unified dimension of $L^2$. The LUTs in ZJU-AERO store Mueller and extinction matrices instead of the amplitude matrix (see Appendix B, which will be further described in Section 3.1). The equations for radar variables based on Mueller and extinction matrices can be found in Appendix D.

## 2.3 Hydrometeors specifications

**Table 1: An overview of the specification for all categories of hydrometeors in the Accurate and Efficient Radar Operator designed by Zhejiang University (ZJU-AERO). Some sophisticated specifications are only tagged with a bibliography and explained with more details in the context to make this table more compact and convenient for reference.**

| Hydrometeor Category | Shape Parameters | Mass-Diameter Relationship | Refractive Index Model | Orientation Preference | Particle Size Distribution |
|---|---|---|---|---|---|
| Rain[1] | Option A1: Spheroid (Brandes et al., 2002) — Option A2: Chebyshev (Chuang and Beard, 1990) | PSD model Using Equal-Volume-Sphere Diameter: $\mathrm{m}(D_{eq}) = aD_{eq}^b$ $(a = \frac{\pi}{6}\rho_w, b = 3)$ | Option B1: (Ellison, 2007) — Option B2: (Liebe et al., 1991) (deprecated) | $p(\boldsymbol{\beta}) \propto \sin(\boldsymbol{\beta}) \cdot$ $\exp\left(-\frac{\boldsymbol{\beta}}{2\sigma_{\boldsymbol{\beta}}^2}\right)$ $(\sigma_{\boldsymbol{\beta}} = 7\deg.)$ (Chandrasekar, 2001) | Option C1: (Marshall and Palmer, 1948) — Option C2: (Thompson et al., 2008) — Option C3: (Wang et al., 2016) — Option C4: (Abel and Boutle, 2012) — Option C5: (Walters et al., 2011) |
| Snow | Spheroid Field research by Garrett et al. (2015); Fitted by (Wolfensberger and Berne, 2018) | Option A1: $\bar{\mathrm{m}}(D_{\max}) = aD_{\max}^b$ $(a = \frac{\pi}{6}\rho_{snow}, b = 3)$ — Option A2: $\bar{\mathrm{m}}(D_{\max}) = \rho_{ice} \cdot \overline{\mathrm{V}}_{ice}(D_{\max})$ | Air–ice matrix using Maxwell–Garnett effective medium approximation | $p(\boldsymbol{\beta}) \propto \sin(\boldsymbol{\beta}) \cdot$ $\exp\left(-\frac{\boldsymbol{\beta}}{2\sigma_{\boldsymbol{\beta}}^2}\right)$ $(\sigma_{\boldsymbol{\beta}}(D_{max}) = 40.0D_{max}^{-0.077}\deg.)$ (Garrett et al., 2015) | Temperature-dependent PSD model: (Field et al., 2005) |
| Graupel | Spheroid Field research by Garrett et al. (2015); Fitted by (Wolfensberger and Berne, 2018) | Option A1: $\overline{\mathrm{m}}(D_{\max}) = aD_{\max}^b$ $(a = \frac{\pi}{6}\rho_{graupel}, b = 3)$ — Option A2: $\overline{\mathrm{m}}(D_{\max}) = \rho_{ice} \cdot \overline{\mathrm{V}}_{ice}(D_{\max})$ | Air–ice matrix using Maxwell–Garnett effective medium approximation | $p(\boldsymbol{\beta}) \propto \sin(\boldsymbol{\beta}) \cdot$ $\exp\left(-\frac{\boldsymbol{\beta}}{2\sigma_{\boldsymbol{\beta}}^2}\right)$ $(\sigma_{\boldsymbol{\beta}}(D_{max}) = 58.07D_{max}^{-0.11}\deg.)$ (Garrett et al., 2015) | Option B1: Microphysics scheme WSM6 (Hong and Lim, 2006) — Option B2: Microphysics scheme Thompson (Thompson et al., 2008) |

[1] The specifications regarding the hydrometeor category of rain are discussed in Section 3.

Table 1 summarizes the hydromteor specifications in ZJU-AERO, with the following important notes:

(1). During the early development stage of ZJU-AERO, we initially used the dielectric model for water proposed by Liebe et al. (1991). However, we transitioned to a more accurate and contemporary dielectric model developed by Ellison (2007). Nevertheless, we retained the outdated option and optical property look-up table from the old dielectric model in our archive for future reference and comparison (see the column of refractive index model in Table 1).

(2). In principle, it is encouraged to use PSD schemes compatible with the microphysics package in the NWP model to ensure consistent hydrometeors settings in simulations. Therefore, ZJU-AERO, which is designed for CMA–GFS/MESO, provides PSD options that are compatible with the single-moment microphysics scheme WSM6 (Hong and Lim, 2006) and Thompson

(Thompson et al., 2008), which are widely used in global and regional operational models of CMA. For instance, option C1 for rain and option B1 for graupel are compatible with the WSM6 package, while option C2 for rain and option B2 for graupel are compatible with the Thompson package (see the column of particle size distribution in Table 1). However, for the snow category, we implemented the PSD scheme of Field et al. (2005) as the only option since it is the widely acknowledged as the best globally applicable temperature-dependent PSD model for solid precipitation. Additionally, we have provided users with some additional PSD schemes for sensitivity assessment, such as option C3 (Wang et al., 2016) and C4 (Abel and Boutle, 2012) for the rain category. Those PSD schemes in the ZJU-AERO that are incompatible with the microphysics package used in the NWP model are referred to as "forced" PSD schemes.

(3). Solid hydrometeor categories, such as snow and graupel, are known to have relatively larger uncertainties associated with parameterizations of aspect ratios and orientation preference. To address these uncertainties, a field research study conducted by Garrett et al. (2012) used an *in-situ* observation instrument called the multi-angle snowflake camera (MASC). This instrument was used to measure the aspect ratios and orientations of over 30,000 solid particles in the Eastern Swiss Alps. The particles were then classified into aggregates (corresponding to the snow category in this study), rimed, and graupel, as described in Garrett et al. (2015). The fitted model from this study was used as a *priori* knowledge of hydrometeor shape specifications in the ZJU-AERO (see the column of shape parameters in Table 1):

$$p(\gamma; D_{max}) \propto (\gamma - 1)^{K(D_{max})-1} \exp\left(-\frac{\gamma - 1}{\Theta(D_{max})}\right), \qquad (6)$$

Eq. (6) provides the probability distribution function of the aspect ratio, in which $\gamma$ is the reciprocal of the aspect ratio (minor axis / major axis, always less than unity). It is assumed to follow a gamma distribution with an offset coefficient of 1 (i.e., $\gamma > 1$). The shape coefficient, $K(D_{max})$, and a scale coefficient, $\Theta(D_{max})$, are functions of particle maximum dimension.

We used the power-law relationships of $K(D_{max})$ and $\Theta(D_{max})$ that were fitted by Wolfensberger and Berne (2018):

$$K_{snow}(D_{max}) = 8.42 D_{max}^{-0.57}; \Theta_{snow}(D_{max}) = 0.053 D_{max}^{-0.79}$$
$$K_{graupel}(D_{max}) = 3.2 D_{max}^{-0.42}; \Theta_{graupel}(D_{max}) = 0.074 D_{max}^{-0.67}, \qquad (7)$$

(4). Since solving PSD parameters from NWP prognostic hydrometeor mass concentrations (for bulk microphysics) requires knowledge of the mass of various-sized particles, the mass–diameter relationship is crucial in determining the PSD (see the column of mass-diameter relationship in Table 1). In this study, all the hydrometeor categories follow the gamma distribution (the widely used exponential distribution is just a special case of gamma distribution when $\mu = 1$):

$$N(D) = N_0 D^{\mu} e^{-\Lambda D} \qquad (8)$$

in which $N_0$ is the intercept, $\Lambda$ is the slope and $\mu$ is the shape coefficient of that gamma distribution. As is often the case (for all PSD options in ZJU-AERO, see Table 1), $\mu$ is a prescribed constant in the microphysics package, while $N_0$ either

equals a prescribed constant or relates to $\Lambda$ by a power-law, in which $x_1$ and $x_2$ are parameters fitted by drop size distribution (DSD) observations (see Section 3.4):

$$N_0 = x_1 \Lambda^{x_2} \tag{9}$$

If a hydrometeor category is of single-moment microphysics scheme, given the mass concentration of arbitrary hydrometeor category "x" $Q_x$ in units of kg·m$^{-3}$:

$$Q_x = \int_0^\infty \overline{m}(D) \cdot N(D) dD , \tag{10}$$

If the mass–diameter relationship can be approximated as power-law form $\overline{m}(D) = aD^b$, in which $\overline{m}(D)$ is the average mass of a given size of a particle (considering that some hydrometeor categories have a probability distribution over shape parameters such as aspect ratio). Then we can solve the unknown parameter $\Lambda$ and determine all relevant PSD information that pertains to that radar gate analytically by plugging Eq. (8) into Eq. (10):

$$\Lambda = \left[ \frac{a x_1 \Gamma(1+b+\mu)}{Q_x} \right]^{\frac{1}{1+b+\mu-x_2}} , \tag{11}$$

Again, there is a microphysics-consistent mass–diameter relationship $\overline{m}(D_{max}) = \frac{\pi}{6} \rho_{sp} D_{max}^3$ ($\rho_{sp}$ is the overall density of the sphere solid precipitation particle) for snow and graupel. Many microphysics schemes, such as WSM6, simply treated solid precipitation categories as spheres with different ice–air mixture ratios and hence different overall densities $\rho_{sp}$.

However, this practice can result in a huge inconsistency between the mass concentration represented by radar operators and the microphysics schemes since the actual average mass of a given size bin is represented by the following probability-weighted averaging over the aspect ratio for solid hydrometeors as shown below:

$$\overline{m}(D_{max}) = \int_1^\infty p(\gamma; D_{max}) m(\gamma; D_{max}) d\gamma , \tag{12}$$

It turns out that fitting Eq. (12) into power-laws is troublesome when the probability distribution function $p(\gamma; D_{max})$ varied dramatically with diameter. This problem will become even worse when we introduce other non-spherical particles, such as snowflakes.

To resolve this error when using a traditional mass-diameter relationship, we implemented a benchmark PSD solver employing a numerical method (namely, Newton–iteration) for particles with complicated morphology specifications. The

simplest case, a PSD represented by an exponential distribution with a fixed intercept parameter $N_0$, can be used as an

example. Here the equation for unknown PSD parameter $\Lambda$ can be expressed as follows:

$$Q_x = \sum_{D_i=D_l}^{D_i=D_u} \overline{\mathrm{m}}(D_i) \mathrm{N}(D_i) \Delta D = \sum_{D_i=D_l}^{D_i=D_u} \overline{\mathrm{m}}(D_i) N_0 e^{-\Lambda D_i} \Delta D, \tag{13}$$

in which the integration in Eq. (10) is truncated within a specific diameter range of $[D_l, D_u]$ and further discretized as a

summation. The expression of the exponential distribution is then substituted in for the second equality. $\overline{\mathrm{m}}(D_i)$ at

discretized size bins is precomputed by Eq. (12) and treated as a constant. The mass $\mathrm{m}(\gamma; D_i)$ for each morphology

specifications can be calculated as the density of the pure ice $\rho_{ice}$ multiplied by the volume occupied by the non-spherical

hydrometeor particle model $\mathrm{V}(\gamma; D_i)$ (calculated from mathematics formulas of geometrical bodies).

We then define the function $F(\Lambda)$ and its derivative $F'(\Lambda)$ in Newton iteration:

$$\begin{aligned} F(\Lambda) &= \sum_{D_i=D_l}^{D_i=D_u} \overline{\mathrm{m}}(D_i) N_0 e^{-\Lambda D_i} \Delta D - Q_x \\ F'(\Lambda) &= -\sum_{D_i=D_l}^{D_i=D_u} \overline{\mathrm{m}}(D_i) N_0 e^{-\Lambda D_i} D_i \Delta D \end{aligned}, \tag{14}$$

Based on the above formulation, the iteration relationship to derive the (n+1)-th guess $\Lambda_{n+1}$ from the n-th guess $\Lambda_n$ can be

expressed as shown below:

$$\Lambda_{n+1} = \Lambda_n - \frac{F(\Lambda_n)}{F'(\Lambda_n)}, \tag{15}$$

While performing iterations online (the benchmark PSD solver) may lead to a decrease in the efficiency of the ZJU-AERO, this problem can be resolved by using bulk-scattering properties (BSP) LUTs instead of single-scattering properties (SSP) LUTs (see Section 3.5).

**3 Database of Hydrometeor Optical Properties**

In Section 2, we provide a comprehensive description of the design of ZJU-AERO. Specifically, we emphasize that the hydrometeor optical properties database includes the elements of $\mathbf{Z}$ and $\mathbf{K}$ in units of mm$^2$ for single-scattering properties and those of $\langle \mathbf{Z} \rangle$ and $\langle \mathbf{K} \rangle$ in units of mm$^2 \cdot$m$^{-3}$ for bulk-scattering properties, both in the FSA convention. In the first two subsections, we will delve into the design of the database (LUT) in ZJU-AERO with more details.

Furthermore, ZJU-AERO currently encompasses three types of hydrometeors: (1) rain, (2) snow, and (3) graupel. Among these hydrometeors, the rain category offers a non-spherical shape parameter option, known as the Chebyshev shape. This shape differs from the traditional spheroid shape commonly used in other radar observation operators. Therefore, we will evaluate the database contents using the Chebyshev raindrop as an example in the last three subsections.

## 3.1 The Multi-Layered Architecture

The ZJU-AERO optical property database is designed with a multi-layered architecture consisting of three levels: (1) the raw single-scattering properties (RSSP) database (level A), (2) the applied single-scattering properties (ASSP) database (level B), and (3) the BSP database (level C). These levels are described in detail in Table 2.

The RSSP-Level A database contains the optical properties of individual particles without any averaging or integration over the shape parameters and orientations, which normally consumed a significant amount of memory resources ($\sim 10^1$GB).

However, using the RSSP in ZJU-AERO would require online integration of orientations and shape parameters, leading to a significant slow down in radar operator performance. On the other hand, if shape and orientation averaging were applied during the the creation of the database and the raw optical data (RSSP) are discarded, the resulting database would lack the flexibility needed for modifying the shapes and orientations distributions. Additionally, future enhancements to the ZJU-AERO database may involve incorporating more sophisticated shape parameters for the non-spherical hydrometeors. Therefore, retaining the

RSSP-Level A database is essential to accommodate uncertainties and increase the convenience in experiments and simulations related to shape and orientation parameters.

Building on the RSSP-Level A database, the ASSP-LevelB database improves the computational efficiency by carrying out averaging or orientation over shape parameters and orientations offline. Finally, the BSP-Level C optical database integrates the optical properties stored in the ASSP-Level B database over PSD and enables fast batch runnings in ZJU-AERO for

operational use. In summary, the multi-layered architecture of the lookup table (stored in netCDF4 format files separately for different database levels) is to strike a balance between the flexibility of the database and the computational efficiency required by radar operators. Future releases of ZJU-AERO will provide software tools for flexible conversions between the database levels, allowing users to easily configure integration parameters.

**Table 2: The architecture design of the multi-layered hydrometeor optical properties database used in ZJU-AERO. The column "Volume" gives an estimation of the external storage that a single database lookup table file for one hydrometeor class and one frequency occupies. Note that only the order of magnitude of storage space is shown in those entries. The grids of dimensions in this table can be found in Table B1 of Appendix B.**

| Database Layer | Dimensions | Variables | Volume |
|---|---|---|---|
| Raw Single Scattering Properties (RSSP) Database - Level A | Shape parameters: e.g., reciprocal of aspect ratio $\gamma$<br>Scattering geometry: elevation $e$, Euler angle $\beta$<br>Temperature: $T$<br>Diameter: $D_{eq}$ / $D_{max}$ | $\mathbf{Z}$ and $\mathbf{K}$<br>in units of mm$^2$ | ~$10^1$ GB |
| Applied Single Scattering Properties (ASSP) Database - Level B | Scattering geometry: elevation $e$<br>Temperature: $T$<br>Diameter: $D_{eq}$ / $D_{max}$ | $\mathbf{Z}$ and $\mathbf{K}$<br>in units of mm$^2$ | ~$10^1$ MB |
| Bulk Scattering Properties (BSP) Database - Level C | Scattering geometry: elevation $e$<br>Temperature: $T$<br>Mass concentration: $Q_x$<br>(Different PSD schemes in separate files) | $\langle\mathbf{Z}\rangle$ and $\langle\mathbf{K}\rangle$<br>in units of mm$^2\cdot$m$^{-3}$ | ~$10^1$ MB |

## 3.2 Scattering Geometry

Figure 3 depicts the Cartesian coordinate system used to determine the scattering geometry of a hexagonal plate particle. The laboratory coordinate system, denoted as $OX_LY_LZ_L$, is aligned by vertically positioning its $OZ_L$ axis and placing its $OX_L$ axis in the vertical plane specified by the incident radar beam and $OZ_L$. This alignment sets the azimuthal angle $\phi_{inc}$ of the radar beam to 0. In Figure 3(a), the direction of the radar beam is shown, which is practically specified using the elevation angle

$e \in [-\pi/2, \pi/2]$. For ground-based radar, this angle is positive, while for space-borne radar, it is negative. In this context, the polar angle of the radar beam, denoted as $\theta_{inc}$, is related to the elevation angle through $\theta_{inc} = \pi/2 - e$. The shape of particle is defined in the particle coordinate system $OX_PY_PZ_P$. In the specific example shown in Figure 3(d), the $OZ_P$ axis is set perpendicular to the basal face of a hexagonal plate. The origin O is placed at the mass center of the particle, and the $OY_P$ axis intersecte two opposite vertices of the hexagonal basal face. Using the ZYZ convention of the Euler angles specified by α, β,

and γ (rotations were performed with respect to the $OZ_P$, $OY_P$, and $OZ_P$ axis, respectively), any arbitrary orientation of a given particle can uniquely be determined (see Figure 3(b)–(d)).

With a specified scattering geometry, we can now outline the procedures for computing scattering properties of particles with fixed orientations (steps 1-3) and then integrate over them for scattering properties with specific orientation preference (step 4):

1.     We used the T-matrix method to compute the T-matrix only once for a given particle and a radar beam wavelength. The guidelines for selecting scattering computation approach are as follows: for both axially symmetric (i.e., the dielectric constant distribution of electromagnetic medium in spherical coordinates $\varepsilon(r, \theta, \varphi)$ is irrelevant with azimuth angle $\varphi$) and homogenous particles, we used the EBCM T-matrix code (Mishchenko and Travis, 1994), while for particles without axial symmetry or those that are inhomogeneous (or shapes that the EBCM approach suffer from numerical stability

issues), we applied the invariant-imbedding T-matrix (IITM) code (Bi et al., 2013; Bi and Yang, 2014; Bi et al., 2022; Wang et al., 2023).

    2.     With the T-matrix computed in Step 1, we efficiently calculated the forward and backward amplitude scattering matrix $\mathbf{S}_{\mathrm{FSA}}$ in Eq. (1) for tuples of scattering geometry ($\alpha$, $\beta$, $\gamma$, and e) using the method of Mishchenko (2000).

    3.     The forward and backward amplitude scattering matrices $\mathbf{S}_{\mathrm{FSA}}$ were then converted into the backscattering Mueller and

extinction matrices, $\mathbf{Z}$ and $\mathbf{K}$, respectively (Mishchenko, 2014).

    4.     When averaging over $\alpha$ and $\gamma$, we considered that the atmosphere is generally horizontally isotropic on the scale of hydrometeor particles. It is believed that they should have no preference for Euler angles $\alpha$ and $\gamma$, except for extreme conditions such as lightening-induced reorientation of ice crystals (Hubbert et al., 2010). Therefore, we performed internal averaging over $\alpha$ and $\gamma$ for elements of $\mathbf{Z}$ and $\mathbf{K}$ in the scattering computation code before generating the RSSP – Level

A database:

$$\overline{\mathbf{X}}(e, \beta) = \frac{1}{4\pi^2} \int_0^{2\pi} \mathrm{d}\alpha \int_0^{2\pi} \mathrm{d}\gamma \, \mathbf{X}(e, \alpha, \beta, \gamma), \mathbf{X} = \mathbf{Z}, \mathbf{K},^3 \tag{16}$$

Hence, the Level A database only has two residual scattering geometry dimensions: (1) **e** and (2) **β**, allowing for a reasonable volume for archiving.

The integration in the Level A to Level B database conversion tool can be formalized as integration over canting angle β

as follows:

$$\overline{\overline{\mathbf{X}}}(e) = \frac{1}{\pi} \int_0^{\pi} \sin(\beta) \mathrm{d}\beta \, \mathrm{p}(\beta) \overline{\mathbf{X}}(e, \beta), \mathbf{X} = \mathbf{Z}, \mathbf{K},^4 \tag{17}$$

Here $\mathrm{p}(\beta)$ is the probability distribution of canting angle β. A Gaussian distribution in polar angle is often used to approximate the probability distribution of canting angle:

---

[3] Overlines over elements of $\mathbf{Z}$ and $\mathbf{K}$ are omitted for brevity for level A database elements elsewhere in this paper.

[4] Two overlines over elements of $\mathbf{Z}$, $\mathbf{K}$, $\langle\mathbf{Z}\rangle$, and $\langle\mathbf{K}\rangle$ are omitted for brevity for level B/C database elements elsewhere in this paper, and in formulas of polarimetric radar variables.

$$p(\beta) = \exp\left(-\frac{\beta}{2\sigma_\beta^2}\right),\tag{18}$$

Here the standard deviation of canting angle, $\sigma_\beta$, can be found in the orientation preference column of Table 1.

Particle symmetry can be considered in the orientation averaging of Eqs. (16) and (17) to avoid redundant evaluation of $\mathbf{Z}$ and $\mathbf{K}$. For example, in the case of a hexagonal plate with 6-fold azimuthal symmetry and xy-plane reflectance symmetry, $\mathbf{Z}$ and $\mathbf{K}$ only need to be evaluated and averaged over $\alpha \in [0, \pi/6]$ and $\beta \in [0, \pi/2]$.

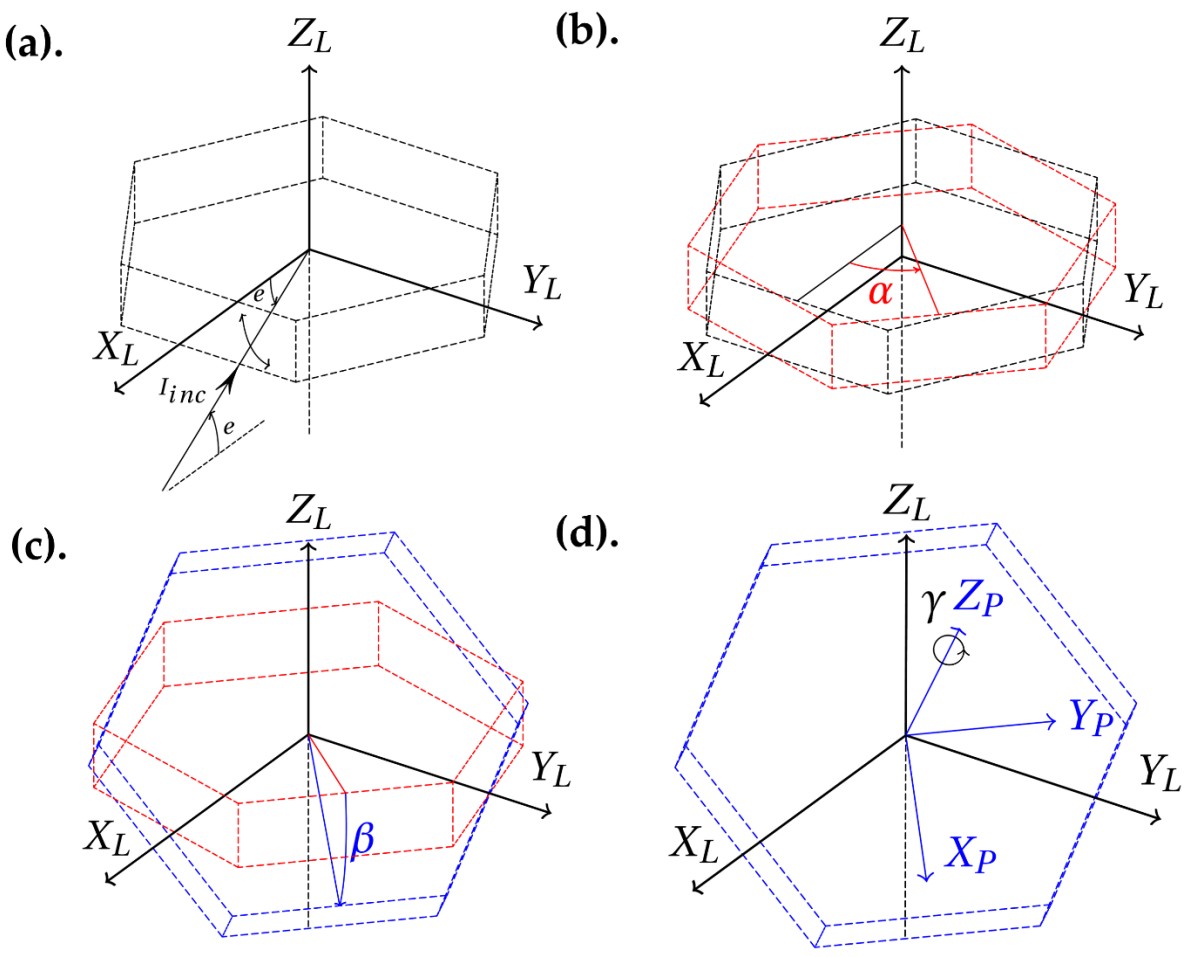


Figure 3: Illustration of the orientation preference of a particle specified with Euler angles (α, β, γ) in which a hexagonal plate is used as an example. Panel (a) shows the laboratory coordinate system OX_LY_LZ_L and the particle with reference orientation (α=β=γ=0). The incident beam comes from the elevation angle of e. Panels (b), (c), and (d) depicts how angles α, β, and γ uniquely determine the orientation of a particle, respectively. The particle coordinate system OX_PY_PZ_P is shown in panel (d) with solid blue
lines.

### 3.3 Raw Single Scattering Properties: Level A

To improve the accuracy of raindrop shape representation in the forward radar operator ZJU-AERO, it is essential to apply an established raindrop model that accounts for various effects such as the surface tension, hydrostatic and aerodynamic pressures, and static electric forces. By incorporating these factors, a more accurate geometric representation of observed raindrop shapes can be achieved compared to the most commonly used spheroid model, as proved by photographs of water drop falling at the terminal velocity in air (Pruppacher et al., 1998). In a study conducted by Chuang and Beard (1990), an equilibrium differential equation was utilized to iteratively determine the shape of a falling raindrop with a given mass, which corresponds to a specific diameter $D_{eq}$. The obtained results were then fitted to Chebyshev polynomials (Chuang and Beard, 1990):

$$\chi(\theta) = \frac{D_{eq}}{2}\left[1 + \sum_{i=0}^{10} a_n(D_{eq})\left(1 + \cos(n\theta)\right)\right],$$
(19)

This eqation involves the distance between the raindrop's surface to its mass center, denoted as $\chi(\theta)$, where $\theta$ is the zenith angle in spherical coordinates (see Figure 4(d)). Chuang and Beard (1990) provided Chebyshev coefficients $a_n(D_{eq})$ on diameter grids. Therefore, for a given $D_{eq}$ value, the corresponding Chebyshev coefficients can be obtained through interpolation.

The Chebyshev model of raindrops deviates from xy-plane reflectance symmetry, resulting in a flatter bottom surface compared to the spheroid. Conversely, the top surface of raindrop described by the Chebyshev model is sharper. Note that the disparity between the two models becomes more pronounced with increasing $D_{eq}$, as shown in Figure 4(a)–(c). It can be noted from Figure 4(c) that larger raindrops are more prone to aerodynamic effects and therefore having a flatter base compared with the reference "spheroid".

Comparing the optical properties of two shapes with significantly different aspect ratios cannot reveal the effects resulting from introducing Chebyshev shapes. To address this, we defined the aspect ratio of the Chebyshev model in Figure 4(d), which represents the vertical maximum dimension $b'$ versus the horizontal maximum dimension $a'$. Figure 5 illustrate a comparison between the aspect ratio of the Chebyshev model, as defined above, and the aspect ratio of the commonly used spheroid raindrop model (Thurai et al., 2007; Brandes et al., 2002). It is apparent that for common raindrops with $D_{eq}$<8 mm, the aspect ratios of the two models are approximately equal. Therefore, we can infer that the differences in optical properties between the spheroid and Chebyshev models arise from higher-order shape specifications rather than the first-order aspect ratio parameter (also proved by comparisons between two shapes with identical aspect ratio, figure not shown).

A detailed examination of the optical properties of Chebyshev model rain droplets and their sensitivity against traditional spheroid model were conducted by Ekelund et al. (2020). To compare the radar variables between the spheroid and Chebyshev models, they used an extended precision version of the EBCM T-matrix code (Mishchenko and Travis, 1994). However, it is important to note that the EBCM may encounter numerical instability issues due to the extremely high imaginary part of refractive indices for liquid water around the K band (10~40 GHz), where the Ku- and Ka-bands are located. To ensure integrity

and accuracy, this study presents an optical property database (with a user-friendly radar operator interface) for the Chebyshev raindrop model at the Ku- and Ka-bands (13.6 and 35.5 GHz, respectively) using the IITM code (Bi et al., 2013).

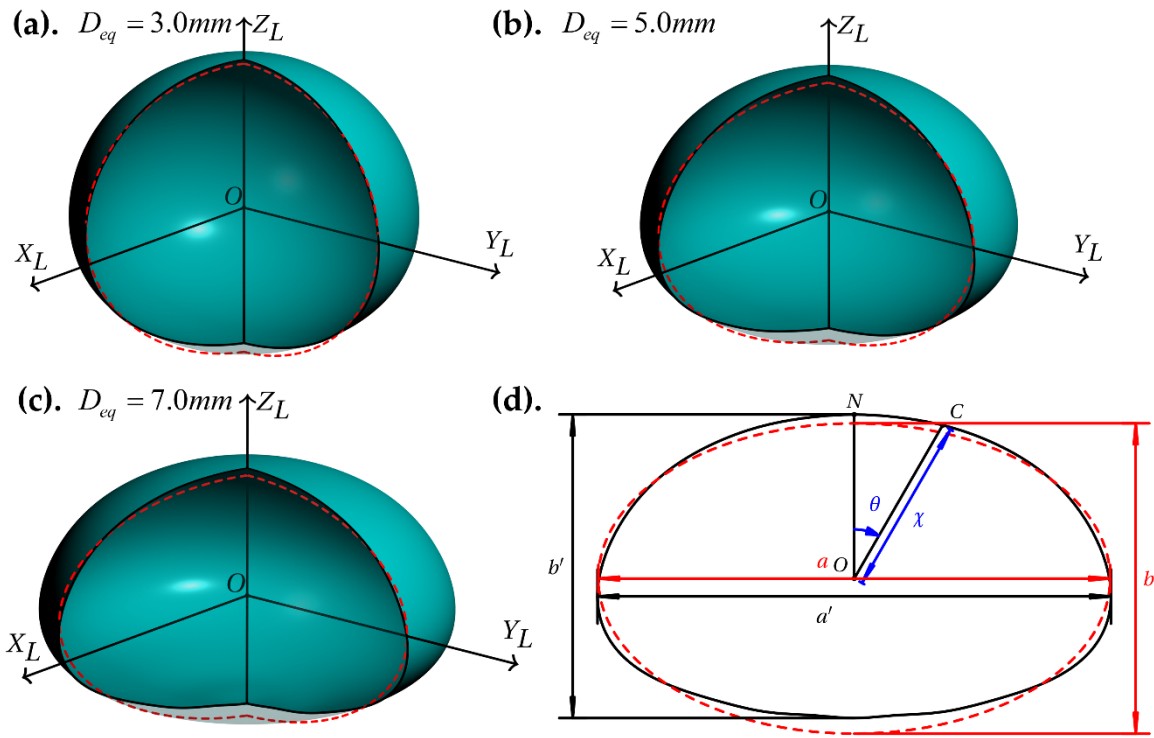


**Figure 4: Illustration of raindrops described based on the Chebyshev model. Panels (a), (b), and (c) display the shapes of the Chebyshev raindrop with equal-volume-sphere diameters ($D_{eq}$) of 3, 5, and 7 mm, respectively. Panel (d) shows the vertical cross-section of the Chebyshev shape corresponding to panel (c), and it also illustrates how the aspect ratio $b'/a'$ is defined for a Chebyshev raindrop. The dashed red lines in all panels show the spheroid model with identical $D_{eq}$ for comparison. The aspect ratio of the**
**spheroid is defined as $b/a$ and $b$ and $a$ are shown in panel (d). The Chebyshev shapes in panels (a)–(c) were set to be partially transparent and displayed in an azimuth angle range [-π/6, π/2].**

To better understand the impact of single-scattering properties on radar variables, it is necessary to analyze the RSSP-Level A database. Additionally, we choose to introduce intermediate quantities called the "SSP factors for radar variables", which are illustrative and facilitate our understanding of how SSP of a given-sized particle affects radar variables.

The SSP factor "[$z_h$]" for the horizontal reflectivity "$z_h$" is defined as:

$$[z_h] = \frac{(Z_{11} - Z_{12} - Z_{21} + Z_{22}) \cdot (2\lambda^4)}{\pi^4 |K_w|^2},$$   (20)

Here, the quantity enclosed in square bracket "[$z_h$]" indicates the SSP factor of the radar variable "$z_h$". we can perform particle ensemble mean on it, which is often indicated by angle brackets (Zhang, 2016):

$$\langle[z_h]\rangle = \frac{2\lambda^4}{\pi^4 |K_w|^2}\left(\langle Z_{11}\rangle - \langle Z_{12}\rangle - \langle Z_{21}\rangle + \langle Z_{22}\rangle\right) = z_h, \tag{21}$$

Here, the last equal is the definition of horizontal reflectivity (see Eq. (D1) in Appendix D).

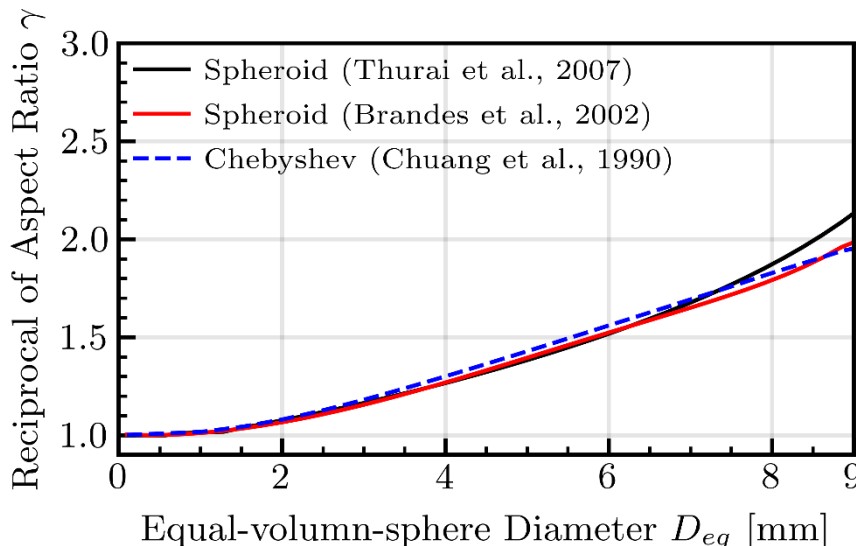

**Figure 5: The aspect ratio–diameter relationships as reported in literature. The vertical axis is the reciprocal of the aspect ratio (i.e., γ). Thurai et al. (2007) and Brandes et al. (2002) fitted _in-situ_ measurements of two-dimensional video disdrometer (2DVD)**
**measurements by segmented polynomials to give the explicit γ-$D_{eq}$ expressions, while the aspect ratios of the Chebyshev raindrop**
**are estimated with the Chebyshev coefficients recorded by Chuang and Beard (1990) with the method mentioned in Figure 4(d).**
**Since raindrops with their $D_{eq}$ larger than 8 mm are beyond the typical size range (Zhang, 2016) and rarely found in nature**
**(Kobayashi and Adachi, 2001), the relationships given by Brandes et al. (2002) and Chuang and Beard (1990) end at 9 mm.**

Similarly, the SSP factor "[$a_h$]" for horizontal attenuation coefficient "$a_h$" is defined (the extinction cross section by essence):

$$[a_h] = K_{11} - K_{12} = \sigma_{ext,h}, \tag{22}$$

If we perform particle ensemble mean on it, we find it is proportional to the definition of horizontal attenuation coefficient

(see Eq. (D5) in Appendix D):

$$\langle[a_h]\rangle = \left(\langle K_{11}\rangle - \langle K_{12}\rangle\right) = 10^3 \cdot a_h, \tag{23}$$

As shown by Eqs. (21) and (23), these factors ([$z_h$] and [$a_h$]) are equivalent radar parameters of radar reflectivity $z_h$ and specific

attenuation $a_h$, respectively, for a single particle of size $D_{eq}$. For further SSP factors for level A database diagnoses, please refer

to Figure 8.

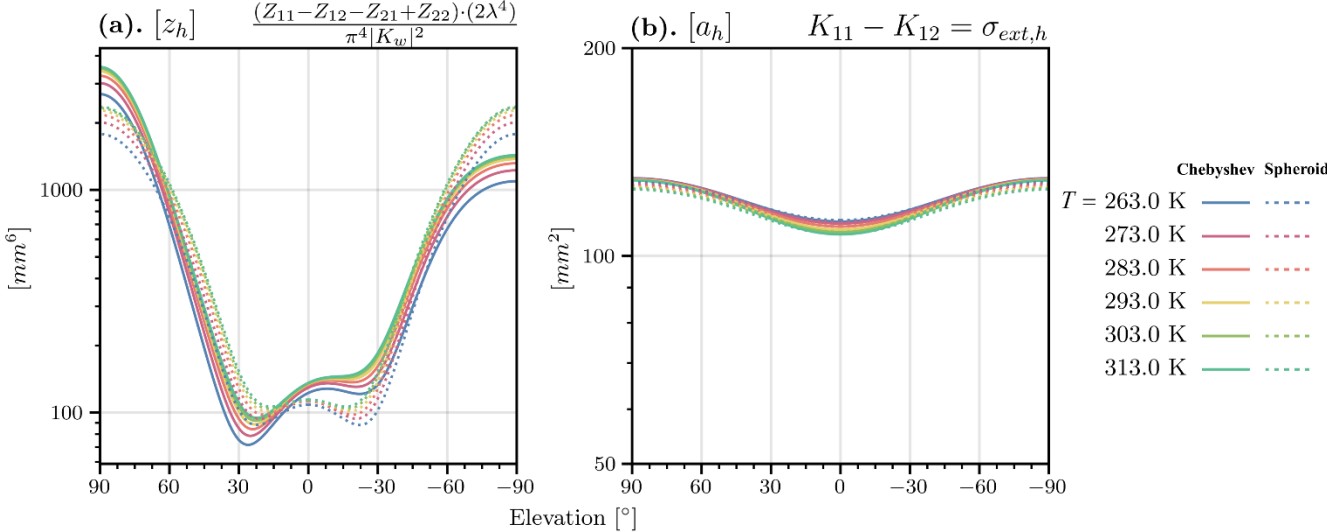

**Figure 6: (a).** SSP factor of horizontal reflectivity and **(b).** SSP factor of horizontal attenuation against the elevation angle of radar beam for a raindrop with an equal-volume-sphere diameter of 7.0 mm at the Ka band (35.5 GHz), with their expression in the upper right corner of the panels. The SSP factors of the Level A database are displayed for different temperatures with lines in different colors. The results of the Chebyshev raindrop analysis are indicated with solid lines while those of spheroid raindrop are indicated with dotted lines. The database entries for β = 0° are shown. Negative elevation angles indicate that the beams are pointing (slanting) downwards. The variations with respect to temperature entirely originate from temperature dependence of water dielectric constants.

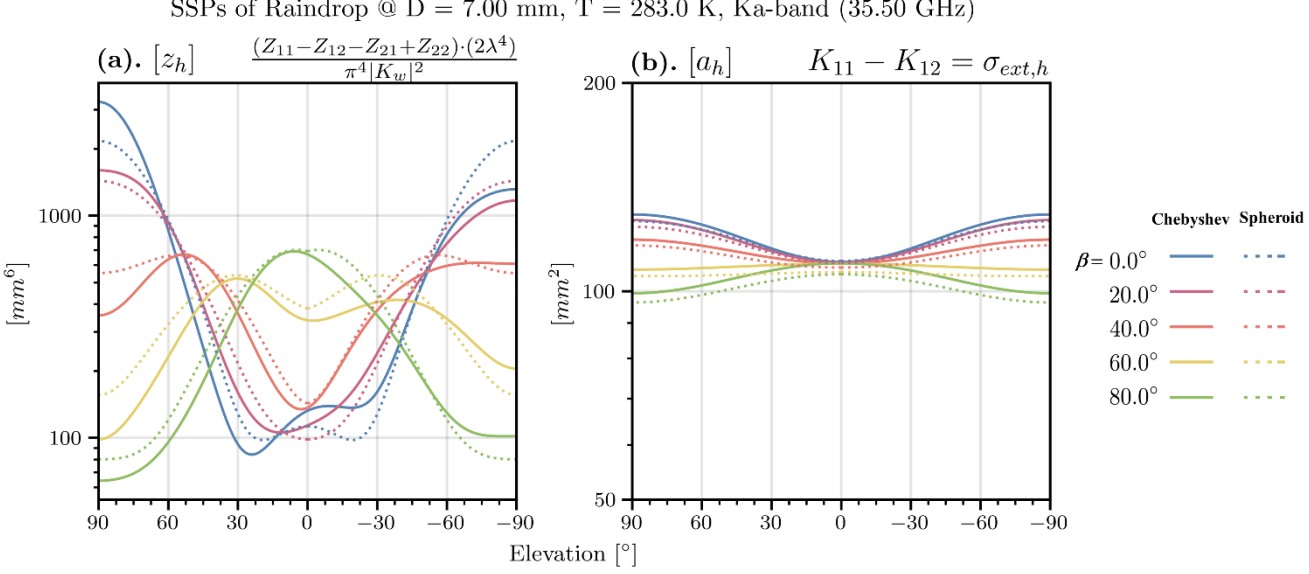

**Figure 7:** The results for different orientation angles (β) are displayed with different colours in a manner resembling Figure 8. The database entries for temperature=10 °C (283 K) are shown.

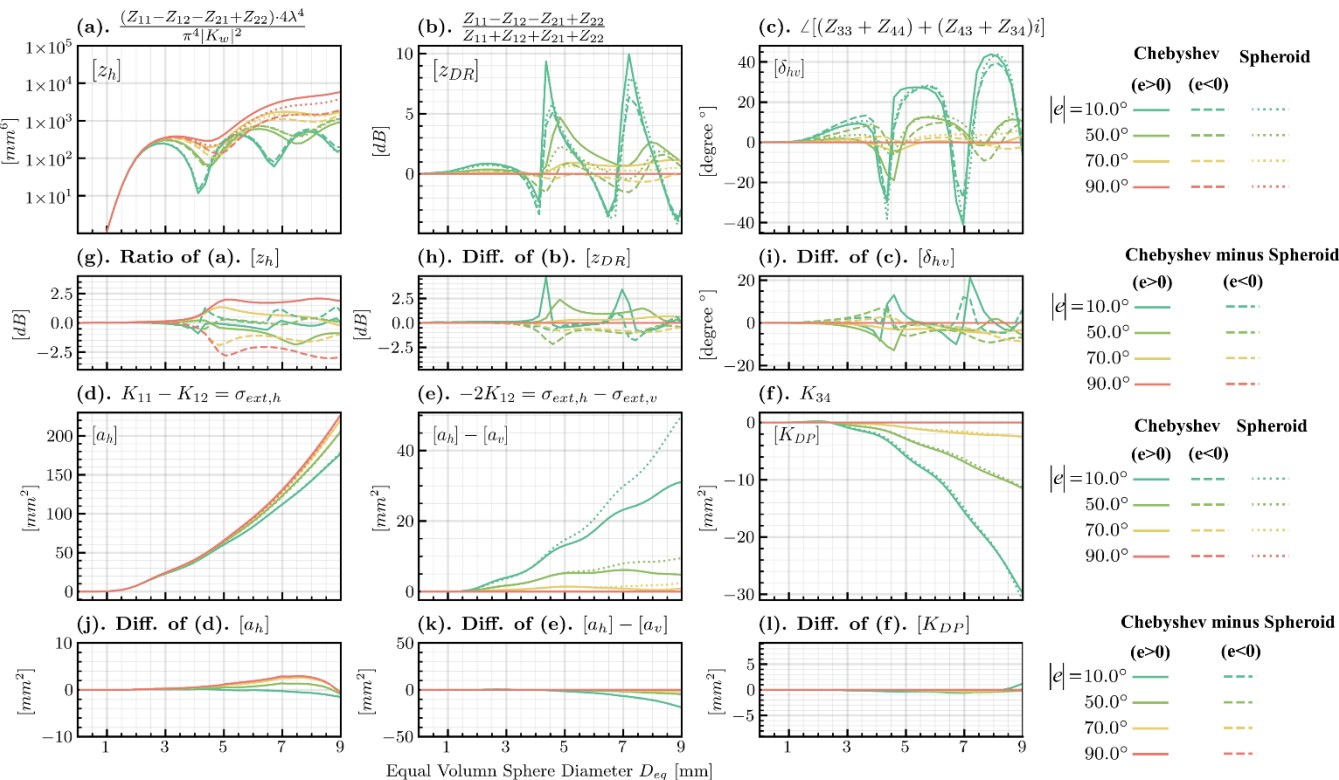

**Figure 8:** The SSP factors of intrinsic radar variables (indicated in the upper left corner) for the Chebyshev model and the spheroid raindrop models against the equal-volume-diameter of the liquid particles at the Ka band (35.5 GHz) and 283 K. Lines with different colours display the SSP factors for different elevation angles of the radar beam. Results of positive elevation angles for spheroids are indicated by solid lines while those of negative elevation angles are indicated by dashed lines. Since the results of spheroids have xy-plane reflectance symmetry, the negative and positive results are merged as dotted lines. The flatter auxiliary panels (g)–(l) (the second and the fourth rows) below the major panels (a)–(f) (the first and third rows) display the corresponding differences (or ratios) of optical properties beween the Chebyshev model and the spheroid models, respectively. The first column shows horizontal polarization SSP factors ([$z_h$] and [$a_h$]) mentioned in Figure 6 and Figure 7, the second column shows SSP factors contributing to observed differential reflectivity ([$z_{DR}$] and [$a_h$]- [$a_V$]), and the third column shows the SSP factors contributing to the observed total phase shift ([$\delta_{hv}$] and [$K_{DP}$]).

It is not surprising to observe that the scattering properties exhibited symmetry with respect to the radar beam elevation angle, e = 0°, for the spheroid model. This is because spheroids possess reflectance symmetry in the xy-plane. However, this symmetry does not hold true for the Chebyshev model. We find that the SSP factor $\left[z_h\right]$ has a more pronounced peak than spheroids near the nadir (specifically, e = 90°, which is frequently encountered for the vertical-profiling cloud radar), as shown in Figure 6(a). Conversely, the peak near the zenith (namely, e = –90°, for the space-borne radar) is weaker. This phenomenon is also found for the Chebyshev model at 94.1 GHz (Ekelund et al., 2020), which can be attributed to its flatter bottom surface and sharper top surface geometries in Ekelund et al. (2020), as depicted in Figure 4. Additionally, we find that the $\left[z_h\right]$ factor for ground-based scattering geometries (e = 0~20°) is close to the values obtained for the spheroid model. However, deviations

from the spheroid model were typically much smaller for lower frequency bands, such as the Ku band (13.6 GHz) (figure not shown here).

However, the forward scattering properties such as $[a_h]$ of the Chebyshev model still maintain their symmetry with respect to the beam elevation angle e=0°, which can be easily understood based on the reciprocal theory (Van De Hulst, 1981). It is important to note that the attenuation effects of Chebyshev raindrops are slightly stronger compared to spheroid ones for near nadir and zenith scattering geometries.

Figure 6 also demonstrates the stability of the deviations between the Chebyshev model and the spheroid model (CmS hereafter) in terms of the temperature dimension. However, the finding is different for $[z_h]$ against the orientation preference dimension β (see Figure 7a). It can be interpreted that varying orientation presents a significantly different "view" of raindrop to the observer, while varying temperature essentially keeps the same "view" of the raindrop with different dielectric constants. It was found that the positive and negative CmSs of $[z_h]$ at nadir and zenith, respectively, hold true only if β < 20°. In cases where particle groups have larger canting angles (β > 20°), the CmS for $[z_h]$ can produce a reversal of their signs (Figure 7a). Since the column "orientation preference" in Table 1 shows that the standard deviation of β = 7° for raindrops in normal turbulence conditions does not exceed this threshold for raindrops, we can infer that the CmS deviations found for β = 0° only show a minor decrease if orientation averaging is performed.

As for $[a_h]$, we learned that the conclusions of CmS at β=0° always hold true when β is sufficiently large (Figure 7b).

We also examined the results across the entire PSD range of raindrops shown in Figure 8. It was found that the aforementioned CmS results for $[z_h]$ were significant when $D_{eq}$ > 3 mm (Figure 8g), while those for $[a_h]$ were significant only for larger raindrops (i.e., $D_{eq}$ > 5 mm, see Figure 8j).

It is worth mentioning that raindrops had significantly higher $[z_h]$ for larger absolute values of the beam elevation angle e (Figure 8a). This can be easily understood since zenith or nadir observation geometries tend to capture a larger cross-section of oblate-spheroid-like models.

As for other SSP factors of polarimetric radar variables, the CmS deviations are either not stable against size spectra (Figure 8h&i) or too small in terms of its base magnitude (Figure 8l), making them not significant for a particle group. However, the CmS effects of SSP factor for differential attenuation ([$a_h$]- [$a_v$]) in Figure 8k is significant at low absolute elevation angles, indicating that the SSP factor of vertical polarization attenuation [$a_v$] is significantly stronger for Chebyshev model (considering [$a_h$] is close for two shapes at low absolute elevation angles in Figure 8j).

The canting angle of raindrops is generally small, so the CmS deviations of SSP factors found in Figure 6 and Figure 8 only show a minor decrease in Level B – ASSP database when compared to the curves of β = 0° in Level A – RSSP database. Hence, we omit further analysis of orientation averaging of raindrops and SSP factors of Level B database here.

**3.4 PSD Options and Analysis of Level-B to Level-C LUT Conversion**

In this subsection, we will focus on describing the PSD options for raindrops and the methods used to analyze their impacts on the conversion of SSP factors of radar variables to intrinsic radar variables (namely, ASSP to BSP conversion).

In ZJU-AERO, a total of six options for PSD schemes for raindrops were available, which are listed in Table 3. All schemes were designed for rain modelled by a single-moment microphysics scheme with exponential distribution assumptions. Each PSD schemes can be visualized as a $N_0$-$\Lambda$ curve, as shown in Figure 9.

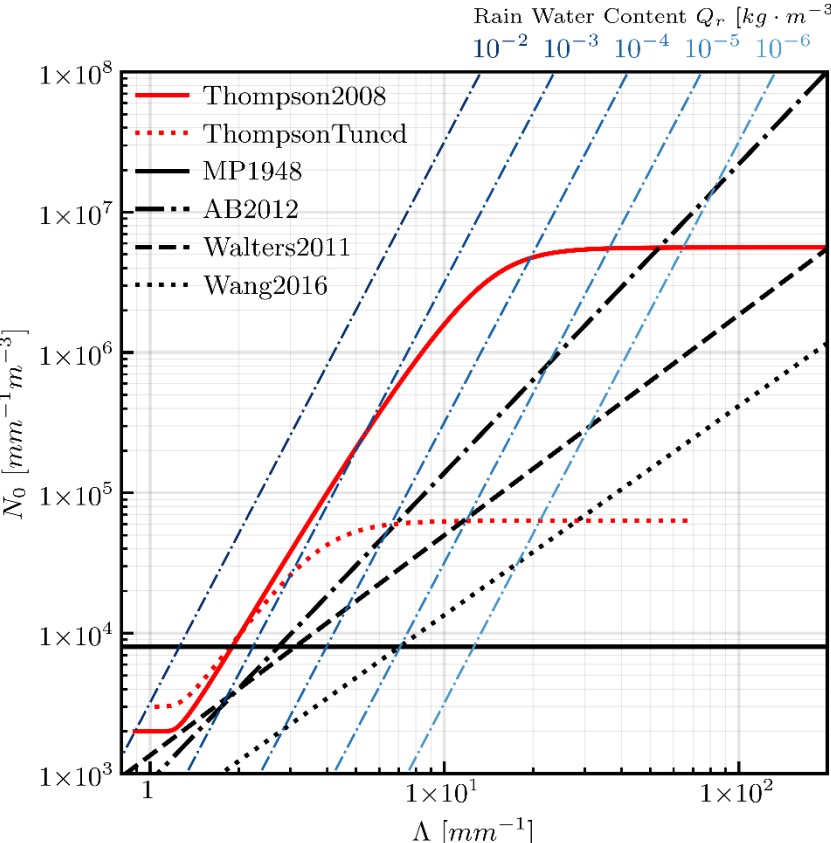

Figure 9: The $N_0$-$\Lambda$ diagram introduced by Abel and Boutle (2012). All the PSD schemes mentioned in Table 1 are represented as black or grey curves in this figure. The rainbow-coloured dash–dotted lines represent isolines of water concentration for rain $Q_R$. The curves extend to the upper right corner as $Q_R$ increases. The solution of ($N_0$, $\Lambda$) for a given PSD scheme and $Q_R$ can be found by determining the intersection of colored curves and black/gray curves in this diagram.

The constant $N_0$ parameterization originally proposed by Marshall and Palmer (1948) and used in microphysics packages, such as Hong and Lim (2006), provides a rough approximation for various liquid precipitation scenarios. However, modern PSD schemes for rain, such as the one by Thompson et al. (2008), aim to capture the observed transition from high concentrations of drizzle-sized drops ($D_{eq}$ < 0.5 mm) in stratocumulus clouds to PSDs dominated by large mm-sized raindrops in heavy precipitation.

We classified the six PSD schemes into two groups by how the intercept parameter $N_0$ is determined. Group A is characterized by a power-law $N_0$-$\Lambda$ relationship, represented as a straight line in the dual-log-scale diagram of Figure 9. On the other hand, the intercept parameter, $N_0$, for group B schemes follows a formalization described by Eq. (24), transitioning from $N_2$ to $N_1$ as the water mass concentration, $Q_R$, increases. This can be visualized as tanh-like curves shown in Figure 9:

$$N_0 = \left(\frac{N_1 - N_2}{2}\right)\tanh\left[\frac{(Q_{R0} - Q_R)}{4Q_{R0}}\right] + \left(\frac{N_1 + N_2}{2}\right), \tag{24}$$

The values of reference rain water mass concentration $Q_{R0}$ and the dynamic range $[N_2, N_1]$ of the intercept parameter are displayed in the group B of Table 3.

**Table 3: The parameters of rain particle size distribution (PSD) schemes available in ZJU-AERO. The group A of PSD schemes: (1) MP1948, (2) AB2012, (3) Walters2011, and (4) Wang2016 can be generalized as a power-law $N_0$-$\Lambda$ relationship (see Eq. (9)) as stated in Section 2.3, while group B of PSD schemes: (1) Thompson2008 and (2) ThompsonTuned can be generalized as Eq. (24), in which $N_0$ is simply determined by the mass concentration of rain. The last scheme tagged as "ThompsonTuned" was proposed in the present study to fit of the observations in the case study presented in Section 4. Note that the 10-based or 1000-based coefficients in the "$x_1$" column of group A are used for unit conversion as the parameters are taken from various studies with divergent unit conventions. Also note that the parameter $Q_{R0}$ is originally a mixing ratio in units of $kg \cdot kg^{-1}$ rather than mass concentration in units of $kg \cdot m^{-3}$, but we have performed the conversion by assuming $\rho_{air}=1.225\ kg \cdot m^{-3}$ of standard atmosphere at sea level (1013.25hPa). It should be noted that the constant air density is only used here to qualitatively determine the position of the Thompson2008 and ThompsonTuned $N_0$-$\Lambda$ curves in Figure 11. Diagnostic air density is used in actual PSD solver of ZJU-AERO.**

**Group A:**

| PSD schemes | Tag | $X_1$ [$mm^{x_2-1} \cdot m^{-3}$] | $X_2$ [-] |
|---|---|---|---|
| Marshall and Palmer (1948) | MP1948 | $8.0 \times 10^3$ | 0.0 |
| Abel and Boutle (2012) | AB2012 | $0.22 \times 1000^{x_2-1}$ | 2.20 |
| Walters et al. (2011) | Walters2011 | $26.2 \times 1000^{x_2-1}$ | 1.57 |
| Wang et al. (2016) | Wang2016 | $14.1 \times 10^{x_2}$ | 1.49 |

**Group B:**

| PSD schemes | Tag | $N_1$ [$mm^{-1} \cdot m^{-3}$] | $N_2$ [$mm^{-1} \cdot m^{-3}$] | $Q_{R0}$ [$kg \cdot m^{-3}$] |
|---|---|---|---|---|
| Thompson et al. (2008) | Thompson2008 | $9.0 \times 10^6$ | $2.0 \times 10^3$ | $1.225 \times 10^{-4}$ |
| Thompson et al. (2008) Tuned[1] | ThompsonTuned | $1.0 \times 910^5$ | $3.0 \times 10^3$ | $3.0 \times 10^{-4}$ |

[1] The ThompsonTuned PSD scheme is for numerical test only, therefore it is not a user option in Table 1.

Figure 9 demonstrates the large variability of intercept parameter $N_0$ for a specified rain water content (RWC) among different PSD schemes. That variability can be shown by measuring the vertical coordinate difference of the intersection points between a given iso-RWC line and different $N_0$-$\Lambda$ lines of PSD schemes. For PSD schemes expressed by exponential distribution, larger

intercept parameter $N_0$ means more smaller drops when RWC is fixed. Therefore, the Thompson scheme (Thompson et al., 2008) priorities smaller drops, while the Wang scheme (Wang et al., 2016) places more emphasis on larger drops.

According to global aircraft *in-situ* measurements carried out by Abel and Boutle (2012), the six single-moment (SM) microphysics assumptions in Table 3 cover the natural variability of raindrop PSDs ranging from continental precipitations to maritime precipitations. Typical continental PSD with intensive large drops such as Wang et al. (2016) and typical maritime

PSD such as Thompson et al. (2008) with a large population of drizzle-sized raindrops are both present. That is why we have chosen the six SM microphysics schemes for the PSD sensitivity test.

It should be noted that although SM microphysics has been used for years and will continue to be used, it has obvious shortcomings in reproducing polarimetric features observed in convective and stratiform events. Since double-moment (DM) microphysics in CMA-MESO is still experimental, the consistent DM scheme in ZJU-AERO is still in development stage.

However, the nature variability of raindrop PSDs for DM microphysics schemes is still within what is displayed in Figure 9, therefore it is safe to use those SM assumptions for a sensitivity test.

To explore the effects of different PSD schemes on the radar variables, a new intermediate quantity called "backscatagrand" is analyzed in Figure 10, which is a spectra of particle size defined as the product of horizontal backscattering cross section $\sigma_{bsca,h}(D)$ and size distribution function $N(D)$. It helps us identify particles of which size-range dominate the energy of

backscattering.

The idea of analysing "backscatagrand" was inspired by the definition of "extagrand" used in the analysis of RTTOV-SCATT bulk-scattering tables (Geer et al., 2021), which helps diagnosing the single-particle contribution to extinction coefficients in radiative transfer simulations of passive microwave sounders. The concept of "extagrand" is based on the insight that the total extinction of an emsemble of particles is primarily influenced by a fraction of particles within a narrow diameter range.

Therefore, by analyzing the SSP for particles within that size range, we can infer the BSP of the entire ensemble of particles. The "extagrand" $\sigma_{ext}(D)N(D)$ in units of $mm \cdot m^{-3}$ is factorized as the product of mass distributions $m(D)N(D)$ in units of $kg \cdot mm^{-1} \cdot m^{-3}$ and the extinction cross section per unit volume $\sigma_{ext}(D)/m(D)$ in units of $mm^2 \cdot kg^{-1}$. These quantities are determined by PSDs and SSPs, respectively. The "extagrand" quantities are appreciable only for size ranges in which both mass distribution $m(D)N(D)$ and mass extinction efficiency $\sigma_{ext}(D)/m(D)$ are large enough to produce significant products "extagrand".

However, for weather radar applications, the principal quantity is backscattering rather than the extinction, as is the case for passive instruments. Hence, we try to analysis quantities of the horizontal polarization backscattering cross section per volume $\sigma_{bsca,h}(D)/m(D)$ (parallel definition of mass extinction efficiency, which we refer to as mass backscattering efficiency hereafter) and the "backscatagrand" $\sigma_{bsca,h}(D)N(D)$ (parallel definition of "extagrand"), in which $\sigma_{bsca,h}(D)$ indicates the horizontal polarization backscattering cross section.

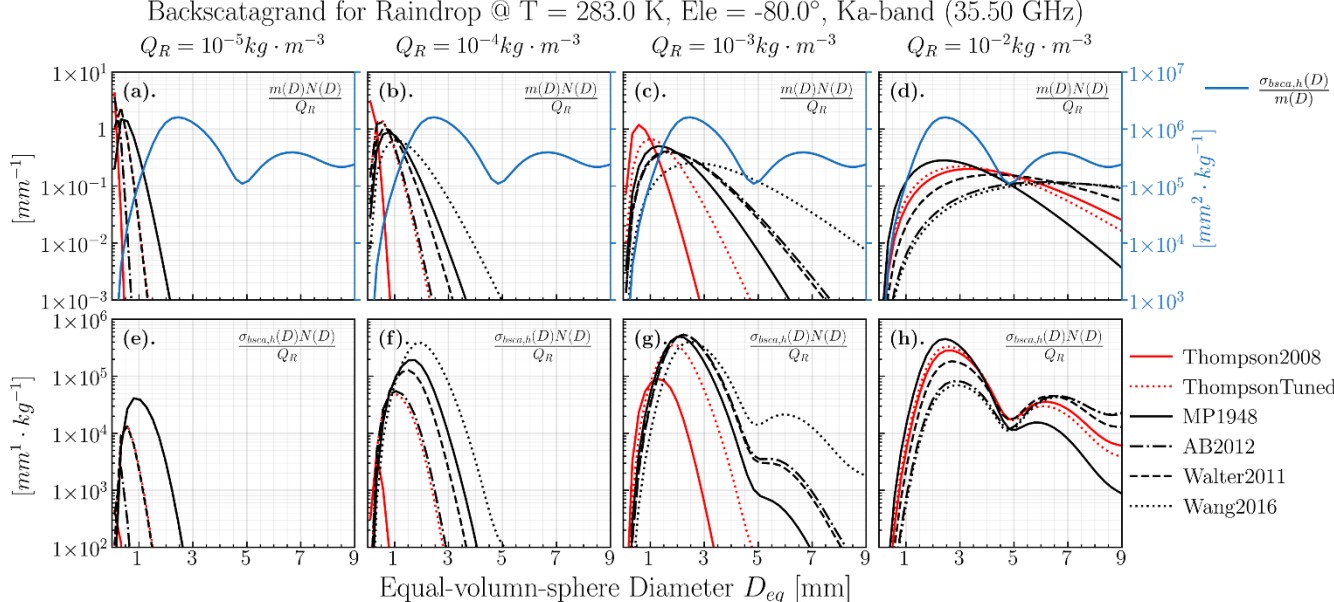

**Figure 10: The analysis of the integration of single particle horizontal polarization backscattering over the PSD.** The horizontal polarization backscattering cross section $\sigma_{bsca,h}$ was obtained using the applied single scattering properties (ASSP) – Level B of the Chebyshev raindrop database. The regular conditions of space-borne radar (T = 283 K, e = –80°) at the Ka band (35.5 GHz) (indicated in the title) were considered. The first row displays the mass distribution m(D)N(D) (normalized by rain mass concentration $Q_R$) for five PSD schemes and four water concentration settings, while the mass backscattering efficiency $\sigma_{bsca,hv}$(D)/m(D) is indicated by the blue lines with the blue axis and labels marked on the right. The second row shows the quantity "backscatagrand" $\sigma_{bsca,hv}$(D)N(D) (also normalized by rain mass concentration $Q_R$) for different PSD schemes and water concentrations, which is a measure of contributions particles make to the horizontal reflectivity $z_h$ (the principal intrinsic radar variable). The red or black curves in the first row multiplied by the blue curve exactly yield the second row's curves.

In Figure 10, we can find that the mass distributions and backscattering contribution spectrum (i.e., the so-called "backscatagrand") exhibit large uncertainties due to natural variability of PSD schemes. The mass backscattering efficiency has multiple oscillations caused by the resonance effect, particularly in high-frequency bands such as the Ka-band. Specifically, two important peaks of mass backscattering efficiency at $D_{eq}$ = 2.5 mm and $D_{eq}$ = 6.5 mm and one important dip at $D_{eq}$=5.0 mm exist (solid blue line in Figure 10(d)). For extremely heavy precipitations ($Q_R \sim 10^{-2}$ kg·m$^{-3}$), the peak of mass distributions might coincide with the dip of mass backscattering efficiency at $D_{eq}$=5.0 mm. This phenomenon is unique to PSD schemes that contains more large drops (such as Wang2016 and AB2012), which leads to a loss of bulk reflectivity if the total mass of raindrop remains constant.

Under all but extremely heavy precipitation conditions ($Q_R<10^{-2}$ kg·m$^{-3}$, see Figure 10c&g), the Thompson2008 PSD scheme stands out as an outlier. Its extremely large $N_0$ compared to other schemes (see Figure 9) leads to an significantly smaller peak in mass distributions as small as $D_{eq}$ < 1.0 mm, while the peak of other PSD schemes is approaching the first peak of mass backscattering efficiency at $D_{eq}$ = 2.5 mm. Accordingly, the total reflectivity computed with the Thompson2008 scheme must be much smaller at $Q_R$=10$^{-3}$ kg·m$^{-3}$ than those computed with other PSDs.






The relative importance of particles in the entire spectrum can be diagnosed with the curves of backscatagrand, which is the ultimate goal to introduce this concept. For example, we can learn from Figure 10h that the contribution of backscattering is dominated by particles with diameter $D_{eq}$ at around 2.5 mm for the MP1948 scheme, while the contribution from particles of 2.5 mm-size and 6.5 mm-size are almost equally important for the Wang2016 and AB2012 schemes. To go further, we can infer that the CmS deviations mentioned in Section 3.3 can only affect the bulk-scattering properties computed with the Wang2016 and AB2012 schemes, since the CmS deviations of backscattering are only significant for particles with $D_{eq}>3$ mm (Figure 8g).

## 3.5 Bulk Scattering Properties: Level C

Figure 11 shows the intrinsic radar variables for the Chebyshev model, which were computed using the BSP database and the spheroid model at the Ka band. In Section 3.4, we hypothesized based on the backscatagrand plot, and we can now confirm these hypotheses individually:

1. The application of PSD schemes, such as Wang2016, can result in a significant reduction in the reflectivity $Z_H$ (by approximately 5 dBZ) for extremely heavy precipitation scenarios compared to the default PSD scheme MP1948 (refer to Figure 11a, $Q_R\sim10^{-2}$ kg·m$^{-3}$).

2. The reflectivity $Z_H$ computed by the Thompson2008 scheme for moderately heavy precipitation scenarios is considerably lower (by over 10 dBZ) compared to other PSD schemes in group A of Table 1 (refer to Figure 11a, $Q_R\sim10^{-3}$ kg·m$^{-3}$).

3. The CmS deviations are only significant (reducing $Z_H$ by about 2 dBZ) for extremely heavy precipitation scenarios and PSD schemes that emphasize larger drops, such as Wang2016 (refer to Figure 11a, $Q_R\sim10^{-2}$ kg·m$^{-3}$).

4. The CmS deviations of attenuations, $\alpha_H$, are never significant for regular mass concentrations of rain $Q_R$ (see Figure 11b), This finding can be explained by the fact that the CmS deviations in RSSP (Figure 8j) are significant only if $D_{eq}>$ 7 mm, which represents a group of drops sharing a small fraction of the mass distribution, even for extremely heavy precipitations.

From Figure 11, it can be concluded that the horizontal polarization intrinsic radar variables $Z_H$ and $\alpha_H$ at the Ka band (actually also for the Ku band, not shown here) are more sensitive to the uncertainty of PSD schemes than the CmS optical properties deviations for space-borne radar observation geometries. The CmS deviations are only notable for extremely heavy precipitation scenarios in which large drops are prominent. Those conclusions focusing on Ka-band (35.6 GHz) generally confirm well with what were found by Ekelund et al. (2020).

The sensitivities of polarimetric intrinsic radar variables with respect to CmS deviations for ground-based radar bands (S/C/X-band) and viewing geometries were found to be negligible. Therefore, they are not shown here.

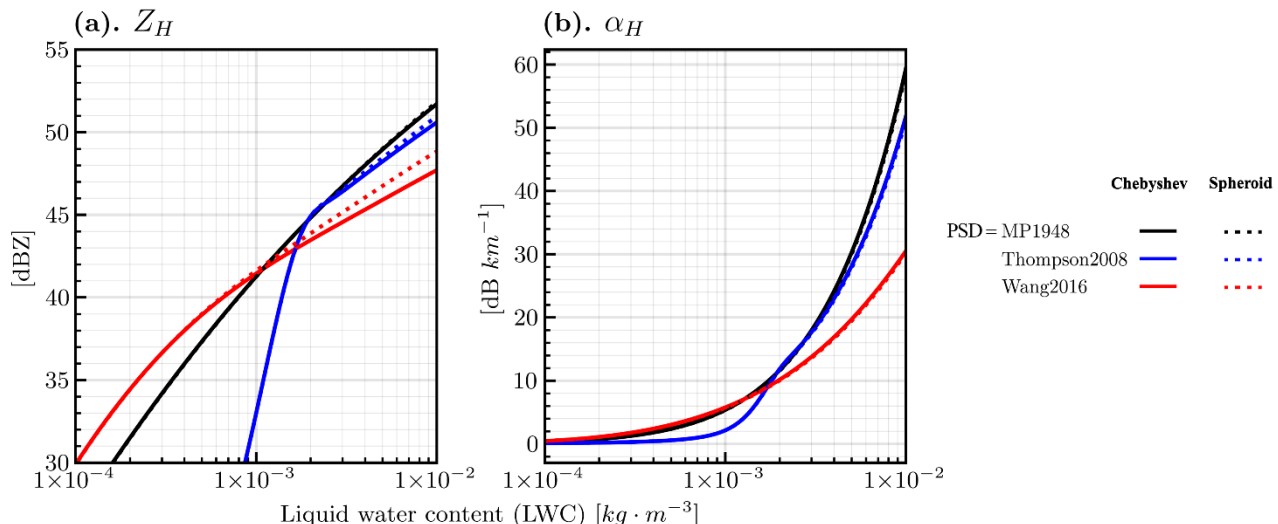

**Figure 11: The intrinsic radar variables against the liquid water content (mass concentration of rain $Q_R$) computed with the bulk scattering properties (BSP) Level C database of the Chebyshev raindrop and spheroid models, which were indicated by solid and dotted curves, respectively. A typical condition of the space-borne radar (T = 283 K, e = −80°) at the Ka band (35.5 GHz) was considered. The panels (a) and (b) display the two intrinsic radar variables (horizontal reflectivity $Z_H$ and horizontal attenuation $\alpha_H$) that can be diagnosed from the corresponding factors of SSP in Figure 8(a) and (d), respectively. The colors of the curves are used to indicate three typical PSD schemes chosen from Table 1. Here, only the results of PSD schemes MP1948, Thompson2008 and Wang2016 are displayed, because MP1948 is a benchmark traditional PSD scheme, while Thompson2008 and Wang2016 are representative of typical maritime and continental precipitation PSD, respectively.**

## 4 Case Studies

To assess the simulation capabilities of ZJU-AERO, we conducted case studies using the real-world data and investigated the sensitivities of the PSD schemes and the new Chebyshev raindrop model. For this study, we chose the GPM core satellite (referred to as GPM hereafter) and specifically analyzed the overpass of Typhoon Haishen, the first super typhoon of the 2020 Northwest Pacific typhoon season, on September 5, 2020, at 09:21UTC. We used the simulation and observation data from the dual-frequency precipitation radar (DPR) onboard the GPM for our analysis.

To conduct the ZJU-AERO simulations, we utilized model grid data generated by the operational run of CMA-MESO, which was initialized at 00 UTC of September 5, 2020. The CMA-MESO grid data had a horizontal resolution of 3 km and consisted of 50 vertical layers. Note that the WSM6 microphysics package was selected for the CMA-MESO operational run. However, any forced PSD schemes could be applied in the simulation of the radar operator, as mentioned in Section 2.3.

To demonstrate the reliability of the forward radar operator for ground-based polarimetric radars, we have also performed a case study of a meso-scale convective system (MCS), which can be found in the user manual (see Section Code Availability). The results are reasonable but relatively trivial compared to previous radar operators, so we will not display them in this section.

## 4.1 Simulation of a Tropical Cyclone Case

According to Iguchi (2020), the GPM-DPR's Ku-band radar basically follows the instrument characteristics of the Tropical Rainfall Measurement Mission (TRMM) Precipitation Radar (PR), while the new Ka-band radar is sensitive to light rain and snow. By combining data from two channels, more accurate estimates of DSD parameters can be obtained (Rose and Chandrasekar, 2006; Liao et al., 2014). The Ka-band radar has two modes (Iguchi et al., 2010): (1) a high-sensitivity mode for light-rain and snow (high-sensitivity beam) and (2) a matched-beam mode in which the sampling volumes of Ka- and Ku-band radar are collocated (matched beam). However, since May 21, 2018, the GPM-DPR has switched its scan pattern, so that now a full swath can be considered as the matched beam in the evaluation of forward radar operator. Therefore, in this case study, we used data from the Ka-band in the matched-beam mode to estimate DSD and drop morphology parameters.

The dual frequency ratio (DFR) is defined as the difference between the log-space measured reflectivity at two channels (Ku- and Ka-bands). Previous studies have shown that the DFR can be used to distinguish between stratiform and convective rain (Le and Chandrasekar, 2012).

Figure 12 displays the observation and simulation of the Ku-band radar reflectivity at different levels, with the last row showing the mismatch between them (i.e., the OmB of reflectivity). Figure 13 presents the cross-section of radar reflectivity between points A and B in Figure 12, separately for the Ku- and Ka-bands. Additionally, the last column shows the DFR as defined above. The radar operator applied the ThompsonTuned PSD option (for developers only) and the Chebyshev morphology options A2 (see Table 3) for the category rain in the simulations. For snow, we use default option A2. For graupel, we use default option A2 and B1. The reflectivities in Figure 12a–d are masked by a flag called "flagPrecip" (available at the ground) offered in the L2A swath data of GPM-DPR, while the simulation of reflectivity in Figure 12e–h is masked by the sensitivity threshold of 12 dBZ to keep it the same with sensitivity of GPM/DPR observation. We applied the attenuation in simulations, while using the "zFactorMeasured" product of GPM/DPR (no attenuation correction applied). As for the calculation of OmB reflectivity, the radar gates for which the reflectivity is undetectable (below the instrument sensitivity) either in simulation or observation are filled with a "background" reflectivity of 0 dBZ to generate the OmB reflectivity.

Based on the comparison between observations and simulations in Figure 12 and Figure 13, it can be found that the regional model of CMA-MESO is able to capture some of storm structures, such as the cyclone eye and eye-wall. However, the structures of outer spiral rain bands in the simulation (Figure 12e and Figure 13d) appear more contiguous and vague, rather than the isolated towering bands depicted by the GPM-DPR measurements (Figure 12a and Figure 13a). Although the NWP model could not accurately predict the cloud and precipitation timing and position, the probability distribution of simulated radar reflectivity should be unbiased when compared with observations (detailed examinations on that will be conducted in Section 4.2); otherwise, a systematic bias in the NWP model or the radar operator might be identified. Since (a) the precipitation forecast of CMA-MESO model can be frequently calibrated against rain gauge network (Bárdossy et al., 2008; Cattoën et al., 2020) and (b) a forecast lead time of 9 hours is beyond the 6-hour spin-up time of rain water content in CMA-NWP (Ma et al., 2021), we assume that the NWP model CMA-MESO has no significant bias in this case.

Notably, the freezing level in this case was found within the altitudes ranging from 4 to 5 km (see the 0 °C NWP model background isothermal line in Figure 13), which is believed to contain abundant melting particles. Actually, the GPM-DPR

observations revealed a weak bright band (BB) signature below the freezing level, which can be recognized at Ku-band (with an average reflectivity enhancement of approximately 3 dB, see Figure 13a) but not so clear at Ka-band (see Figure 13b). As reported by long-term radar observation statistics of melting layers documented in Fabry and Zawadzki (1995), the magnitude of BB signature in melting layer is significantly weaker in deep-convection regions. Therefore, the weak BB signatures in melting layer of this tropical cyclone case can probably be attributed to the large riming rates (Zawadzki et al.,

2005) in convective precipitations of tropical cyclone eyewall and rain bands. However, the simulations we conducted did not show a BB signature, which was attributed to not considering melting or mixed-phased particles, and the shortcoming of the microphysics scheme. Considering the lack of melting schemes in ZJU-AERO, it is expected that the melting layer would exhibit a positive OmB signature due to lack of melting hydrometeors in ZJU-AERO. However, in Figure 13g and h, we encountered difficulties in identifying continuous positive mean bias of the OmB for both the Ku- and Ka-bands in the

melting layer. This challenge arose due to the large mislocation errors of precipitation predicted by the NWP model used in this study. To address this issue, future analysis could employ horizontal averaging or examine the probability distribution function of reflectivities in the melting layer. Implementation of melting particle models and their relevant validations will be subject to upcoming publications.

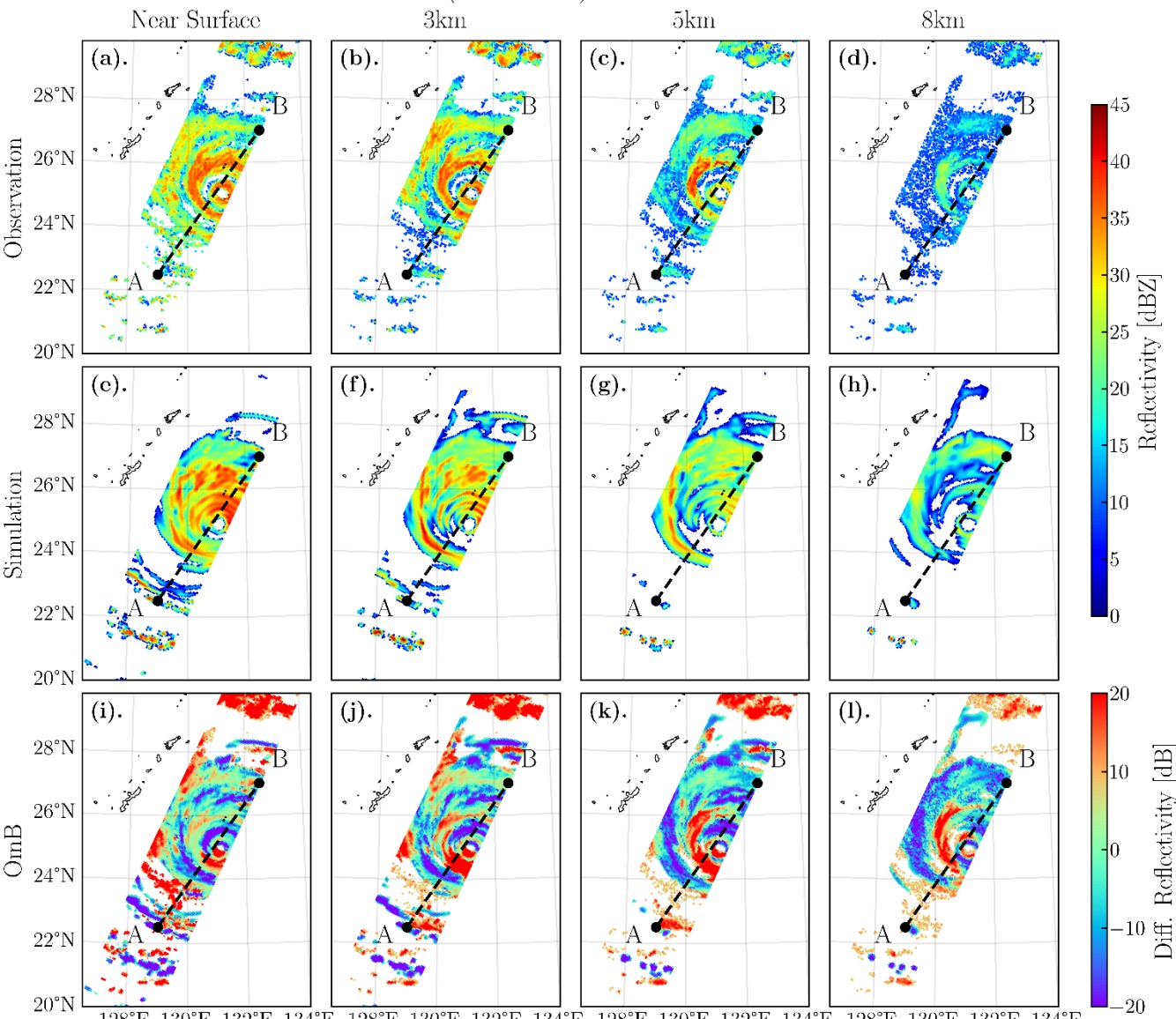

**Figure 12: Panels (a)-(d) display the Global Precipitation Mission-Dual Frequency Precipitation Radar (GPM-DPR) observed radar reflectivity at the Ku band (13.6 GHz) for the overpass of Typhoon Haishen at 09:21 UTC on September 5, 2020. The results shown at different columns corresponds to four altitude levels (namely, NearSurface (the first clutter-free gate), 3, 5, and 8 km). The second row (panels e–h) shows the simulated radar reflectivity by applying the +9H forecast output of the CMA-MESO (initiated at 00 UTC on September 5, 2020) to the ZJU-AERO. The last row (panels i–l) shows the observation minus simulation (OmB) of radar**
**reflectivity of the four levels. The cross section indicated by the dashed line between A (129°E 22.5°N) and B (132.5°E 27°N) was selected for further studies as shown in Figure 13.**

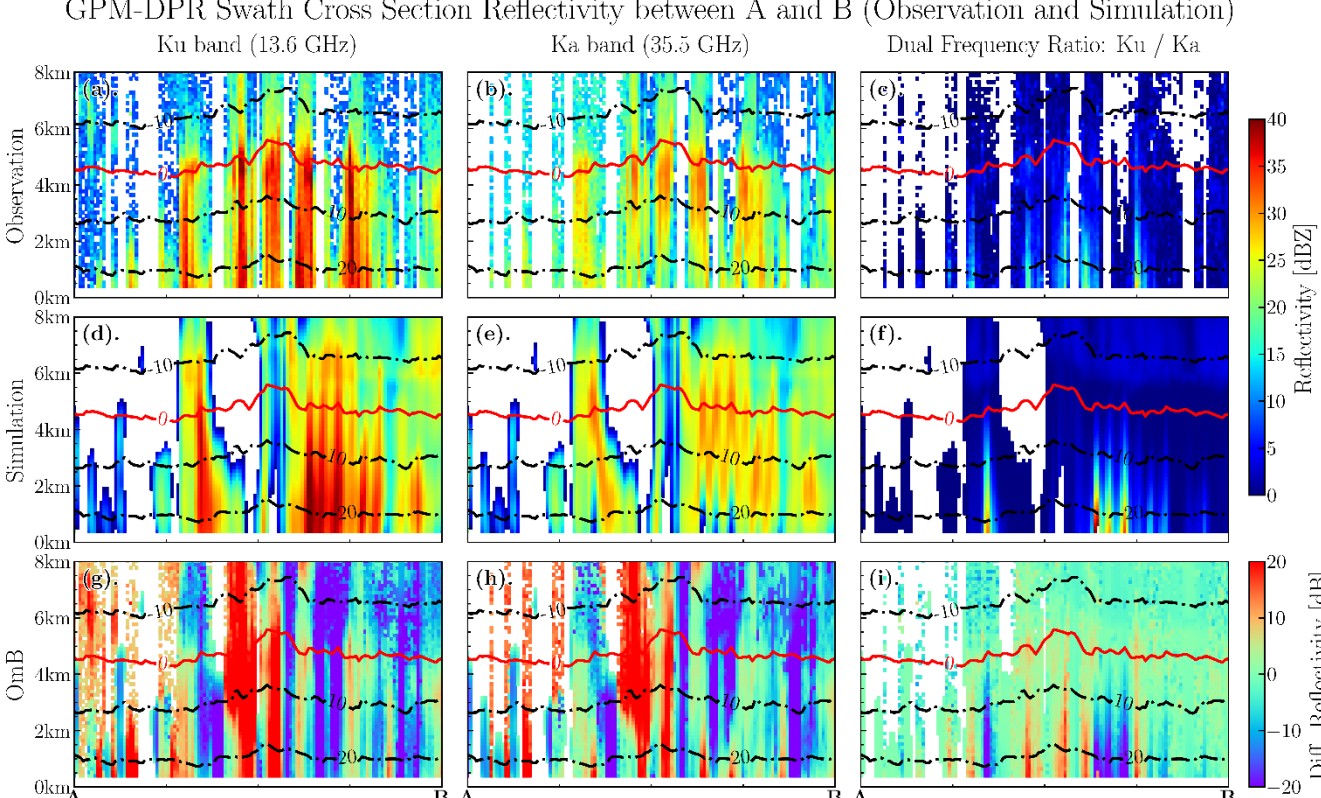

**Figure 13: The cross-section of radar reflectivity, indicated by the line AB in Figure 12. The GPM-DPR observations are shown in the first row (panels (a)–(c)), simulations of the radar operator are shown in the second row (panels (d)–(f)), while the OmB of reflectivity is shown in the third row (panels (g)–(i)). As for the arrangement of panels in column-view, the first, second, and third columns display the results of the Ka and Ku bands and the dual-frequency ratio (DFR, Ku/Ka), respectively. The temperature of model background state is indicated by isothermal lines in each panel. Numbers of the contour labels have the unit of Celsius degree.**

Moreover, extremely large DFR values (>30 dB) were found in simulations (Figure 13f) but not in observations (Figure 13c) in the 0 to 2 km layer, which could be attributed to an over-estimation of attenuation in the Ka-band simulation for heavy precipitations (Figure 13e). This hypothesis is supported by two facts: (1) many weak reflectivity (~10 dBZ) regions are found right beneath the strong reflectivity gates aloft at around 4 km for the Ka band (Figure 13e) and (2) the weak reflectivity regions of the Ka band collocate with the strong reflectivity region (~40 dBZ) of the Ku band (Figure 13d).

The OmB plots, as shown in Figure 12 and Figure 13, are useful tools for verifying and calibrating new observation operators and identifying their deficiencies. Overall, the results are generally reasonable and demonstrates the capability of ZJU-AERO in simulating the reflectivity of dual-frequency spaceborne radar such as GPM-DPR. More analysis on bias of probability distribution of reflectivities / DFR will be presented in the next subsection.

## 4.2 Sensitivity Assessments on Hydrometeor PSDs and Morphologies

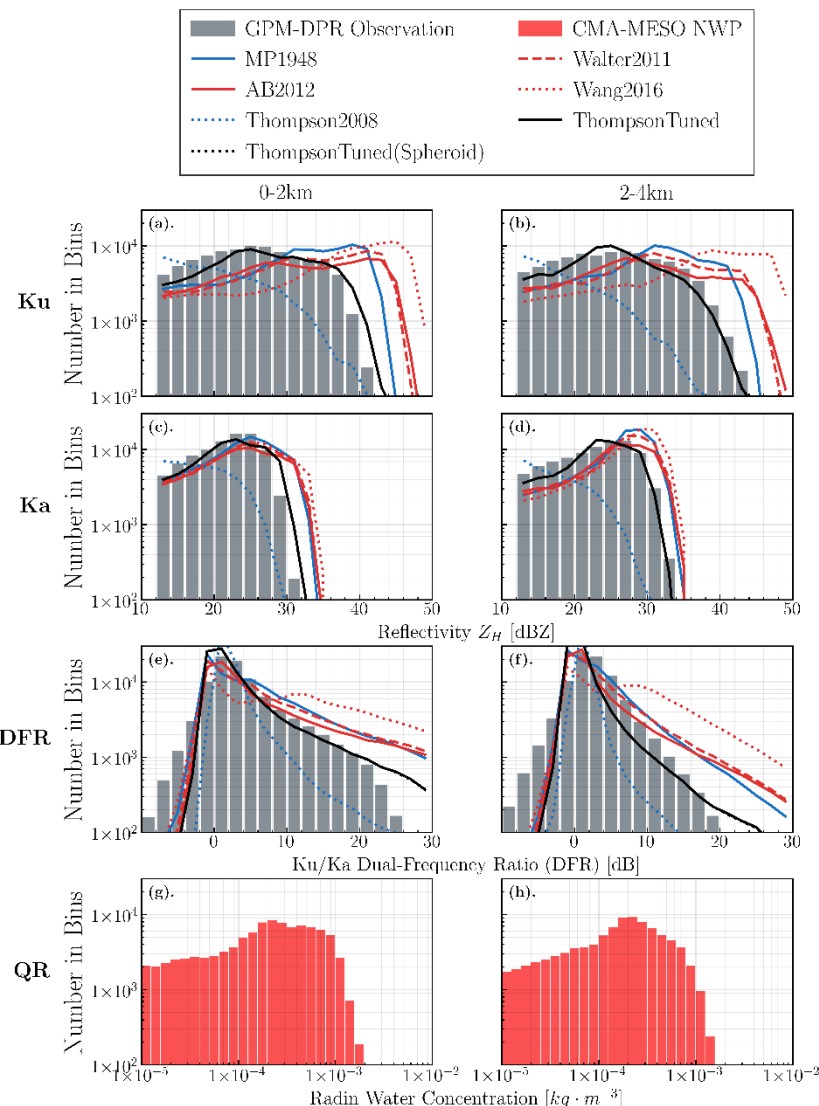

**Figure 14: The observation and simulation distributions of radar reflectivity at the Ka band (the first row), Ku band (the second row) bands, and the dual frequency ratio (DFR) (the third row). The last row shows the log-scale histogram of rain water content ($Q_R$) predicted by CMA-MESO. The data of distributions were gathered from the reflectivity of all the radar gates in the four altitude layers (namely, the 0–2 km layer, 2–4 km layer as the tags on the top of the columns suggest). The first two layers (0–2 and 2–4 km) were primarily composed of liquid hydrometeors, while the layers above 4km contain melting and solid particles (results not shown here). The observation distributions of reflectivity are shown by grey bars in the panels, while the simulation distributions are indicated by curves with different colors and styles. The data of observations and simulations are binned equivalently between 12dBZ to 50dBZ with a bin-size of 2dB.**

We also performed sensitivity assessments for the PSD and non-spherical morphology options of the rain hydrometeor category. Figure 14 shows the observation and simulation distributions of radar reflectivity at the Ku- and Ka-bands and the dual frequency ratio (DFR) at two altitude layers (0-2 km, 2-4 km). Since we were tuning the liquid hydrometeor, the melting

and solid layer in the last two columns (4-6 km and 6-8 km) did not concern us. The radar operator applied the Chebyshev morphology in all the simulations except the group with the label of "ThompsonTuned(Spheroid)", which was simulated with the spheroid raindrop option. Other options of ZJU-AERO are identical with those of Figure 12.

Based on statistical analysis (see Figure 14g and h), we found that the mass concentration of rain in the CMA-MESO at the storm rain bands was primarily dominated by moderately heavy rain ($Q_R \sim 10^{-3}$ kg·m$^{-3}$) rather than extremely heavy rain ($Q_R \sim 10^{-2}$ kg·m$^{-3}$).

According to Section 3.5/Figure 11, applying the Thompson2008 PSD leads to a significant reduction in the simulated reflectivity in conditions of moderately heavy rain ($Q_R \sim 10^{-3}$ kg·m$^{-3}$), which is consistent with our findings in Figure 14(a), (b), (c), and (d) for the Ku- and Ka-bands of liquid layers.

Due to the strong wind shear in the rain bands of tropical cyclone, large drops can be broken apart (Radhakrishna, 2022), causing the rain PSD in such conditions to behave irregularly and deviate from the prevailing parameterizations. Figure 14(a),

(b), (c), and (d) demonstrate that almost all the PSD schemes significantly overestimated the reflectivity for both Ku- and Ka-bands, except for the Thompson2008 scheme, which underestimated the reflectivity.

To address the discrepancy, we designed a new PSD scheme, referred to as the ThompsonTuned, by tuning the parameters in Eq. (24). The optimization procedure can be formulated as follows.

We use a scoring method to quantify the match between observation and simulation histograms, as proposed by Geer and

795 Baordo (2014):

$$s = \sum_{j=1}^{nbands} \sum_{i=1}^{nbins} \left| \log\left( \frac{N_{sim}^{j}(i)}{N_{obs}^{j}(i)} \right) \right|, \tag{25}$$

Here, s represents the final score, with $N_{sim}$ and $N_{obs}$ indicating the counts in the i-th bins of the simulation and observation histograms, respectively. The superscript 'j' denotes j-th band. To prevent infinite values in summation, we assume a count of 0.1 in bins with empty values in either the simulation or observation histograms.

Next, we establish a grid of free parameters for optimization, which includes three parameters (N1, N2, and QR0). Due to the broad range of these free parameters, the grid is set up in a quasi-logarithmic scale.

Subsequently, simulations are conducted and the score from Eq. (25) is evaluated for each grid point. By identifying an optimal region in the parameter space with the best score, we refine the grid in that region to pinpoint a more precise parameter subregion. This iterative process is carried out to fine-tune the parameters.

Finally, we identified an optimal point at which the black lines in Figure 14(a), (b), (c), and (d) closely matched the observed distributions. The parameters are listed in Table 3 and plotted in the $N_0$-$\Lambda$ diagram of Figure 9. The tuned PSD scheme is reasonable as it falls between the MP1948 and Thompson2008 schemes in the $N_0$-$\Lambda$ diagram, with an emphasis on smaller particles compared to MP1948. This may be attributed to the unusual DSD in tropical cyclones.

While no *in-situ* DSD observations are available to support the tuned PSD schemes used in this study, the implications of the

810 tuning experiments are interesting, considering that the matched-beam observation of Ku and Ka bands were designed for the DSD estimation.

As suggested by Section 3, the CmS effects were negligible in this case, as it is difficult to discriminate the solid and dashed black lines in Figure 14. This is likely due to the moderately heavy rainfall in tropical cyclones with strong wind shears, which prevent the large drops from highlighting the CmS deviations. However, the CmS effects could be significant in cases of extremely heavy rain in the supercell storms according to our observation system simulation experiment (OSSE). An OSSE of GPM-DPR overpassing a meso-scale convective system (recorded with extreme heavy precipitation events) reported CmS decrease effects of more than 1 dB at Ku-band and more than 2 dB at Ka-band (figures now shown). This sensitivity test is consistent with what we found in Figure 11(a) and demonstrates the value of introducing Chebyshev-shape raindrop model in certain scenarios (e.g., vertical pointing cloud radar, airborne radar and spaceborne radar working at high-frequency bands). As for polarimetric radar variables such as $Z_{DR}$ and $K_{DP}$ for ground-based radar at side-viewing geometry, the CmS effects are generally negligible.

## 5 Summary and Ongoing Tasks

In summary, Section 2 of our study introduced the basic formulations and concepts of design in the ZJU-AERO. These concepts included the general procedure of the software, radar variable calculations, and the available hydrometeor settings (shape parameters, dielectric constant models, canting angles, particle size distributions). Formulations of polarimetric radar variables are derived starting from single particle back-scattering Mueller matrix Z and extinction matrix K.

In Section 3, we highlighted the unique features of ZJU-AERO, specifically its multi-layered design for the optical database of non-spherical particles. We demonstrated this by displaying the scattering properties using the example of the Chebyshev-shape raindrop particle model, comparing it to the properties of traditional spheroid raindrops. We conducted LUT demonstrations for two database layers: Level-A (raw single scattering properties database), and Level-C (bulk scattering property database). We also introduced a new intermediate quantity named as "backscatagrand" to diagnose the PSD integrations of optical properties. We concluded that the Chebyshev-shape raindrop model shows noticeable differences of bulk scattering properties (compared to spheroid model) only at zenith and nadir viewing geometries and for milimeter-wavelength radar bands. This difference is more prominent for continental DSD models (e.g., Wang2016) with larger drops. These deviations can reach up to 2-3 dB at Ka-band for spaceborne radars in heavy continental precipitation regions where large drops dominate. Given the lower uncertainties in simulating reflectivity of liquid phase compared to solid and melting phases, such a difference deserves attention in specific applications such as comparing data from ground-based and spaceborne radar observations (Warren et al., 2018).

Furthermore, In Section 4, we validated the simulation reliability and capability of ZJU-AERO by analyzing a case study of a tropical cyclone using input from the CMA-MESO for simulating spaceborne radar observations. We found that ZJU-AERO provides reasonable simulation results, except for the bright band signature at melting layer, which can be attributed to the current version of ZJU-AERO, not considering melting or mixed-phased particles. We also performed sensitivity assessments of PSDs and morphology options for rain in ZJU-AERO and found that the ThompsonTuned single-moment PSD scheme

provides the best fit of the reflectivity histogram in the simulation to the GPM-DPR observation. However, using either the Chebyshev-shaped raindrop particle model or the spheroid model makes little difference to the simulation results since the tropical cyclone precipitation has a maritime DSD, where small drops dominate.

Currently, ZJU-AERO is an efficient forward radar operator that has the advantages of handling complexities of non-spheroid particle models. Therefore, it is a powerful research tool for studying the sensitivities of polarimetric radar observations with respect to the non-sphericity of hydrometeor particles. It also applies parallel acceleration techniques to boost its performance, allowing operational applications of data assimilation in NWP models employing single-moment (SM) microphysics (such as CMA-GFS / MESO). ZJU-AERO forward radar operator can be applied in data assimilation (DA) studies using indirect DA methods such as Bayesian approach (Caumont et al., 2010) and direct DA methods such as Ensemble Kalman Filter (EnKF), which both requires no tangent-linear (TL) or adjoint (AD) versions of the forward radar operator.

ZJU-AERO also has some limitations. For example, it currently cannot be applied in DA research based on the variational method. Simplification and linearization works are involved to obtain a TL / AD version of the forward operator. Moreover, PSD solvers for DM microphysics schemes have already been implemented in ZJU-AERO for the experimental CMA-MESO DM microphysics, but there are many validation and evaluation works to be done. Also, unlike the EMVORADO, which use a distributed-memory parallel design and interface with COSMO-NWP model online, ZJU-AERO applies a shared-memory parallel design and the NWP model input should be stored in external files.

Overall, the results were satisfactory, and the ZJU-AERO operator is ready for experimental and operational usage. However, several aspects of ZJU-AERO still need improvement, and we have listed the ongoing development tasks as follows:

1. Improve frozen hydrometeors modelling by introducing irregular-shaped aggregates and riming snow.
2. Improve the modelling of melting particles (including melting snow, melting graupel and melting hail) by using layered inhomogeneous modelling rather than an effective medium approximation for the mixture matrix.
3. Compute the optical properties of melting particles using the IITM code and extend the SSP/BSP database with a new dimension of water fraction.
4. Develop more concrete models for single-crystal, aggregated, and rimmed snow to replace the "soft spheroid" model and create a corresponding SSP/BSP database.
5. Model hail as non-spheroid and inhomogeneous particles.
6. Include cloud ice, which plays an significant role in the high-frequency bands of spaceborne radar.
7. Do more tests for double-moment microphysics schemes in CMA-MESO.
8. Conduct more case studies, particularly with measurements from the spacebore radar FY3-RM/PMR. Notably, the L1 product of spaceborne radar FY3-RM/PMR has been accessible online (released by National Satellite Meteorological Center (NSMC) on November 22, 2023) after the manuscript of this paper is finished. We have already implemented an external I/O module to interface with the data format of FY3-RM/PMR in ZJU-AERO, but the time coverage of observation data is too limited for us to find a good demonstration case. Therefore, more case studies and fine-tuning should be conduceted with future measurements from FY3RM-PMR.

In conclusion, ZJU-AERO is an observation operator that facilitates the exploitation of measurement data from both space-borne and ground-based radars. Its versatility and effectiveness make it a valuable tool for data assimilation in CMA-GFS/MESO. Moreover, ZJU-AERO has the potential to be applied in various other studies within a wide range of contexts.

**Appendix A: Specifications on Spaceborne Radar Trajectory Solver**

For space-borne radars onboard rain measurement satellite platforms, such as the Tropical Rainfall Measuring Mission/Global Precipitation Measurement/Fengyun3-Rain Measurement (TRMM/GPM/FY3RM), the trajectory of the radar beam can be treated without considering the beam-bending effects while still maintaining precision. The WGS84 coordinates of satellites, denoted as **A (scLat, scLon, scAlt),** in addition to the centre of foot-print, **A₁ (Lat, Lon),** can be obtained through the satellite radar L1/L2 products thereafter referred to as swath files. These coordinates can then be used to calculate the local elevation angle of a given radar gate, **C,** using the knowledge of trigonometry in Figure A1(b). The length of segments **H** and $R_E$ can be computed by converting the (latitude, longitude, altitude) WGS84 coordinates to the Earth-Centre–Earth-Fixed (ECEF) coordinates. The range of radar gate **AC** is provided by the space-borne radar observation system in the L1 product of GPM/DPR known as "scRangeEllipsoid" (Iguchi et al., 2010). When neglecting the beam-bending phenomenon in the space-borne radar detection, the local elevation angle, e', can be expressed as e' = e-α in which e is the elevation angle on the satellite. The angle α could be determined using trigonometry, given that AC represents the range of the radar gate C.

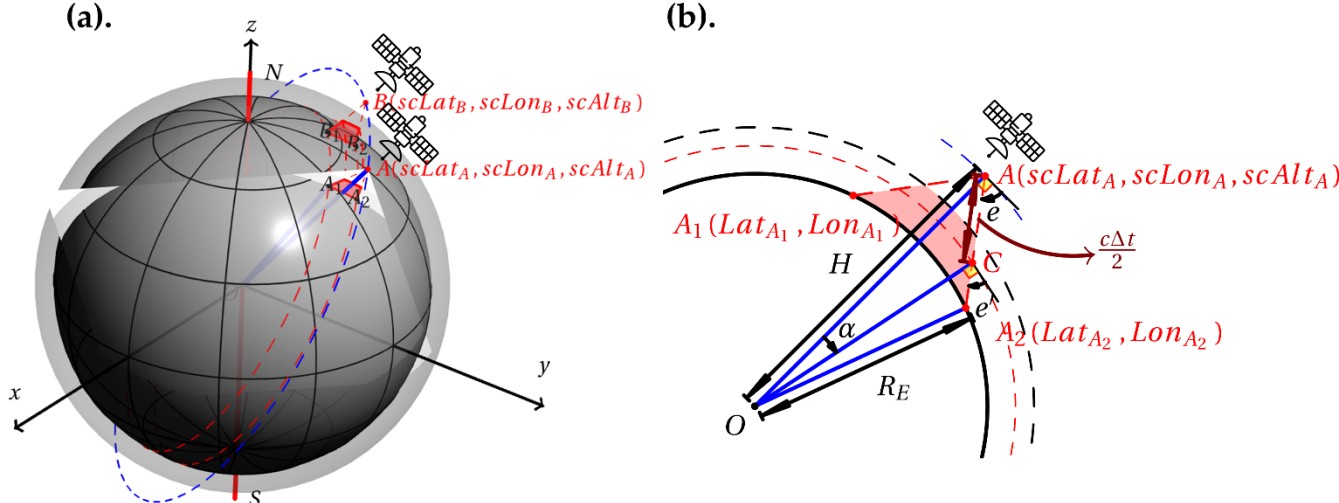

**Figure A1: Conceptual graphs depicting the observation geometry of space-borne radar, with panel (a) showing a 3D graph illustrating the inclined orbit of a precipitation measuring satellite by a blue dashed line. The satellite positions A and B were selected from the orbit and the triples (scLat, scLon, scAlt) on A and B were measured using the WGS84 coordinate system. The space-borne radar is capable of performing cross-track scans, creating a swath between two parallel red cycles on the Earth. The space spanned by two red isosceles trapezoids indicates that the valid scan volume in troposphere between orbit positions A and B in the troposphere. In panel (a), we selected the white plane protruding from the Earth (a plane determined by the Earth's center O in addition to the two end footprints (A₁ and A₂) of a single cross-track scan) to examine the geometric relationships in panel (b).**

## Appendix B: Grids of LUT Dimensions in ZJU-AERO

The grids of LUT dimensions are presented in Table B1. Only the Z and K matrix elements that appear in the formulas of polarimetric radar variables in Eq. **Error! Reference source not found.-Error! Reference source not found.** are stored in the database. For temperatures, we set the minimum temperature for super-cooled liquid particles as 233 K, considering that homogeneous freezing starts at lower temperatures. For solid hydrometeors, the lowest temperature in the database is 203K, and lower environment temperatures encountered are taken as 203K. We provided LUTs for 6 bands that are widely used for ground-based and spaceborne weather radars. The ranges of diameters and aspect ratios for solid hydrometeors are suggested by observations of field-research (Garrett et al., 2015), while the range of diameters and aspect ratios of liquid hydrometeor have already been discussed in Section 3.3. The number of diameter bins is sufficient to produce a reasonable size spectrum of hydrometeors for various hydrometeor mass concentrations, as shown in Figure 10. The range of mass concentration grids for BSP LUTs is the same with that of Geer et al. (2021). Other external LUTs, such as Eriksson et al. (2018) need to undergo format conversions and interpolations (e.g., for different grids of diameter) before it can be applied in ZJU-AERO.

**Table B1: Grid specifications for Look-Up Table (LUT) dimensions in ZJU-AERO. The format "[START:A, END:B, STEP:C]" represents a sequence of numbers ranging from A to B with an increment of C. The format "LIN[MIN:A, MAX:B, NUMBER:C]" denotes a sequence of evenly-spaced numbers on a linear scale ranging from A to B with a total number of C values, while "LOG[MIN:A, MAX:B, NUMBER:C]" denotes a sequence of evenly-spaced numbers on a log scale.**

| Dimensions | Grids |
|---|---|
| Matrix Elements | Mueller Matrix $\mathbf{Z}$ : $Z_{11}$, $Z_{12}$, $Z_{21}$, $Z_{22}$, $Z_{33}$, $Z_{34}$, $Z_{43}$, $Z_{44}$ |
|  | Extinction Matrix $\mathbf{K}$ : $K_{11}$, $K_{12}$, $K_{34}$ |
| Temperatures [K] | Solid Hydrometeor (Snow, Graupel): 203, 213, 223, 233, 243, 253, 263, 273 |
|  | Liquid Hydrometeor (Rain): 233, 238, 243, 248, 253, 258, 263, 268, 273, 278, 283, 288, 293, 298, 303, 308, 313, 318 |
| Frequencies [GHz] | 2.7 [S], 5.6 [C], 9.41 [X], 13.6 [Ku], 35.6 [Ka], 94.1 [W] |
| Mass Concentrations [kg·m$^{-3}$] | LOG[MIN: $10^{-6}$, MAX: $10^{-2}$, NUMBER: 161] |
| Diameters [mm] | Snow ($D_{max}$): LIN[MIN: 0.2, MAX: 20, NUMBER: 128] |
|  | Graupel ($D_{max}$): LIN[MIN: 0.2, MAX: 15, NUMBER: 128] |
|  | Rain ($D_{eq}$): LIN[MIN: 0.1, MAX: 9, NUMBER: 128] |

| Elevations [°] | [START: -90, END: 90, STEP: 1] | | |
|---|---|---|---|
| Beta Angles [°] | [START: 0, END: 180, STEP: 1] | | |
| Reciprocal of Aspect Ratio [-] | Snow: [START: 1.1, END: 5.1, STEP: 0.2] | | |
| | Graupel: [START: 1.1, END: 3.1, STEP: 0.1] | | |
| | Rain: Single Value (see Figure 5) | | |

To alleviate the truncation and quadrature error in particle size integration, a renormalization technique is applied after the particle number in each size bin is calculated. We first calculate the renormalization factor $\kappa$ :

$$\kappa = \frac{\sum_{i=1}^{nbins} m(D_i)N(D_i)\Delta D}{Q_m}, \tag{B1}$$

Here, $m(D_i)$ is the mass of hydrometeor particle with a diameter of $D_i$, $N(D_i)$ is the particle number in that size-bin, while $\Delta D$ is the diameter step of the numerical integration. Then ($\kappa < 1$) gives the ratio between the mass concentration presented by our PSD and the mass concentration of hydrometeors given by NWP model $Q_m$. Then we can calculate the corrected particle number distribution $N^{corr}(D_i)$:

$$N^{corr}(D_i) = \frac{N(D_i)}{\kappa}, \tag{B2}$$

**Appendix C: Forward Scattering Alignment (FSA) and Backward Scattering Alignment (BSA)**

Although the optical properties of hydrometeors are consistently computed and stored in the FSA convention, this convention is not used for mono-static radar applications. Figure C1 displays the differences between the Back Scattering Alignment (BSA) convention and the FSA convention (Chandrasekar, 2001). The incident light propagates along $–X_L$ direction and comes into contact with the particle at the origin **O**. $\hat{h}, \hat{v}, \hat{k}$ are used to represent unit vectors of horizontal-polarization, vertical-

polarization, and propagation direction of electromagnetic wave, respectively. If we use the definition of h and v unit vectors under the FSA convention, the horizontal unit vector of backscattering light is then inconsistent with the incident light, $(\hat{h}_{FSA}^{inc}, \hat{v}_{FSA}^{inc}, \hat{k}_{FSA}^{inc}) = (-\hat{h}_{FSA}^{bsca}, \hat{v}_{FSA}^{bsca}, -\hat{k}_{FSA}^{bsca})$, in which "bsca" indicates backscattering light. To resolve this convention conflict between polarimetric radar observation system and scattering computation, BSA forces the following relationship by

reversing the direction of the backscattering horizontal unit vector definition: $\left(-\hat{h}_{FSA}^{bsca}, \hat{v}_{FSA}^{bsca}, -\hat{k}_{FSA}^{bsca}\right) = \left(\hat{h}_{BSA}^{bsca}, \hat{v}_{BSA}^{bsca}, \hat{k}_{BSA}^{bsca}\right)$.

Hence, the amplitude scattering matrix in the BSA convention is related to that of FSA:

$$
\begin{bmatrix} S_{hh} & S_{hv} \\ S_{vh} & S_{vv} \end{bmatrix}_{BSA} = \begin{bmatrix} -1 & 0 \\ 0 & 1 \end{bmatrix} \begin{bmatrix} S_{hh} & S_{hv} \\ S_{vh} & S_{vv} \end{bmatrix}_{FSA} = \begin{bmatrix} -S_{hh} & -S_{hv} \\ S_{vh} & S_{vv} \end{bmatrix}_{FSA}, \tag{C1}
$$

The following amplitude scattering elements used for polarimetric radar variables computation are represented in the BSA convention unless otherwise stated (namely, a conversion is needed in the core submodule before obtaining radar variables).

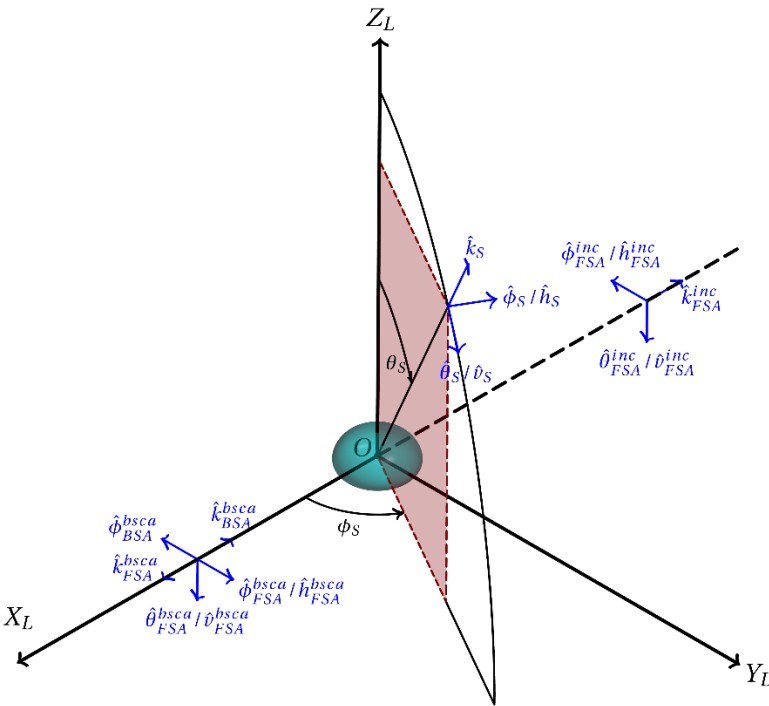

**Figure C1: The unit vectors in the definition of the amplitude scattering matrix. We assumed that the particle is located at the origin of the laboratory coordinate system, and the incident light propagated along the –X$_L$ direction. The unit vectors subscripted with Forward Scattering Alignment (FSA) are unit vectors defined in the FSA convention, while unit vectors defined by Back Scattering Alignment (BSA) convention are indicated by subscription BSA.**

**Appendix D: Calculations of Radar Variables**

Next we describe how to perform intrinsic radar variables calculations using bulk matrices $\left\langle \mathbf{Z} \right\rangle$ and $\left\langle \mathbf{K} \right\rangle$:

1. Horizontal/Vertical reflectivity $z_{h/v}$ in units of mm$^6$·m$^{-3}$:

$$z_h = \frac{\lambda^4}{\pi^5 |K_w|^2} \int_0^\infty \sigma_{bsca,h} \, N(D) dD = \frac{4\pi\lambda^4}{\pi^5 |K_w|^2} \int_0^\infty |S_{hh}|^2 \, N(D) dD$$

$$= \frac{2\lambda^4}{\pi^4 |K_w|^2} \int_0^\infty (Z_{11} - Z_{12} - Z_{12} + Z_{22}) \, N(D) dD \qquad , \tag{D1}$$

$$= \frac{2\lambda^4}{\pi^4 |K_w|^2} \left( \langle Z_{11} \rangle - \langle Z_{12} \rangle - \langle Z_{21} \rangle + \langle Z_{22} \rangle \right)$$

$$z_v = \frac{2\lambda^4}{\pi^4 |K_w|^2} \left( \langle Z_{11} \rangle + \langle Z_{12} \rangle + \langle Z_{21} \rangle + \langle Z_{22} \rangle \right), \tag{D2}$$

In Eq. (D1), $\sigma_{bsca,h}$ indicates the horizontal backscattering cross-section of a particle (namely, $\sigma_{bsca,h} = 4\pi |S_{hh}|^2$). $\lambda$ is the

wavelength of radar, $K_w = |(\varepsilon_w - 1)/(\varepsilon_w + 2)|$ is the dielectric factor ($\varepsilon_w$ is the dielectric constant of water at the wavelength of radar for a fixed temperature of 283.15 K). Similarly, vertical reflectivity can be derived in Eq. (D2).

2. Differential reflectivity $z_{dr}$ (dimensionless):

$$z_{dr} = \frac{z_h}{z_v} = \frac{\langle Z_{11} \rangle - \langle Z_{12} \rangle - \langle Z_{21} \rangle + \langle Z_{22} \rangle}{\langle Z_{11} \rangle + \langle Z_{12} \rangle + \langle Z_{21} \rangle + \langle Z_{22} \rangle}, \tag{D3}$$

3. Specific differential phase shift (deg.·km$^{-1}$) upon propagation, $K_{DP}$, is defined by

$$K_{DP} = 10^{-3} \cdot \frac{180}{\pi} \cdot \lambda \int_0^\infty \Re(S_{hh}^{fwd} - S_{vv}^{fwd}) \, N(D) dD = 10^{-3} \cdot \frac{180}{\pi} \cdot \int_0^\infty \frac{2\pi}{k_0} \Re(S_{hh}^{fwd} - S_{vv}^{fwd}) \, N(D) dD$$

$$\qquad \qquad , \tag{D4}$$

$$= 10^{-3} \cdot \frac{180}{\pi} \cdot \langle K_{34} \rangle$$

Here, $10^{-3}$ in Eq. (D4) represents the coefficient in unit conversion from mm$^2$·m$^{-3}$ to km$^{-1}$, and the coefficient $180/\pi$ is used to convert the unit of result from radii·km$^{-1}$ to deg.·km$^{-1}$. Here, the superscripts of "fwd" over matrix elements indicate that these are elements of the forward scattering matrix.

4. The one-way linear-scale attenuation coefficient[5] of horizontal/vertical polarization $a_{h/v}$ in units of km$^{-1}$ :

---

[5] The one-way specific attenuation in dB-scale $\alpha_{h/v}$ in units of dB·km$^{-1}$, which relates to the linear-scale attenuation coefficient by

$\alpha_{h/v} = 10\log_{10}(e) \cdot a_{h/v} = 4.343 \cdot a_{h/v}$, is also used in many studies of weather radar that consider the change of logarithm-base from e to 10.

$$a_h = 10^{-3} \int_0^\infty \sigma_{ext,h}\, N(D)dD = 10^{-3} \int_0^\infty \frac{4\pi}{k_0} \Im(S_{hh}^{fwd})\, N(D)dD$$

$$= 10^{-3} \int_0^\infty (K_{11} - K_{12})\, N(D)dD = 10^{-3} \left( \langle K_{11} \rangle - \langle K_{12} \rangle \right) \tag{D5}$$

$$a_v = 10^{-3} \left( \langle K_{11} \rangle + \langle K_{12} \rangle \right), \tag{D6}$$

Again, $10^{-3}$ in Eq. (D5) serves as the coefficient in unit conversion from mm$^2\cdot$m$^{-3}$ to km$^{-1}$. $\sigma_{ext,h}$ indicates horizontal

extinction cross-section of a given particle (i.e., $\sigma_{ext,h} = \dfrac{4\pi}{k_0} \Im\left(S_{hh}^{fwd}\right)$). Similarly, vertical attenuation coefficient can be

derived in Eq. (D6).

5. Total differential phase shift upon backscattering $\delta_{hv}$ in units of deg. is represented as shown below:

$$\delta_{hv} = \frac{180}{\pi} \angle \left( \int_0^\infty S_{hh} S_{vv}^*\, N(D)dD \right) = \frac{180}{\pi} \angle \left( \int_0^\infty [0.5(Z_{33} + Z_{44}) + 0.5(Z_{43} - Z_{34})i]\, N(D)dD \right)$$

$$= \frac{180}{\pi} \angle \left( \left[ \langle Z_{33} \rangle + \langle Z_{44} \rangle \right] + \left[ \langle Z_{43} \rangle - \langle Z_{34} \rangle \right] i \right) \tag{D7}$$

The notation '$\angle$' in Eq. (D7) indicates the phase of the complex value. The coefficient $180/\pi$ is used to convert the unit of result from radii to deg.

6. Co-polar correlation coefficient $\rho_{hv}$ (dimensionless):

$$\rho_{hv} = \frac{\left| \int_0^\infty S_{hh} S_{vv}^*\, N(D)dD \right|}{\sqrt{\int_0^\infty |S_{hh}|^2\, N(D)dD \cdot \int_0^\infty |S_{vv}|^2\, N(D)dD}} = \frac{\left| \int_0^\infty [0.5(Z_{33} + Z_{44}) + 0.5(Z_{43} - Z_{34})i]\, N(D)dD \right|}{\sqrt{\int_0^\infty 0.5(Z_{11} - Z_{12} - Z_{21} + Z_{22})\, N(D)dD \cdot \int_0^\infty 0.5(Z_{11} + Z_{12} + Z_{21} + Z_{22})\, N(D)dD}},$$

$$= \frac{\left| (\langle Z_{33} \rangle + \langle Z_{44} \rangle) + (\langle Z_{43} \rangle - \langle Z_{34} \rangle)i \right|}{\sqrt{(\langle Z_{11} \rangle - \langle Z_{12} \rangle - \langle Z_{21} \rangle + \langle Z_{22} \rangle) \cdot (\langle Z_{11} \rangle + \langle Z_{12} \rangle + \langle Z_{21} \rangle + \langle Z_{22} \rangle)}} \tag{D8}$$

In Eq. (D8), the co-polar correlation coefficient $\rho_{hv}$ is the amplitude of complex co-polar correlation coefficient $\widetilde{\rho_{hv}}$, whose phase is the total differential phase shift upon backscattering in Eq. (D7).

Please note that the summation over hydrometeor types were omitted in Eqs. (D1)-(D8) for the sake of clarity. Readers can easily obtain the more complicated real expressions of radar variables by applying extra summations over hydrometeor types for $\langle \mathbf{Z} \rangle$ and $\langle \mathbf{K} \rangle$ elements before carrying out calculations.

The aforementioned radar variables $z_{h/v}$, $a_{h/v}(\alpha_{h/v})$, $z_{dr}$, $K_{DP}$, $\delta_{hv}$ and $\rho_{hv}$ are often referred as intrinsic radar variables determined by the atmosphere and hydrometeor conditions in local radar gates. Under the assumption of first-order multiple scattering model, although the wave is scattered only once before it is received, the two-way propagation effects such as attenuation and phase-shift are taken into account by using the wave number of effective medium composed of air and
hydrometeors along the beam trajectory (Zhang, 2016). We can derive the observable radar variables as shown below:

$$z'_{h/v}(r_g) = z_{h/v}(r_g) \cdot \exp\left(-2\int_{r=0}^{r=r_g} a_{h/v}(r)\mathrm{d}r\right), \tag{D9}$$

$$\Phi_{DP}(r_g) = 2\int_{r=0}^{r=r_g} K_{DP}(r)\mathrm{d}r + \delta_{hv}(r_g), \tag{D10}$$

In Eqs. (D9) and (D10), $r_g$ is the range of the radar gate, the variable of integration $r$ is the range along the radar beam trajectory. $z'_{h/v}$ marked with a prime is the horizontal / vertical observable reflectivity of the radar gate, attenuated on the way
from transmitter to the particle and the way back from particle to the receiver. $\Phi_{DP}$ is the total phase shift interpreted as the differential phase shift upon two-way propagation plus the differential phase shift upon backscattering.

$$Z'_{H/V} = 10\log_{10}\left(z'_{h/v}\right), \tag{D11}$$

$$Z'_{DR} = 10\log_{10}\left(z'_{dr}\right) = 10\log_{10}\left(\frac{z'_h}{z'_v}\right), \tag{D12}$$

Eqs. (D11) and (D12) give the observable horizontal / vertical reflectivity $Z'_{H/V}$ in units of dBZ and differential reflectivity
$Z'_{DR}$ in units of dB.

**Acknowledgements**

This research was supported by National Natural Science Foundation of China (U2342213;42022038); National Key Research and Development Program of China (2022YFC3004004). We are grateful to NASA and JAXA for providing the L2A data of the dual-frequency spaceborne precipitation radar GPM/DPR, which serves as a good validation dataset for the present
spaceborne forward radar operator.

## Code Availability

Codes of the forward radar operator ZJU-AERO V0.5 and the packaged Conda environment and the user manual are available on ZENODO (https://doi.org/10.5281/zenodo.11307123). The database of scattering properties (i.e., the LUTs) are also released with the software package.

## Data Availability

Two cases of this forward radar operator are presented in the user manual of ZJU-AERO for the demonstration purposes of its usage for the space-borne and ground-based radar, respectively. The NWP model grid data and the radar observation products for those two demonstration cases are also available on ZENODO (https://doi.org/10.5281/zenodo.11307206).

## Author Contribution

Hejun Xie performed the coding, visualization of the LUTs, and designed the case study experiments. Lei Bi and Wei Han supervised this study. All authors contributed to the writing of the paper.

## Competing Interests

The contact author has declared that none of the authors has any competing interests.

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
