# Peer review of "ZJU-AERO V0.5: An Accurate and Efficient Radar Operator Designed for CMA-GFS / MESO with Capability of Simulating Non-spherical Hydrometeors"

_Geoscientific Model Development, 2023_

## Referee Comment (RC2)

**Review Xie et al. „ZJU-AERO V0.5: An Accurate and Efficient Radar Operator Designed for CMA-GFS / MESO with Capability of Simulating Nonspherical Hydrometeors "**

The paper presents the radar forward operator ZJU-AERO that is capable of simulating radar observables for both ground-based and air-/spaceborne sensors and of considering effects of non-spherical, non-homogeneous hydrometeors incuding polarimetry, designed particularly for use along with the Chinese Meteorology Administration's numerical weather prediction models. A larger fraction of the paper deals with evaluation of the Chebyshev shape model for rain suggested by Chuang and Beard (1990) compared to more classical spheroid models.

In general, this topic is suitable for publication in GMD. However, the paper "oscillates" between some very basic, textbook-like descriptions entangled with some quite specific, not always relevant details, but lacking clear direction and clarity and detail at places where it would start to be really interesting. I suggest publication to be considered after major revisions.

Please find below my detailed review report.

**General remarks**

The manuscript, in my view, presents a mixture of on the one hand quite basic, but very detailed content and on the other hand seemingly more novel, but hurried over, too superficial content. It seems to me like an extract from a thesis, with individual parts not stitched together very well for providing a consistent and concise work with a clear common thread. It contains a number of rather fancy illustrations that, however, do not necessarily provide relevant information, and that often are not covered with appropriate detail in text, ie that rather seem to be there to look nice than to make a relevant contribution (eg Figs1, 4, 5, 6).

It often remains unclear what has been done by the authors, what has been by other work, and what has been taken over quite directly from other works (general approaches, more specific methods, algorithms, possibly even code, …), and how particularly the latter has been incorporated into this work. For example, L139 refers to Zeng16 for sub-beam sampling and averaging methods in submodules B1 and B2 in a way that seems to imply Zeng16 describes a part of ZJU-AERO ("see Zeng et al. (2016) for more about"); however, Zeng16 describes another, independent radar forward operator (RFO) and it remains unclear whether the reference is "just" to a general approach or to specific formulas and their implementation or to the taking over of entire code bits. Similar for the Wolfensberger and Berne (2018) reference in L146. Also, L309 "To address these uncertainties, a field research study was conducted" also at best leaves it unclear whether the study was done by the authors (that's what this formulation suggests to me) or others. So please be more clear in your formulations!

Improve presentation style:

- First present results (describe them), then discuss / explain them, then conclude.

- Figure contents, when shown, need to get presented and discussed, otherwise leave them out (e.g. polarimetric variables).

- Equations should always be integrated in text, ie be part of a sentence that makes sense when being read out loud, not stand alone and/or a few lines before or after they are introduced.

- Introduce math symbols at first use. Use unambiguous notation, e.g. avoid to use λ for both radar wavelength and PSD parameter, Dmax for maximum diameter (or max dimension?) and upper PSD integration limit, ...

- Introduce acronyms at first use. Then use them, without re-introducing.

- Figure captions here are generally overlong with partly redundant or irrelevant content. Figure caption are supposed to describe what is seen in the figure. All details on data processing, implementations, setups and the like should instead be in the text.

Improve language:

- Spell-check.
- Use tenses consistently throughoughout the manuscript. For things done in this study and presented in this paper: either present or past; I suggest present tense.
- Be careful with fill words (e.g. "however", "nevertheless", "fortunately" (?!?), "It is worth to note/mentioning", "obviously", ...) that inappropriately (unintentionally?) relativize or judge.
- e.g. L108: what is an "approximately(!) 4/3Re-radius curve"? L110 "rarely" rather "weakly" or "negligible", ...

**Specific remarks**

Abstract: It introduces ZJU-AERO as polarimetric RFO with one main objective bein the assimilation of ground-based radar polarimetric variables. The remainder of the manuscript, however, has little on polarimetry – it introduces/defines the variables, presents some calculations, but without any relevant discussion of them.

Introduction: It might be worth mentioning Zhang21 in the state-of-the-art review of RFOs, too, as it targets fast applications like data assimilation and contains a melting scheme module.

L48f: "Zeng et al. (2016) [...] but it focuses on non-polarimetric variables." – Recent versions of the Zeng et al. (2016) RFO EMVORADO also cover simulations of polarimetric radar variables (see Trömel21, Shrestha22).

L131ff: As this covers two distinct submodules, B1 and B2, make clear in the text which descriptions belong to which submodule.

L147ff: Where is the input to module (D) taken from? User set? Model driven?

L149f: "regular morphology [...] an analytical expression can then be used to retrieve the PSD" – Too specific at this point in the manuscript, where it is everything but obvious what particle morphology has to do with PSD.

L153ff: Which module does this? Language/grammar of this paragraph needs care.

L158: Does this module have a letter-name? "calculates intrinsic polarimetric radar variables on each radar gate" – Wouldn't that rather be on each subbeam-gridpoint within each radar gate, or where is this done?

L153ff: Do numbered items 5.-7. all belong to module (E)? If so, better make them subitems 5.1-5.3?

Fig2: It is unclear, how module (E) links to the multi-level database (and remains so throughout the manuscript): Is the DB external and can other DB be used, too (e.g. the ARTS-SSDB by Eriksson18)? What data would they need to provide? Does module (E) get LevelB data? Why cover LevelC here then? Where does LevelA->B conversion gets its settings/assumptions on orientation preference and shape parameters from? Is that hard-coded?

L173ff & Fig3: This does not seem particularly relevant. If relevant, give it its own subsection at an appropriate place in the manuscript. Under "Flow Chart and Concepts" it definitely seems out of place and far too specific.

L195ff: "Since those topics have already undergone sufficient discussions in other works concerning radar operators, and we just inherited those settings and options from them. Readers with an interest in these options may refer to the bibliography mentioned in the ZJU-AERO general workflow description text." – That's far too general and not appropriate for a scientific publication as this does not facilitate reproducability of your research. You need to reference clearly, which parts come from which reference and specifiy what "inherit" means in each case (general approach, specific methods, implementations, …).

Sec2.2: Most of the time I appreciate when papers provide sufficient and clear theoretical background. So I have ambiguous feelings about criticizing this section. However, it seems a bit too basic and too drawn out textbook knowledge. Beside that, it's incomplete: What is L (L209), what are the h, v, k (L215)? It jumps from theoretical background to very specific ZJU-AERO things (database Level A/B units). Figure 4 does not (at least not to me) "explain why the BSA convention should be introduced" – as the reader, I don't care whether it "should be introduced" – just that you introduce it (if it is relevant).

L227: "we also have the 4x4 real matrix" – formulation. "one can define"? what are the actual properties of S and of Z and K, what's the difference, what are their respective roles/uses? what is "the Stokes vector space"

L238: This paragraph doesn't make sense to me. Maybe a language issue? What's the role of the optical theorem here? "This is because" – what exactly is because of what?

L243ff: I find conversion formulas S -> Z,K actually more relevant than providing radar variable formulas in terms of S.

L265: What is the tilded rhohv? "is the magnitude of […] whose amplitude is" – i might be off, but in my understanding the amplitude is the magnitude of a complex number. So why would they be two different variables (tilded rhohv and deltahv)?

L279: What's the difference between "reflectivity factor" and "reflectivity"? The path attenuation?

Tab1: Shape distribution information is missing here (in text, Wolfensberger and Berne (2018) is given, ie Garrett et al. (2015) can't contain this).

L316: "in which y is the reciprocal of the aspect ratio" – Define what aspect ratio is for you. There are different definitions around. Compare, e.g., Ryzhkov's (minor axis / major axis, ie AR<=1) or Mishchenko (rotational axis / symmetry-plane axis, ie AR<1 for oblates, AR>1 for prolates).

L321: "The mass–diameter relationship is crucial in determining the PSD" – why?

Sec2.3: For rain, is there also a shape distribution assumed?

L336: "it seems that a microphysics-consistent mass-diameter relation […]$Dmax^3$ for snow and graupel" – it seems??? how would $Dmax^3$ be microphysics-consistent? What is rho_sp?

Eq22: That doesn't make sense to me. In this notation $F(\lambda)$ will be 0. What are F and F' anyways?

L355: What is V, where do you get that come from?

Sec3: There is far too little information on how the database is constructed, e.g. how many/which grid points are used for each of the dimensions, what are the integration limits etc. And how that has been decided. And how much impact that has on the final radar variables.

L366: "This shape differs from the traditional spheroid shape commonly used in other radar observation operators. Therefore, in this section, we will delve into the design of the database" – Why is that relevant for the DB design, i.e., how does that affect the DB design or is a good example for it? Moreover, where design is discussed (subsecs 3.1 and 3.2), it's neither relevant whether it's rain or any other hydrometeor category nor what the shape model is. On the other hand, subsections 3.3-3.5 do not present or discuss DB design, but evaluate the contents of the DB.

L386: "of the lookup table (stored in netCDF4 format)" – Are all DB Levels stored in one table/file?

Tab2: "that a single database for one hydrometeor and once frequency occupies" – what is a "single database"? As opposed to a (single) lookup table (file)? Does "one hydrometeor" refer to one hydrometeor class (ie all sizes) or a single particle size?

Tab2: Angle brackets typically indicate integrated variables. Note, that even RSSP and ASSP Z and K in your DB are already integrated (over azimuthal orientation and shape/canting distributions, respectively), ie they should technically have angle brackets, too.

Tab2: Are the full 16 elements of both Z and K kept (since only the diagonal 2x2 blocks of Z and only 3 entries of K are used)?

Tab2: How are the different PSD handled in the DB? Separate tables (files) (or DBs?)?

L396: This whole paragraph seems rather basic. Or standard. A textbook reference, eg to Mishchenko, would suffice in my view (but references are urgently needed in any case!). It stops to elaborate, however, where it would get more interesting, e.g. how the reference orientation is set for prolate or for irregularly shaped particles.

L404f: "ZYZ convention" is at least ambiguous since there are two coordinate systems with unidentical axes, ie with respect to which of these axes (L- or P-system) are the rotations performed?

L408: What is meant/referred to by "arbitrary orientation preferences"? Do you mean arbitrary orientations (but that wouldn't cover numbered item 4)? or actually  orientation preferences, but numbered item 4 is neither arbitrary (random in alpha and gamma is already quite specific).

Eq(24): Standalone equation without any referring text??? Why X has one and two overline(s) here, but Z and K don't have in Tab2 RSSP and ASSP entries, respectively?

L432: "Eq. (25) gives [...] Level A to Level B database conversion tool" – p(beta) given here is not general, but specifies a Gaussian distribution in polar angle with standard deviation sig_beta.

L434: "It is important to consider particle symmetry" – Not that important, in my view. One can save a bit of computational resources, but results will still be correct when symmetries are ignored. However, it is far more important – and relevant for ensuring correct results – to not accidentally implicitly assume any symmetries in arbitrary particle shapes (considering or neglecting flipped and/or mirror-symmetric counterparts).

L439f: "a more accurate [...] commonly used spheroid model" – add reference for this statement (Ekelund20 rather concluded that Chuang and Beard model is no significant improvement at least regarding radar reflectivity compared to the spheroid model)

L442: "The obtained results were then fitted [...]" – by whom? Are you still summarizing Chuang and Beard (1990)? Or is this your own work? Reference properly – this applies to the whole paragraph, actually (if it's all Chuang and Beard, reduce to only repeat the crucial info and refer to Chuang and Beard for the rest – e.g. why are the truncation and the Deq range and spacing in the coefficient calcs relevant here?)

L452: "Comparing [...] with significantly different aspect ratios is meaningless" – Why is that? Predicted radar variables (for a given D or PSD) are the relevant parameters to be compared,

regardless of the underlying aspect ratio, or the maximum dimension, or any other possible affecting particle property. It is interesting, though, to understand which and how impact each of the properties contributes.

L458: "Therefore, we can confidently assert" – I do not fully agree as this does not provide info yet on how much difference in SSP results from such seemingly small diffs in gamma. Only comparing SSP of particles with equivalent gamma and Deq could provide that confirmation.

L460: Does equal Deq imply equal Dmax (i.e. a=a')? Hence do diffs in gamma exclusively result from differences of minor axes b and b'?

L467: "The initial examination [...] by Ekelund et al. (2020)" – Reformulate. This seems to imply Ekelund20 were the first to study rain drop SSP (I'm not even fully sure, they were the first using the Chuang and Beard and/or the Chebyshev model – actually, Aydin gives a short summary of Chuang and Beard effects in their Chapter on cm- and mm-wave scattering from hydrometeors in the Mishchenko00 book).

L468: "they used a modified version of the EBCM code" – modified in what way? Ekelund et al. themselves state they used "the Fortran T-matrix code developed by  by Mishchenko (2000)" and don't mention modifications.

L471: "To ensure [...] a user-friendly radar operator interface, this study presents an optical proerty database [...] using the IITM code" – Check language: is it really that presenting the study ensures those things? What does user-friendliness have to do with shape model and T-Matrix code applied?

L479: "raindrops with their Deq larger than 8mm disperse easily" – Provide a reference for this statement. Ekelund20 argue with 5mm, supported by two references.

L482: "we need to introduce intermediate quantities called the SSP factors" – Why do you NEED to introduce them? You just chose to (hopefully because they are illustrative and facilitate understanding). However, what do you mean by "SSP factors", what are they? Explain!

L483: "The SSP factor [zhv] for the unpolarized reflectivity" – Why using unpolarized reflectivity when radars ALWAYS measure a specific polarization? Using hv as subscripts in z seems a bad choice since zhv usually refers to horizontally transmitted/vertically received reflectivity occuring as the nominator in the definition of linear depolarization ration, LDR.

L486: "enable us to assess the contribution of particles with a diameter Deq to the radar reflectivity" – How? Provide a proper explanation of what the SSP factors are. And isn't that, ie "contribution of particles with Deq to reflectivity", what backscatagrand later will provide?

Fig8/Fig9: Looks fancy, but would a simple cartesian plot not suffice, be easier to digest and be less prone to biased interpretation? Particularly since the center of b) panels is not 0, but 60 (ie deviations are even smaller than they appear here). Radial axis of a) panels is unclear – it's neither linear nor true logarithmic, isn't it? (also, the middle value, I guess, should rather be $5.10^2$ instead of $5.10^3$). Displaying z-equivalents in log scale (ie dB) seems more suitable than linear scale (ie $L^2$ units) to judge relevance/significance of differences in radar applications. Why are you showing the elevation range, apart from -30 to 30° masked sector, twice? That's not symmetry, but identical parameters elevations!

Fig8: Why is T-variation relevant to show? (Is variation with T entirely due to temperature dependence of refractive index, or are there any other T-dependent parameters?)

L517ff/Fig8: "we found that the [zhv] factor for ground-based observation geometries (e=0~20°) was significantly weaker" – First, differences at e=0° seem (close to) 0 to me. Second, what differences do you consider "significant"? I find them anyways hard to judge in linear units, this would be much easier in dB units.

L524ff: "demonstrate the stability of the deviations [...] in terms of the temperature [...] However, the finding is different for [zhv] against the orientation preference" – Yes, of course. Varying orientation present a significantly different (geometrical) cross-section to the observer, while varying T (apart from phase changes) essentially keeps the same "view" of the roughly same particle.

L528f: "Fortunately, the column 'orientation preference' in Table 1 shows that the standard deviation of beta=7° did not exceed this threshold for raindrops." – Fortunately? First, consider language: fortunate might be that rain drop tumbling is typically small, but it hasn't to do with fortune that your Table shows the value it shows (I hope at least; or did you throw dices?). Second, why would higher sigma be "unfortunate"? And what would you do in such an "unfortunate" case, like eg in strong turbulence cases?

Fig10: That is quite a rich plot. Considering that, there is very little presentation of its content and discussion/interpretation of it in the text. Particularly, there is practically none at all on the polarimetric variables. Without that, skip the respective plots/panels.

L540f: "focus on describing the PSD options [...] (namely, ASSP to BSP conversion" – This subsection is supposed to be about DB Level B, i.e. according to Fig.2 the orientation and shape distribution averaging. Hence unclear why PSD appears here already.

L542: "a total of six options for PSD schemes for raindrops [...] listed in Table 3" – Inconsistency? Tab.1 only lists 4 options?!

Tab3: Meaning of x1 and x2 parameters to be introduced/shown by an equation. What is the relation of the six schemes listed here to the Cx schemes listed in Tab1?

L572: "Figure 11 demonstrates that the uncertainty of intercept parameter No [...]" – How? Explain better. Do you really mean uncertainty? Or rather spread among PSD schemes?

L572: "Thompson scheme [...] priorities smaller drops" – prioritizes. Can this be seen in Fig11? How?

L575ff: "For example, stratocumulus [...] production of unusual PSDs" – I do not understand what you are trying to say and what this has to do with PSD in RFO or even with DB Level B/RSSP.

L579: "we introduced the concept of 'backscatagrand'" – Language/writing style: You didn't introduce anything yet, you are going to do that now. Don't announce that, but do it. Not just by naming it and already pointing to results, but EXPLAIN it. The current attempt on explanation (L583ff) is too confused, too little to the point.

Fig12: Include Thompson tuned here as it is used as the basic PSD in Sec4 (there even before discussing the need for and approach of tuning).

L604: "It can be concluded from Figure 12" – Before making conclusions, present the results and discuss them.

L609: "unique to modern PSD schemes" – Is it? Why? What defines a "modern" PSD scheme? Is T08 one or not? Besides, though not exactly coinciding peak & dip for MP48, backscatagrand still exhibits a kind of dip (or hiatus in its decrease).

L610: "leading to a loss of bulk reflectivity" and L615f: "T08 scheme must be much smaller" – Would ease discussion (by reducing unnecessary speculation), if you not just state, but show that. E.g. indicate resulting bulk zhv for each PSD and QR in the Figure 12 panels (lower row), e.g. by a symbol at placed at end of x-axis.

L617: "relative importance [...] can also be diagnosed with [...] backscatagrand" – that's its ultimate purpose!

L625: "we made speculations" -> "we hypothesized". However, it would make for a smoother read if you avoid the need for hypothesizing/speculation by already presenting the resulting equivalent bulk properties belonging to the Figure 12 curves.

L627ff: Which of these conclusions actually require/are affected by the chosen shape model? How do your findings on Chebyshev shape model compare to Ekelund20's findings?

L642f: "sensitivities of polarimetric intrinsic radar variables [...] were not examined in this paper" – Then there is no reason to present them at all!

Fig13: Is presenting polarimetric variables for close-to-nadir angles useful, or aren't they more interesting in ground-based measurements? Why only 3 PSD schemes included here? It might be more useful to present ZH and ZV separately (though not necessarily for for close-to-nadir observations).

Sec 3.3-3.5: What do these subsecstions provide beyond what Ekelund20 already did? That is, what justifies this lengthy presentation (it's 10 pages, ie ¼ of the whole manuscript). Particularly since Ekelund20 concluded differences in reflectivity from Chebyshev and spheroid shapes are negligible for ground-based weather radars and still pretty small in nadir-viewing geometries.

L668: "has switched its scan pattern" – How was it before? And/or how is that relevant here?

L678: "the structures [...] are described as contiguous and vague" – rather "appear more contiguous and vague"? Is this from visual impression or any quantitative measure?

L681: "simulated radar reflectivity should be unbiased [...] we assume [...] was roughly unbiased" – Are the reflectivites unbiased? Analyze and show! On what grounds do you assume the model to be unbiased?

L685f: "observations revealed a prominent bright band" – Really? Where? I can't identify that. Prominent BBs aren't particularly common in inhomogeneous atmosphere states.

L689f: "it was expected [...] have a positive OmB signature" – Why? Due to model lacking melting scheme? Be clear.

Fig14: What RFO setup is used here? Describe in text, not in figure caption!

L698: "altitude levels(namely, 0, 3, 5, and 8km)" – Isn't 0km in the blind zone?

L700: The post-processing information should go into the text and be described more clearly.

L713: "a minor negative OmB bias was observed" – Visually or using any quantitative means?

L713f: "which could be attributed to the over-attenuation" – What is over-attenuation? Why does it occur in Ka, not Ku? And only in simulations, not observations? Actually, it hasn't been made clear whether attenuation was considered in the simulations and whether the GPM observations used here were corrected for attenuation affects or not.

L718: "The OmB plots [...] useful tools for verifying and calibrating new observation operators and identifying their deficiencies" – How do you separate observation operator deficiencies from NWP model deficiencies?

L719: "the bias of distributions is reasonable and demonstrates the capabilities" – You did not really present any distributions (ie statistics)? Just scene snapshots.

Fig16: Why not binning simulation data equivalently to observations? Why showing 4-6 and 6-8km layers when not relevant for comparing rain PSD (and shape model) effects?

L734: "Based on statistical analysis" – why not showing the PDF of QR?

L738f: "As a result, applying the Thompson2008 PSD could lead [...] (Figure 13)" – Rather "According to Sec.../Fig.13, ...". Could? Under which conditions? Or does it?

L745: "Fortunately, we identified an optimal point" – By luck? Describe the tuning procedure.

L754: "However, we still believe that the CmS effects could be" – Reformulate. Believing is a bad argument in science. What supports this believe? References cited by Ekelund20, however, suggest that drops do not get larger in heavier storms, but break up instead and form more smaller ones.

L760: "demonstrated the data analysis tools in each layer" – Which layer? Which tools? You presented some analysis of rain drop DB data, but not any tools.

L765ff: Instead of a lengthy and fairly uncertain to-be-done wishlist, I'd prefer more summary of main findings of the analyses. Like a statement whether considering more sophisticated shape models than spheroids is actually really relevant.

L765: What is "geometric characterization"? Before dealing with melting particles, presenting frozen hydrometeor modelling seems more appropriate.

L777: "modules to interface with the data" – How does that fit (or where are these indicated) in Fig2 Flow chart?

**Spelling & other minor things**

L48: Remove second "described".

L78: Add reference for PMR and DPR properties (a more current one than Iguchi03, covering the instruments that are actually flying).

L79ff: Add reference for PR2 and RM constellation (what is RM, by the way?).

L83: "This study is organized into ..." – rather: "This paper is organized as follows."

L110: "rarely" rather "weakly" or "negligible"

L128: "This step facilitates" – In the meaning of "prepares for" (things done in module (C))?

L147: "specifies the settings of hydrometeor" – rather "properties"? Hydrometeors.

L167: "would be performed" – is? The whole "If" seems unnecessary – if N=M=1, then is can simply be seen to be performed over 1x1 pts.

L197: "At this point, we should" – This sentence should, if there at all, be at beginning of Sec2.2. Further, why "should" you?

L244: "Svv/hh" – wrong order of subscripts? It's not zh that is a function of Svv, but zv, is it?

L249 and elsewhere: "without dimension" -> "dimensionless"

L256f: Either use two separate equations for ah and av or use ± and ∓.

L262: In my understanding, that angle symbol typically rather indicates phase, not amplitude (see e.g. Wikipedia on complex numbers). Why not use the explicit calc formula as in zh and rhohv calcs (ie |x|)?

L264: "radii to deg. Again." – typo?

L316 and elsewhere: "possibility distribution" -> "probability distribution"

L322: "the hydrometeor category follows" – which hydrometeor category?

L326: "is a prescribed constant in the microphysics package" – which microphysics package? of ZJU-AERO? the CMA-NWP?

L330: "solve the unknown parameter $\lambda$" – provide $\lambda$ formula here, not 5 lines later (equivalently Eq(19) already belongs at "given the mass concentration" just before).

L360: what are BSP and SSP? First occurence, so spell out.

L361, 365: "now": what is that supposed to express?

L405: "we could uniquely determine" -> "any arbitrary orientation of the particle can uniquely be determined"

L407: "With a specified orientation and observation geometries" – Check grammar. Also, terminology is uncommon ("observation geometry" involves an observer in my interpretation, here it's rather the scattering geometry).

L409: "We used the T-matrix code" – I assume, here not the (or any specific) code is meant, but the T-Matrix approach or method.

L427: "the particle with original orientation" – original? Do you mean the reference orientation (alpha=beta=gamma=0°), ie where L- and P-system are oriented identically?

L442f: "as shown in Eq(26)" – put Eq(26) here and integrate into sentence.

L451: "more prone to aerodynamical effects" – meaning? why relevant here?

L481: "it is necessary to visualize" – To analyze, I agree. But that does not necessarily require visualization.

L485: "often referrred to as [...]" – Provide reference.

Eq(27): Check notation in equation. Really zhv == <[zhv]>, ie the SSP factor for the unpolarized reflectivity is equal the unpolarized reflectivity itself?

L495: "The beta dimension in the database are fixed at 0°" – Rather, the database entries for beta=0° are shown, while beta dimensions has more entries as suggested by Fig9.

L500: As L495, T was not fixed in the database, I assume, but the results for T=10°C shown here.

Fig10: In caption, explicitly point out which SSP factors are shown in each panel (particularly, info on what is shown in row 1 vs row 3 is missing).

L516: "This phenomenon [...], which is likely attributed" – Ekelund20 attribute it to that.

L539: "show a minor decay": Minor deviations/differences compared to beta=0° results? Or what is meant by decay?

L550: "Among the six schemes, group A is characterized" – First introduce that you distinguish two groups before characterizing each.

L564: "by assuming q_air=[...] of standard atmosphere" – At pressure of...?

Fig11: As RWC lines are only background information, present them less prominently (thinner lines, uncolored lines? or maybe as weakly coloured filled contours?). Could PSD scheme curve style be selected to make groups A and B easily distinguishable?

Fig12: Indicate more clearly in the upper-row panels that black & red curves are m(D)N(D) measuring on lefthand y-axis, while blue is sig_bsc/M(D) measuring on the righthand y-axis (e.g add blue in legend. and/or color the sig_bsc/m(D) formula in the top right of the panels blue). Axis labels and legend text should be increased to a readable font size.

Fig12: Use identical Deq ranges on x-axes over all panel for better comparability. Maybe scale lefthand y-axes (upper and lower panel) by QR to make shape of PSD better comparable.

L598f: "while the single-particle unpolarized backscattering per unit particle mass sig_bsc,hv(D)/m(D) is indicated" – Use the previously introduced (L592) terminology 'mass backscattering efficiency', otherwise you add confusion.

L602f: "Readers can verify that the" – Remove that unnecessary and odd intro (reader anyways cannot; they can at max roughly qualitatively guess), simply leave the following statement.

L605: "exhibit large uncertainties" – Are these indeed uncertainties? Or, simpler, differences?

L612: "Under moderately heavy precipitation conditions […] outlier" – Under all but the heaviest conditions, actually.

L614: "even though" -> "while"(?)

L663: "GPM-DPR's Ku-band radar is similar to" – meaning what?

L665: "accurate estimates of DSD can be obtained" – rather "more accurate" as they still have uncertainties. Reference missing.

L670: "defined as the difference" – of log space reflectivities, isn't it?

L684: "freezing level […] was found within" – in obs or model?

L686f: "It's worth noting" – Why? How is this relevant here, telling me what?

L694: "We plan to report […] in upcoming publications" – Rather "Implementation of […] will be subject to upcoming publications"

Fig14: Color scale with quasi-white region (around 18dBZ) seems not the best in combination with a white background.

L711: "The radar operator […]" – Remove; it doesn't belong in caption and is redundant anyways.

L752: "Not surprisingly" -> rather "As suggested by results in Sec3"

L762: "assessments of PSDs and morphology options" – Add: for rain.

L835: Geer et al. has long ago been published as proper GMD paper. Update.

**References**

[Eriksson18] Eriksson, P., R. Ekelund, J. Mendrok, M. Brath, O. Lemke, and S. A. Buehler: A general database of hydrometeor single scattering properties at microwave and sub-millimetre wavelengths. Earth Syst. Sci. Data, 10, 1301-1326, doi: 10.5194/essd-10-1301-2018, 2018.
[Shrestha22] Shrestha, P., Mendrok, J., and Brunner, D.: Aerosol characteristics and polarimetric signatures for a deep convective storm over the northwestern part of Europe – modeling and observations. Atmos. Chem. Phys., 22, 14095-14117, doi: 10.5194/acp-22-14095-2022, 2022.
[Trömel21] Trömel S, Simmer C, Blahak U, et al.: Overview: fusion of radar polarimetry and numerical atmospheric modelling towards an improved understanding of cloud and precipitation processes. Atmos Chem Phys., 21, 17291-17314, doi: 10.5194/acp-21-17291-2021, 2021.
[Zhang21] Zhang, G., J. Gao, and M. Du: Parameterized Forward Operators for Simulation and Assimilation of Polarimetric Radar Data with Numerical Weather Predictions. Atmos Sci., 38(5), 737-754, doi:10.1007/s00376-021-0289-6, 2021.

---

## Author Comment (AC1)

**Reviewer # 1**

**Summary:**

Radar forward operators transform the model's prognostic variables to the observed radar variables and are crucial tools in a number of applications including model evaluation, retrieval development, and data assimilation. The manuscript presents a newly developed radar forward operator based on the T-matrix method for calculating scattering amplitudes. The radar forward operator uses a non-spheroid hydrometeor model, namely the Chebyshev-shaped model to characterize the raindrops. The difference between the Chebyshev-shaped model and the spheroid model for calculating raindrop radar variables was compared in different single-moment (SM) schemes. However, the effect of the Chebyshev-shaped model on radar variables may only be notable in the presence of extremely large raindrops.

Overall, the topic of the manuscript is scientifically interesting and relevant. The presentation of the paper is reasonably good. The writing quality is fine and clear. I do have several comments below which I think translate into a recommendation for major revisions. I do not think there is anything fundamentally wrong with the manuscript, but considerable clarification on some points is needed as well as an expansion of the case results on polarimetric radar variables.

Response: We appreciate the reviewer's positive feedback and constructive comments on improving the manuscript.

We have implemented six typical single-moment (SM) drop size distribution (DSD) models into the radar operator and found that the Chebyshev raindrop model shows noticeable differences of bulk scattering properties (compared to the spheroid model) for electromagnetic waves of millimeter-wavelength at zenith and nadir observation geometries. This difference is more prominent for continental DSD models (e.g., Wang2016) with larger drops. The reflectivity contrast between zenith and nadir viewing angles can be as high as 4-5 dB, as illustrated in Figure 1. Such a difference is non-existent for spheroid raindrops with reflectance symmetry. Given the lower uncertainties in simulating reflectivity of liquid phase compared to solid and melting phases, such a difference deserves attention in specific applications such as comparing data from ground-based and spaceborne radar observations.

Figure 1: The BSP of raindrop for Wang2016 PSD scheme temperature=283K, elevation=1 deg., and Ka-band (35. 5GHz) for spheroid and Chebyshev shape raindrop at nadir (elevation=-90 deg.) at zenith (elevation=90 deg.)

Actually, this study not only presents an alternative raindrop particle model in the forward radar operator, but also demonstrates the capability for ZJU-AERO to handle particle models with unprecedented complexities.

Additionally, due to the current length of this manuscript, we have included a simulation case of ground-based polarimetric radar variables in the user manual of the software (released with the software package) rather than adding it to the research paper. This simulation case effectively showcases that our forward radar operator works properly in simulating polarimetric ground-based radar variables.

For further details, please refer to the specific responses provided below.

**Major comments:**

1. In the ABSTRACT and INTRODUCTION, the authors claimed the main purpose of developing this radar forward operator is to assimilate the polarimetric radar variables. However, it is not clarified in what follows that the operator has the potential to be able to be used for assimilating polarimetric radar. For the data assimilation purpose, the simplicity and efficiency of the radar forward operator are necessary. How computationally efficient is this radar forward operator? The variational method requires the tangent linear (TL) and adjoint (AD) operators, so whether this complex forward operator can be easily linearized. If the goal is to use it for data assimilation, then the manuscript needs to include sufficient discussion about the relevant

aspects (advantages, limitations, and alternatives) of the radar forward operator.

Response:

Thanks for the suggestions.

(1) We have added specifications on the techniques used to accelerate the forward simulations in section 2:

*"The above procedures of steps 1-5 (excluding the first step transferring the model state data from model grid to regular grid) are independent for each single beam, which guarantees the top-level scalability of the forward operator. Therefore, we are using the shared-memory python parallel library (multiprocessing) to speed up the simulation.*

*In addition, for those computationally intensive tasks (such as the trilinear interpolation in step 3), we are employing the technique of mixed programming (building C/Fortran-extensions that interface with python scripts) to further accelerate the computation.*

*The performance of the forward operator is generally satisfactory: ZJU-AERO can complete a ground-base station volume scan with 9 Plan Position Indicator (PPI) sweeps in 2 minutes on a modern laptop with a 6-core CPU (i7-10750H) if online size distribution integration is performed and the operator takes Level-B single scattering property database as input. If Level-C bulk scattering property database is used, such a volume PPI scan can be completed even faster (in 30s). Such efficiency can support data assimilation purposes, while also preserving flexibility for research purposes."*

(2). The first application of ZJU-AERO in CMA-GFS / MESO is planned to be used with the Bayesian approach. The data assimilation framework of the Bayesian method involves clustering the characteristics of atmosphere profiles using a statistic method. It only requires forward relationships that maps model states to the observational space, thus not needing tangent-linear (TL) or adjoint (AD) operators.

The development of future hydrometeor control variables in CMA-GFS/MESO by the variational method also shows promise. However, this approach requires more work in linearizing ZJU-AERO.

The direct linearization of this forward operator could be complex. One approach to circumvent the difficult could be to first find polynomial fits to the level-C bulk scattering database, and then compute the derivatives of those fits with respect to the input parameters (e.g., temperatures, mass concentrations, and number concentrations).

A discussion of the advantages and limitations of ZJU-AERO has been added, which can be found in Section 5 (Summary and Ongoing Tasks) of this paper:

*"Currently, ZJU-AERO is an efficient forward radar operator that has the advantages of*

*handling complexities of non-spheroid particle models. Therefore, it is a powerful research tool for studying the sensitivities of polarimetric radar observations with respect to the non-sphericity of hydrometeor particles. It also applies parallel acceleration techniques to boost its performance, allowing operational applications of data assimilation in NWP models employing single-moment (SM) microphysics (such as CMA-GFS / MESO). ZJU-AERO forward radar operator can be applied in data assimilation (DA) studies using indirect DA methods such as Bayesian approach (Caumont et al., 2010) and direct DA methods such as Ensemble Kalman Filter (EnKF), which both requires no tangent-linear (TL) or adjoint (AD) versions of the forward radar operator.*

*ZJU-AERO also has some limitations. For example, it currently cannot be applied in DA research based on the variational method. Simplification and linearization works are involved to obtain a TL / AD version of the forward operator. Moreover, PSD solvers for DM microphysics schemes have already been implemented in ZJU-AERO for the experimental CMA-MESO DM microphysics, but there are many validation and evaluation works to be done. Also, unlike the EMVORADO, which uses a distributed-memory parallel design and interface with COSMO-NWP model online, ZJU-AERO applies a shared-memory parallel design and the NWP model input should be stored in external files."*

2. In section 3.4, six SM microphysics schemes are used to explore the effects of different PSD schemes on the simulation of radar variables. It is not clear why the authors chose these six SM microphysics schemes. The shortcomings of the SM microphysics scheme are obvious in reproducing polarimetric features observed in convective storms and stratiform events. Although the SM microphysics scheme has been used for years and will continue to be used, the authors should clarify this point. Also, it is not clear that there are six schemes in Figure 11, while there are only five schemes in Figure 12 and three schemes in Figure 13. Based on the statistics in Figure 16, it seems that the Thompson or ThompsonTuned scheme has the best simulation results, and of course, the authors present only the results of the ThompsonTuned scheme in Figures 14 and 15. This is also clearly demonstrated in Figure 11. Therefore, is it necessary to evaluate other SM microphysics schemes?

Response: (1). Yes, it is necessary to clarify why we have chosen the six SM microphysics schemes for the sensitivity test in this study. Basically, we chose those schemes because they generally cover the natural variability of raindrop PSDs from continental precipitations (e.g., Wang2016) to maritime precipitations (e.g., Thompson 2008). Continental precipitations are characterized by having more large rain drops, while maritime precipitations have a larger population of drizzle-sized raindrops. We have added two paragraphs to address this issue in Section 3.4:

*"According to global aircraft in-situ measurements carried out by Abel and Boutle (2012), the six single-moment (SM) microphysics assumptions in Table 3 cover the natural variability of raindrop PSDs ranging from continental precipitations to maritime precipitations. Typical continental PSD with intensive large drops such as Wang et al. (2016) and typical maritime PSD such as Thompson et al. (2008) with a large population of drizzle-sized raindrops are both present. That is why we have chosen the six SM*

*microphysics schemes for the PSD sensitivity test.*

*It should be noted that although SM microphysics has been used for years and will continue to be used, it has obvious shortcomings in reproducing polarimetric features observed in convective and stratiform events. Since double-moment (DM) microphysics in CMA-MESO is still experimental, the consistent DM scheme in ZJU-AERO is still in development stage. However, the nature variability of raindrop PSDs for DM microphysics schemes is still within what is displayed in Figure 10, therefore it is safe to use those SM assumptions for a sensitivity test."*

(2). We agree that it is better to display the result of ThompsonTuned in Figure12. We added that to improve consistency:

[Figure]

As for Figure 13, we only displayed the results of the SM PSD scheme MP1948, Thompson2008, and Wang2016, because displaying all 12 lines of 6 the schemes would make the figure too busy to read. MP1948 represents a traditional SM PSD scheme, while Thompson2008 and Wang2016 are representative of typical maritime and continental precipitation PSDs, respectively. We added sentences in captions of Figure 13 to justify the reason for omitting PSD schemes:

*"Here, only the results of PSD schemes MP1948, Thompson2008 and Wang2016 are displayed, because MP1948 is a benchmark traditional PSD scheme, while Thompson2008 and Wang2016 are representative of typical maritime and continental precipitation PSD, respectively."*

(3). As for whether it is necessary to evaluate other SM microphysics schemes, we believe the answer is positive. However, since there is a large degree of freedom for SM microphysics schemes, it is difficult to find a globally optimized SM microphysics assumption for all precipitation scenarios (stratiform and convective). Therefore, although we have found one locally optimized SM scheme for a tropical cyclone case through sensitivity studies, there is still much work to be done.

3. The authors describe a forward operator that can simulate ground-based and space-borne polarimetric radar observations, but only show the observation and simulation of the Ku-band radar reflectivity in the case study. This is clearly insufficient for proving the reliability of the forward operator. Simulated polarimetric radar variables (such as differential reflectivity, specific differential phase, and Co-polar correlation coefficient) should also be shown.

Response: Thanks for the suggestion, we have conducted a case study to simulate ground-based polarimetric radar variables using our forward operator. The simulation results demonstrate that our forward radar operator works properly in simulating ground-based polarimetric variables and generally agrees well with previous radar simulators.

The results of the case study are displayed in Figure 2 and Figure 3 of this response letter, which can also be found in the user manual of the radar operator (released together with the software package).

From Figure 2 and Figure 3, it can be observed that the SM PSD scheme in the simulation tends to overestimate the large drops in this heavy precipitation event, resulting in larger simulated ZDR values (see Figure 2b) compared to the observed values in Figure 3b. It is known in the field of qualitative precipitation estimations (QPEs) that heavy precipitation events with tropical maritime characteristics tend to have more smaller drops and larger precipitation rates for a given reflectivity value (e.g., 50 dBZ). The low ZDR value (~1dB) accompanied by large reflectivity values (> 50dBZ) in the liquid phase indicates extremely large precipitation rates contributed by intensive small drops. This cannot be accurately described by a SM microphysics parameterization.

Overall, the simulation results produced by ZJU-AERO using the SM microphysics input for ground-based polarimetric variables are reasonable. However, more research is needed to accurately reproduce the melting layer features observed in Figure 2b and 2c.

[Figure]

Figure 2: The simulated polarimetric radar variables of a PPI snapshot (elevation angle=1.5 deg.) for a meso-scale convective system (MCS) recorded with heavy precipitation event in Henan Province, China. The observation was performed by CINRAD 98DP S-band Radar of Z9371 Zhengzhou site at 6:00UTC on July 20, 2021. All the radar operator simulations were conducted based on the T+6H forecast model grid data of WRF using the Thompson microphysics scheme, which was initialized at 00:00 UTC on July 20, 2021. We used default settings in ZJU-AERO (spheroid particle models and microphysics-consistent PSD solvers).

Considering the current length of this manuscript, we have added those simulation results in the user manual of the software (which was released together with the software package) instead of including them in the research paper. That is because the simulation results are relatively trivial compared to previous radar operators, and there are no significant scientific issues that require documentation:

We have added the following statement in the introduction of Section 4:

*"To demonstrate the reliability of the forward radar operator for ground-based polarimetric radars, we have also performed a case study of a meso-scale convective system (MCS), which can be found in the user manual (see Section Code Availability). The results are reasonable but relatively trivial compared to previous radar operators, so we will not display them in this section."*

[Figure]

Figure 3: The observed polarimetric radar variables of a PPI snapshot (elevation angle=1.5 deg.) for a meso-scale convective system (MCS) recorded with heavy precipitation event in Henan Province, China. The observation was performed by CINRAD 98DP S-band Radar of Z9371 Zhengzhou site at 6:00UTC on July 20, 2021.

4. There are some concerns about the value of introducing the Chebyshev-shaped model for raindrops. In Figure 13, it is shown that the ZDR simulated by the Chebyshev-shaped model and the spherical model in the Thompson scheme only show a pronounced difference when the liquid water content is larger, and the differences in the other radar variables are very small. Figure 16 shows that the Chebyshev-shaped model and the spherical model have the same simulation distributions on radar reflectivity. It's not clear if ZDR would have a different result. This goes back to the previous comment. In addition, whether there would be a difference between the Chebyshev-shaped model and the spherical model if the statistics were performed at different locations of the typhoon.

Response: (1). Thanks for the suggestion. Further discussions regarding the value of introducing Chebyshev-shaped model for raindrops are necessary.

Based on our findings, the deviations between the Chebyshev and Spheroid models (indicated as CmS) are prominent only for Ku-, Ka-, and W-band radars, and for large drops (Deq>5mm) at zenith and nadir viewing geometries (see Figure 8 in the manuscript).

To explore the sensitivity of spaceborne radar (Ku- and Ka-band) observed reflectivity with respect to the shape of raindrop model at extreme scenarios when large drops dominates, we have conducted observation system simulation experiments (OSSEs) (see Figure 4 - Figure 7).

We assumed an imaginary spaceborne radar GPM-DPR overpass for the MCS event discussed in Figure 3 and performed simulations for both Chebyshev and Spheroid raindrop particle model settings.

Figure 4 shows that assuming raindrops as Chebyshev-shape rather than spheroids can lead to a decrease in simulated reflectivity for more than 1 dB in heavy precipitation regions (Figure 4i). This is also true for Ka-band if no instrument sensitivity mask (12 dBZ) is applied, and the CmS reflectivity can be even larger (more than 2 dB, see Figure 5i, which is consistent with Figure 13a in the manuscript).

This decrease is due to the fact that the top of large Chebyshev raindrops tends to have much lower backscattering cross-section than Spheroids (Figure 8a in manuscript), and the extinction cross-section of large Chebyshev raindrops is slightly higher than spheroids (Figure 8b in manuscript). These two effects reinforce each other, resulting in lower simulated reflectivity for the Chebyshev experimental group.

[Figure]

Figure 4: Results of Ku-Band (13.6 GHz) OSSE raindrop particle model sensitivity test. The first row displays the simulated reflectivity assuming spheroid raindrops, while the second row displays those assuming Chebyshev-shape raindrops. The final row displays the Chebyshev – Spheroid (CmS) deviations of reflectivity. The four columns of panels display the simulation results at different altitudes, among them the first (0km) and second column (3km) is pure liquid phase. The case is the same MCS

event as that in Figure 1, but "forced" continental DSD scheme "Wang2016" is used to explore the extreme CmS sensitivity when large drops dominate.

[Figure]

Figure 5: Results of Ka-Band (35.5 GHz) OSSE raindrop particle model sensitivity test. Similar with Figure 4.

[Figure]

Figure 6: Similar with Figure 5 but no instrument sensitivity threshold (12dBZ) is applied to the simulation results.

[Figure]

Figure 7: Similar with Figure 5 but no attenuation effect is considered.

However, if the instrument sensitivity mask is applied for Ka-band (see Figure 5), then the observed reflectivity of liquid layer in heavy precipitation regions falls below the instrument sensitivity threshold and becomes invisible. This is because heavy precipitation regions tend to have much higher path-integrated attenuation (PIA), which makes the large negative values of CmS in the lower liquid phase (see Figure 5i) invisible. This does not mean that the CmS effect is not important for Ka-band; it can have significant effect for vertical pointing (VP) ground-based radars and airborne radars (looking from beneath or inside liquid clouds), for which the raindrops in heavy precipitation regions can be visible since the PIA can be much smaller.

In summary, the Chebyshev-shape raindrop model can have significant effects in certain scenarios, as observed in our OSSEs. However, the CmS effect shows little difference for the ThompsonTuned scheme (Figure 16 in the manuscript) because small drops dominate in this PSD scheme.

(2). Regarding the ZDR difference, it is only prominent for side-viewing geometry (elevation angle ~ 0 deg.), as the ZDR signatures of precipitation particles diminish for nadir and zenith viewing geometries. Therefore, we need to inspect the Look-Up Tables (LUTs) of Chebyshev-shaped raindrops at an elevation angles of 5 deg. and Ka-band.

Figure 8 shows that the ZDR difference between the Chebyshev-shape and spheroids at side-viewing geometry is small, only prominent for several narrow diameter bands when oscillation occurs. Consequently, for ground-based radars working at lower-frequency bands (S/ C / X-bands, figure not shown), the CmS effect for polarimetric variables is negligible.

[Figure]

Figure 8: The single scattering properties of Chebyshev shapes and Spheroid shapes at side viewing

geometry (for ground-based radars) at Ka-band (35.5 GHz).

This study is not only about presenting an alternative raindrop particle model in forward radar operator, but it also demonstrates the capability for ZJU-AERO to handle the complexities of non-spheroid particle models. Similar research will soon be applied to other types of hydrometeors, such as melting and solid particles. The potential for improvement when applying complex non-spheroid models for melting and solid particles should be much higher than for raindrops.

(3). Based on the above discussions, we expected that if statistics are performed in different regions of a tropical cyclone, the differences between Chebyshev-shape and Spheroid raindrop particle models tend to be minimal, because small drops dominate in this case. We have conducted sensitivity tests for different PSD schemes in different regions of the tropical cyclone (core region and outer spiral rainbands), as shown in Figure 9 - Figure 11. The core region is marked in Figure 9, while the outer precipitation regions are recognized as outer spiral rainbands. In both the core region and outer spiral rainbands of the tropical cyclone, the ThompsonTuned PSD scheme provided the best fit in Figure 10 and Figure 11.

[Figure]

Figure 9: Same as Figure 14 in manuscript, but region mask is added for tropical cyclones cores (the region within the red dash-dotted cycle). The out spiral rain bands of tropical cyclone are recognized as regions beyond the dash-dotted cycle.

Therefore, we have added these discussions to the final paragraph of Section 4.2:

*"...... An observation system simulation experiment of GPM-DPR overpassing a meso-scale convective system (recorded with extreme heavy precipitation events) reported CmS decrease effects of more than 1 dB at Ku-band and more than 2 dB at Ka-band (figures now shown). This sensitivity test is consistent with what we found in Figure12(a) and demonstrates the importance of introducing Chebyshev-shape raindrop model in certain scenarios (e.g., vertical pointing cloud radar, airborne radar and spaceborne radar working at high-frequency bands). As for polarimetric radar variables such as $Z_{DR}$ and $K_{DP}$ for ground-based radar at side-viewing geometry, the CmS effects are generally negligible."*

[Figure]

Figure 10: Similar with Figure 16 in manuscript, but only observations in tropical cyclone core (see Figure 9) are used for statistics.

[Figure]

Figure 11: Similar with Figure 16 in manuscript, but only observations in tropical cyclone outer spiral rain bands (see Figure 9) are used for statistics.

**Minor comment:**

1. Title: I suggest that the name of the operator would be used the words that more objectively, characterized the operator, instead of some subjective adjectives like "accurate and efficient".

Response: Thank you for your suggestion. After careful considerations, we have decided not to change the title. This software has already been used by several research groups in China, and we have other research papers currently under review using the same software name.

2. L17: which the Doppler variables are meant here?

   Response: Here, Doppler variables refer to radial velocity (VRAD) and spectrum width (WRAD).

   *"......, excluding the Doppler variables such as radial velocity and spectrum width."*

3. L125: Qh (hail) is included in Figure 2, but not here. Please be consistent.

   Response: Thank you for pointing that out. Hail has not been implemented in the current version, so Qh has been removed from Figure 2 in the manuscript to maintain consistency.

4. L195: "…undergone sufficient discussions in other works concerning radar operators, and we just inherited those settings and options from them", please add specific references.

   Response: Thank you for your feedback. We have added specific references to support this statement:

   *"In this paper, we do not elaborate on the algorithm details of certain issues, such as (a) trilinear interpolation, (b) sub-beam sampling and antenna pattern weighted averaging, and (c) ray-tracing trajectory solver. For trilinear interpolation, we follow the approach described in Wolfensberger and Berne (2018) to interpolate the model grid data to radar gates of each sub-trajectory gridpoint. Regarding sub-beam sampling and antenna pattern weighted averaging, we utilized Gauss-Hermite quadrature as outlined in Caumont et al. (2006). As the ray-tracing trajectory solver, we offer users both an offline beam trajectory solver (4/3RE) and an online beam trajectory solver (Zeng et al., 2014). All of these methods are reimplemented in Python, using either efficient numpy/scipy API or customized cython extensions."*

5. L229: Z and K are very critical variables, and their specific definitions should be given.

   Response: Thank you for your suggestions. We have provided specific definitions for matrix Z and K:

   *"Apart from the 2×2 complex amplitude scattering matrix S defined on complex electric field vector bases, one can define the 4×4 real matrix, known as the Mueller matrix, Z, and extinction matrix, K, which describe polarimetric light scattering and extinction properties of particles on the real Stokes vector bases, respectively. We kept the definitions of Z and K consistent with those mentioned in a study by Mishchenko (2014):*

$$\mathbf{I}^{sca}\left(r\hat{\mathbf{n}}^{\mathbf{sca}}\right) = \frac{1}{r^2}\mathbf{Z}\left(\hat{\mathbf{n}}^{\mathbf{sca}},\hat{\mathbf{n}}^{\mathbf{inc}}\right)\mathbf{I}^{inc}, \text{ (3)}$$

$$\mathbf{I}\left(r\hat{\mathbf{n}}^{inc}\right)\Delta S = \mathbf{I}^{inc}\Delta S - \mathbf{K}\left(\hat{\mathbf{n}}^{inc}\right)\mathbf{I}^{inc} + O(r^{-2}), \text{ (4)}$$

*Eq. (3) and (4) give the definitions of Mueller matrix Z and extinction matrix K, respectively. Here, $\mathbf{I}^{sca}$ and $\mathbf{I}^{inc}$ represent the Stokes vector $[I, Q, U, V]^T$ of scattering and incident light, while the unit vectors $\hat{\mathbf{n}}^{\mathbf{sca}}$ and $\hat{\mathbf{n}}^{\mathbf{inc}}$ indicate the unit vectors of directions of scattering and incident light, respectively. $r$ is the distance of detector from the particle,*

*while* $\Delta S$ *is the receiving surface of detector aligned normal to and centered on the straight line extending from the particle in the direction of the unit vector* $\hat{\mathbf{n}}^{\text{inc}}$. $O(r^{-2})$ *in Eq. (4) represents that the forward scattering term decreases by the inversed square law."*

6. Eq. (5): It is suggested that Zh and Zv be separated into two formulas. The plus and minus signs are very confusing here.

   Response: Done.

7. L398: "…incident radar beam and OZL This alignment sets the azimuthal…" should be "…incident radar beam and OZL. This alignment sets the azimuthal…"

   Response: Yes, a missing "." is added now.

8. L411: How to determine if a particle is axial symmetric in the program.

   Response: Basically, the axial symmetry is determined before calculating scattering properties database (LUTs) for a specific particle model, such as the Chebyshev-shape raindrop and hexagonal ice crystals.
   For example, the dielectric constant distribution of electromagnetic medium $\varepsilon(\theta,\varphi,r)$ in spherical coordinates is not relevant with the azimuth angle $\varphi$ for Chebyshev-shape raindrops, but it is relevant with azimuth angle $\varphi$ for hexagonal ice crystals, which only have 6-fold azimuthal symmetry. Consequently, we can conclude that Chebyshev-shaped raindrops have axial symmetry while hexagonal ice crystals do not, based on mathematical expressions rather than a computer program.
   We added specifications to make the statement clearer:
   *"……For both axially symmetric (i.e., the dielectric constant distribution of electromagnetic medium in spherical coordinates $\varepsilon(r,\theta,\varphi)$ is irrelevant with azimuth angle $\varphi$)……"*

9. L456: "It is apprarent" should be "It is apparent"?

   Response: Yes, typo is fixed now.

10. L564: Why use a constant air density instead of diagnostic air density. Air density decreases with increasing altitude, it also changes with variations in pressure, temperature, and humidity.

    Response: Yes, we did not specify the point clearly here: the constant air density is only used in determine the position of $N_0$-$\lambda$ curve of Figure 11 in the manuscript. That is because if diagnostic air density is considered, then we cannot determine the $N_0$-$\Lambda$ curve in Figure 11 (the position of $N_0$-$\Lambda$ curve depends on $Q_{R0}$ expressed as mass concentration rather than mixing ratio). That means constant air density is only used in qualitative diagnose of $N_0$-$\Lambda$ curve plot, while in real PSD solver of ZJU-AERO, the diagnostic air density is used.
    Notes have been added to the caption of Table 3 in the manuscript:
    *"……It should be noted that the constant air density is only used here to qualitatively determine the position of the Thompson2008 and ThompsonTuned N0-$\Lambda$ curves in Figure 11. Diagnostic air density is used in actual PSD solver of ZJU-AERO."*

11. L684: Why does not it show the temperature profile? There is no temperature profile how to determine the altitude of freezing level.

    Response: Yes, thanks for the suggestion. We have added Isothermal lines

of model background state to each panel of Figure 15 to show the temperature profile. The red isothermal line at 0℃ indicates the freezing level.

[Figure]

GPM-DPR Swath Cross Section Reflectivity between A and B (Observation and Simulation)

Add in description in caption of Figure 15:

*"…… The temperature of model background state is indicated by isothermal lines in each panel. Numbers of the contour labels have the unit of Celsius degree."*

12. L686: It is difficult to see the features of the melting layer in Figure 15a and b.

Response: We acknowledged that the features of melting layer are weak at Ku-band and not prominent at Ka-band for this case, where the precipitation system is not so horizontally homogeneous. We updated the descriptions in the manuscript accordingly:

*"Actually, the GPM-DPR observations revealed a weak bright band (BB) signature below the freezing level, which can be recognized at Ku-band (with an average reflectivity enhancement of approximately 3 dB, see Figure 14a) but not so clear at Ka-band (see Figure 14b)."*

13. L690-692: This may be due to the operator not considering melting or mixed-phased particles, and the shortcoming of the microphysics scheme.

Response: Yes, the reason why ZJU-AERO cannot simulate melting layer signatures is rearranged as follows:

*"However, the simulations we conducted did not show an obvious BB, which was attributed to not considering melting or mixed-phased particles, and the shortcoming of the microphysics scheme."*

14. Figure 15: It is suggested to add temperature profile.

Response: We have added the temperature profile in Figure 15 by including isothermal lines (see response to minor comment 11).

15. The SUMMARY is not sufficient.

Response: Thanks for the suggestions. We have expanded the summary

section and make it complete:

*"In summary, Section 2 of our study introduced the basic concepts of design and formulations in the ZJU-AERO. These concepts included the general procedure of the software, radar variable calculations, and the available hydrometeor settings (shape parameters, dielectric constant models, canting angles, particle size distributions). We also derived the formulation of polarimetric radar variables starting from single particle back-scattering Muller matrix Z and extinction matrix K.*

*In Section 3, we highlighted the unique features of ZJU-AERO, specifically its multi-layered design for the optical database of non-spherical particles. We demonstrated this by visualizing the scattering properties using the example of the Chebyshev-shape raindrop particle model, comparing it to the properties of traditional spheroid raindrops. We conducted LUT demonstrations for two database layers: Level-A (raw single scattering properties database), and Level-C (bulk scattering property database). We also introduced a new intermediate quantity named as "backscatagrand" to diagnose the PSD integrations of optical properties. We concluded that the Chebyshev-shape raindrop model shows noticeable differences of bulk scattering properties (compared to spheroid model) only at zenith and nadir viewing geometries and for milimeter-wavelength radar bands. This difference is more prominent for continental DSD models (e.g., Wang2016) with larger drops. These deviations can reach up to 2-3 dB at Ka-band for spaceborne radars in heavy continental precipitation regions where large drops dominate. Given the lower uncertainties in simulating reflectivity of liquid phase compared to solid and melting phases, such a difference deserves attention in specific applications such as comparing data from ground-based and spaceborne radar observations (Warren et al., 2018).*

*Furthermore, in Section 4, we validated the simulation reliability and capability of ZJU-AERO by analyzing a case study of a tropical cyclone using input from the CMA-MESO for simulating spaceborne radar observations. We found that ZJU-AERO provides reasonable simulation results, except for the bright band signature at melting layer, which can be attributed to the current version of ZJU-AERO, not considering melting or mixed-phased particles. We also performed sensitivity assessments of PSDs and morphology options in ZJU-AERO and found that the ThompsonTuned single-moment PSD scheme provides the best fit of the reflectivity histogram in the simulation to the GPM-DPR observation. However, using either the Chebyshev-shaped raindrop particle model or the spheroid model makes little difference to the simulation results since the tropical cyclone precipitation has a maritime DSD, where small drops dominate."*

---

## Author Comment (AC2)

**Reviewer # 2**

**Summary:**

The paper presents the radar forward operator ZJU-AERO that is capable of simulating radar observables for both ground-based and air-/spaceborne sensors and of considering effects of non-spherical, non-homogeneous hydrometeors including polarimetry, designed particularly for use along with the Chinese Meteorology Administration's numerical weather prediction models. A larger fraction of the paper deals with evaluation of the Chebyshev shape model for rain suggested by Chuang and Beard (1990) compared to more classical spheroid models. In general, this topic is suitable for publication in GMD.

However, the paper "oscillates" between some very basic, textbook-like descriptions entangled with some quite specific, not always relevant details, but lacking clear direction and clarity and detail at places where it would start to be really interesting. I suggest publication to be considered after major revisions.

Please find below my detailed review report.

Response: Thank you for your detailed review report. We appreciate your feedback and have taken your comments into consideration for improving the manuscript.

(1). We acknowledge the presence of basic, textbook-like descriptions in the paper. This was intentional to ensure clarity and consistency in the use of symbols, especially considering potential conflicts in conventions found in popular meteorological radar textbooks.

(2). We have made significant improvements to enhance the direction, clarity and level of detail in the paper. We have addressed specific areas where more details were needed to make the content more engaging and informative.

For a more detailed response to each of your comments, please refer to the term-by-term responses provided below.

**General remarks:**

The manuscript, in my view, presents a mixture of on the one hand quite basic, but very detailed content and on the other hand seemingly more novel, but hurried over, too superficial content. It seems to me like an extract from a thesis, with individual parts not stitched together very well for providing a consistent and concise work with a clear common thread. It contains a number of rather fancy illustrations that, however, do not necessarily provide relevant information, and that often are not covered with appropriate detail in text, i.e. that rather seem to be there to look nice than to make a relevant contribution (e.g. Figs1, 4, 5, 6).

It often remains unclear what has been done by the authors, what has been by other work, and what has been taken over quite directly from other works (general approaches, more specific methods, algorithms, possibly even code, …), and how particularly the latter has been incorporated into this work. For example, L139 refers to Zeng16 for sub-beam sampling and averaging methods in submodules B1 and B2 in a way that seems to imply Zeng16 describes a

part of ZJU-AERO ("see Zeng et al. (2016) for more about"); however, Zeng16 describes another, independent radar forward operator (RFO) and it remains unclear whether the reference is "just" to a general approach or to specific formulas and their implementation or to the taking over of entire code bits. Similar for the Wolfensberger and Berne (2018) reference in L146. Also, L309 "To address these uncertainties, a field research study was conducted" also at best leaves it unclear whether the study was done by the authors (that's what this formulation suggests to me) or others. So please be more clear in your formulations!

Response: (1). We have made structural adjustments to the paper, including moving the beam trajectory solver of the spaceborne radar to Appendix A, in order to improve compactness and clarity.
(2). We have made efforts to integrate the figures better with the content of the study.
(3). We have modified our formulations to make it clearer what contributions were made by our work and what was taken from other sources.
*"In this paper, we do not elaborate on the algorithm details of certain issues, such as (a) trilinear interpolation, (b) sub-beam sampling and antenna pattern weighted averaging, and (c) ray-tracing trajectory solver. For trilinear interpolation, we follow the approach described in Wolfensberger and Berne (2018) to interpolate the model grid data to radar gates of each sub-trajectory gridpoint. Regarding sub-beam sampling and antenna pattern weighted averaging, we utilized Gauss-Hermite quadrature as outlined in Caumont et al. (2006). As the ray-tracing trajectory solver, we offer users both an offline beam trajectory solver (4/3RE) and an online beam trajectory solver (Zeng et al., 2014). All of these methods are reimplemented in Python, using either efficient numpy/scipy API or customized Cython extensions."*
*"We applied the sub-beam sampling and averaging methods as described in Zeng et al. (2016)."*
*"we adopted the approach described in Appendix A of Wolfensberger and Berne (2018) and reimplemented it as a Cython extension"*
*"To address these uncertainties, a field research study conducted by Garrett et al. (2012) used an in-situ observation instrument called the multi-angle snowflake camera (MASC)."*

**Improve presentation style:**
- First present results (describe them), then discuss / explain them, then conclude.

Response: OK. We have removed any concluding statements before the discussion to ensure a better flow.

- Figure contents, when shown, need to get presented and discussed, otherwise leave them out (e.g. polarimetric variables).

Response: Understood. We have removed the BSP polarimetric results in Figure 12 and the PSD results of the melting layer and solid layer in Figure 15. However, we have retained the SSP polarimetric results and discussed why they are not significant.

- Equations should always be integrated in text, i.e. be part of a sentence that makes sense when being read out loud, not stand alone and/or a few lines before or after they are introduced.

Response: Noted. We have integrated every Equation into the text so that they are part of a sentence.

Introduce math symbols at first use. Use unambiguous notation, e.g. avoid to use λ for both radar wavelength and PSD parameter, Dmax for maximum diameter (or max dimension?) and upper PSD integration limit, …

Response: We have added introductions to every math symbol at its first use. We now use λ for radar wavelength and Λ for PSD parameter to avoid confusion. We use $D_{max}$ for maximum dimension and $D_u$ for the upper truncation limit of size integration.

- Introduce acronyms at first use. Then use them, without re-introducing.

Response: We have checked the use of acronyms and avoided re-introducing them.

- Figure captions here are generally overlong with partly redundant or irrelevant content. Figure caption are supposed to describe what is seen in the figure. All details on data processing, implementations, setups and the like should instead be in the text.

Response: We have revised the figure captions (Figure 13-15) to focus on describing the content of the figures rather than providing details on the post-processing information and RFO setup information.

**Improve language:**
- Spell-check.

Response: Checked.

- Use tenses consistently throughout the manuscript. For things done in this study and presented in this paper: either present or past; I suggest present tense.

Response: Fixed. Tenses has been examined throughout the paper.

- Be careful with fill words (e.g. "however", "nevertheless", "fortunately" (?!?), "It is worth to note/mentioning", "obviously", …) that inappropriately (unintentionally?) relativize or judge.

Response: Fixed.

- e.g. L108: what is an "approximately(!) 4/3Re-radius curve"? L110 "rarely" rather "weakly" or "negligible", …

Response: Fixed.

**Specific remarks:**

1. Abstract: It introduces ZJU-AERO as polarimetric RFO with one main objective be in the assimilation of ground-based radar polarimetric variables. The remainder of the manuscript, however, has little on polarimetry – it introduces/defines the variables, presents some calculations, but without any relevant discussion of them.

    Response: (1). We have added an analysis of SSP factors of polarimetric radar variables to address the lack of discussion on polarimetry (see response to remark 56). That explains why we do not expand on the CmS effects of polarimetric radar intrinsic variables for ground-based radars: the effect of introducing Chebyshev shape is not significant for ground-based radar frequencies and viewing angles (see

response to remark 72).

(2). The calculations of polarimetric variables of ground-based radar can be found in user manuals (Please refer to the response to Reviewer 1 / Major comment 3). The user manual was released together with the software package, focusing on technical issues and validations. The simulation results are trivial compared with previous radar operators with few scientific issues need documentation. We hope this paper to focus on scientific issues, and there is only an interesting raindrop shape (Chebyshev shape raindrop) need scientific validation, but introducing it has little effect on ground-based radar observations.

(3). Yes, the capability to simulate ground-based polarimetric radar variables is a by-product. At first, we want to develop a polarimetric RFO for spaceborne radar FY3G/PMR. Then we make ZJU-AERO capable of simulating ground-based radar.

We modified the abstract to focus not so much on ground-based radar (It can simulate, but not so many validations have been performed):

*"The main objective of developing this observation operator was to assimilate observations from the Precipitation Measurement Radar (PMR). It is also capable of simulating ground-based radar's polarimetric radar variables, excluding the Doppler variables such as radial velocity and spectrum width."*

2.  Introduction: It might be worth mentioning Zhang21 in the state-of-the-art review of RFOs, too, as it targets fast applications like data assimilation and contains a melting scheme module.

    Response: Done.

    *"Moreover, the fast parameterized forward radar operator developed by Zhang et al. (2021) contains a melting scheme module, targeting data assimilation purposes."*

3.  L48f: "Zeng et al. (2016) […] but it focuses on non-polarimetric variables." – Recent versions of the Zeng et al. (2016) RFO EMVORADO also cover simulations of polarimetric radar variables (see Trömel21, Shrestha22).

    Response: References and reviews of those studies are added in the state-of-the-art review of RFOs.

    *However, Later developments on EMVORADO have enabled its capability on simulating dual-polarization variables and conducted sufficient evaluations (Trömel et al., 2021; Shrestha et al., 2022).*

4.  L131ff: As this covers two distinct submodules, B1 and B2, make clear in the text which descriptions belong to which submodule.

    Response: Done.

    *"For ground-based radar (B1), the temperature and humidity profiles above the radar site are used to solve the trajectory online. ……*

    *Nevertheless, for space-borne radar (B2), the beam trajectory is computed using the geometry shown in Figure 3, and no sub-trajectory averaging is supported at this time."*

5.  L147ff: Where is the input to module (D) taken from? User set? Model driven?

    Response: Since ZJU-AERO is an offline RFO (i.e., it loads model grid

data from external storage), there is no true "model driven" hydrometeor parameters in ZJU-AERO, but only "model consistent" hydrometeor parameters.

Specifically, there is an ad-hoc module called "const" in ZJU-AERO, loading constant parameters of hydrometeors. We have internal presets in constant module to keep RFO consistent with microphysics (only Thomspon and WSM6 currently). However, users can also modify those presets and apply "forced" hydrometeor parameters for research purposes (see Table 1).

*"The hydrometeor submodule (D) specifies the properties of hydrometeors in each radar gate along each trajectory, usually by loading presets of microphysics-consistent constants of hydrometeors. This includes the orientation preference, probability distribution of particle morphology, particle size distribution (PSD), and other parameters. Those presets of hydrometeor properties can also be modified by users to perform "forced" (i.e., inconsistent with microphysics) simulations for research purposes."*

6. L149f: "regular morphology [...] an analytical expression can then be used to retrieve the PSD" – Too specific at this point in the manuscript, where it is everything but obvious what particle morphology has to do with PSD.

    Response: Yes, we modified the description here from specific contents to more general ones.

    *"The PSD parameters of hydrometeors are solved in this step, from prognostic bulk hydrometeor variables in NWP models (mass concentrations and number concentrations). For more details, see Section 2.3."*

7. L153ff: Which module does this? Language/grammar of this paragraph needs care.

    Response: (1). The PSD integration of SSP can be conducted either online (research mode) or offline (operational mode). For the former case, this step is carried out by the core module (E). While for the later case, this step is done by external tool scripts when generating bulk scattering database. We have updated the flow chart and description text to make it clearer.

    (2). Several typos were fixed.

8. L158: Does this module have a letter-name? "calculates intrinsic polarimetric radar variables on each radar gate" – Wouldn't that rather be on each subbeam-gridpoint within each radar gate, or where is this done?

    Response: (1). Yes, it is the core module (E).

    (2). Yes, precisely. It calculates intrinsic polarimetric radar variables on each sub-trajectory gridpoint within each radar gate.

9. L153ff: Do numbered items 5.-7. all belong to module (E)? If so, better make them subitems 5.1-5.3?

    Response: (1). Yes, they all belong to module (E).

    (2). Good suggestion! We made them subitems 5.1-5.3.

10. Fig2: It is unclear, how module (E) links to the multi-level database (and remains so throughout the manuscript): Is the DB external and can other DB be used, too (e.g. the

ARTS-SSDB by Eriksson18)? What data would they need to provide? Does module (E) get LevelB data? Why cover LevelC here then? Where does LevelA->B conversion gets its settings/assumptions on orientation preference and shape parameters from? Is that hard-coded?

Response:

(1). Yes, the DB is external.

(2). Other DB can also be used, but need extra format conversion works.

(3). They need to provide backward and forward amplitude scattering matrix for various scattering geometries.

(4). There are two modes (operational mode, use level C DB and research mode, use level B DB), see modified text below.

(5). Conversion scripts obtain assumptions from user-friendly alterable self-explanatory yaml files, not hard-coded.

[Figure]

*Figure 2: A conceptual flow chart of ZJU-AERO. The parallelogram boxes represent input data (including numerical weather prediction (NWP) output, radar specifications, and optical properties look-up tables). The green round rectangles indicate the names of the key submodules of ZJU-AERO. The yellow dashed boxes indicate the key data structure used in the simulations. The diamonds represent the points at which crucial "if-else" judgements can be conducted during processing.* Those dashed arrows in this flowchart represent external database generation steps carried out using released tool scripts of ZJU-AERO.

*"5.1 The bulk scattering properties of particles in each radar gate are computed by integrating the single scattering properties across the size spectrum of each hydrometeor type and summing over hydrometeor types,* which can be conducted either online (E1, research mode) or offline (E2, operational model) in ZJU-AERO. The scattering properties LUTs of ZJU-AERO are consulted in this step, which is composed of three levels (Level A, Level B, and Level C). The multi-layered architecture will be introduced in detail in Section 3.1. The research mode is more flexible since it accesses the Level B database for single scattering property, while the operational mode is more efficient by accessing the Level C database for bulk scattering properties straightforward. We provide users with scripts for level A to B and Level B to C conversions (integration parameters alterable in YAML configure files)."*

11. L173ff & Fig3: This does not seem particularly relevant. If relevant, give it its own subsection at an appropriate place in the manuscript. Under "Flow Chart and Concepts" it definitely seems out of place and far too specific.

   Response: Yes. We moved those specific contents elaborating on spaceborne trajectory module to appendix A.

   *"For more details on beam trajectory module of space-borne radar (B2), please refer to Appendix A."*

12. L195ff: "Since those topics have already undergone sufficient discussions in other works concerning radar operators, and we just inherited those settings and options from them. Readers with an interest in these options may refer to the bibliography mentioned in the ZJU-AERO general workflow description text." – That's far too general and not appropriate for a scientific publication as this does not facilitate reproducibility of your research. You need to reference clearly, which parts come from which reference and specify what "inherit" means in each case (general approach, specific methods, implementations, ...).

   Response: Yes. We specified the reference of those methods for better reproducibility.

   *"In this paper, we do not elaborate on the algorithm details of certain issues, such as (a) trilinear interpolation, (b) sub-beam sampling and antenna pattern weighted averaging, and (c) ray-tracing trajectory solver. For trilinear interpolation, we follow the approach described in Wolfensberger and Berne (2018) to interpolate the model grid data to radar gates of each sub-trajectory gridpoint. Regarding sub-beam sampling and antenna pattern weighted averaging, we utilized Gauss-Hermite quadrature as outlined in Caumont et al. (2006). As the ray-tracing trajectory solver, we offer users both an offline beam trajectory solver (4/3RE) and an online beam trajectory solver (Zeng et al., 2014). All of these methods are reimplemented in Python, using either efficient numpy/scipy API or customized Cython extensions."*

13. Sec2.2: Most of the time I appreciate when papers provide sufficient and clear theoretical background. So I have ambiguous feelings about criticizing this section. However, it seems a bit too basic and too drawn out textbook knowledge. Beside that, it's incomplete: What is L (L209), what are the h, v, k (L215)? It jumps from theoretical background to very specific ZJU-AERO things (database Level A/B units). Figure 4 does not (at least not to

me) "explain why the BSA convention should be introduced" – as the reader, I don't care whether it "should be introduced" – just that you introduce it (if it is relevant).

Response: (1) We understand your concern about the level of detail provided in the theoretical background. As this study is the first potential publication of radar operator ZJU-AERO, we felt it was important to provide clear definitions of symbols and conventions used in this study, especially considering the potential conflicts in conventions found in popular textbooks of meteorology radar.

(2) We have added definitions for symbols such as L (symbol of length in dimensional analysis) and h, v, k (unit vectors for horizontal-polarization, vertical-polarization, and propagation direction of electromagnetic wave.)

(3) We agree that mentioning the database levels is inappropriate in this section. We have removed that information and focused on explaining the commonly used units of variables.

(4) Regarding Figure 4 (Figure 3 in the revised manuscript), we have revised the text accordingly.

*"Figure 3 displays the differences between the Back Scattering Alignment (BSA) convention and the FSA convention (Chandrasekar, 2001)."*

14. L227: "we also have the 4x4 real matrix" – formulation. "one can define"? what are the actual properties of S and of Z and K, what's the difference, what are their respective roles/uses? what is "the Stokes vector space"?

Response: We have made the following revisions based on your feedback:

(1). We agree that "one can define" is a better formulation.

(2). We have added definitions for Z and K, explaining their differences and respective roles/uses.

(3). We have clarified that the amplitude matrix S is defined on complex electric field vector bases, while the Mueller and extinction matrices are defined on real Stokes vector bases.

*"Apart from the 2×2 complex amplitude scattering matrix S defined on complex electric field vector bases, one can define the 4×4 real matrix, known as the Mueller matrix, Z, and extinction matrix, K, which describe polarimetric light scattering and extinction properties of particles on the real Stokes vector bases, respectively. We kept the definitions of Z and K consistent with those mentioned in a study by Mishchenko (2014):*

$$\mathbf{I}^{sca}\left(r\hat{\mathbf{n}}^{sca}\right) = \frac{1}{r^2}\mathbf{Z}\left(\hat{\mathbf{n}}^{sca},\hat{\mathbf{n}}^{inc}\right)\mathbf{I}^{inc}, \quad (3)$$

$$\mathbf{I}\left(r\hat{\mathbf{n}}^{inc}\right)\Delta S = \mathbf{I}^{inc}\Delta S - \mathbf{K}\left(\hat{\mathbf{n}}^{inc}\right)\mathbf{I}^{inc} + O(r^{-2}), \quad (4)$$

*Eq. (3) and (4) give the definitions of Mueller matrix Z and extinction matrix K, respectively. Here, $\mathbf{I}^{sca}$ and $\mathbf{I}^{inc}$ represent the Stokes vector $[I, Q, U, V]^T$ of scattering and incident light, while the unit vectors $\hat{\mathbf{n}}^{sca}$ and $\hat{\mathbf{n}}^{inc}$ indicate the unit*

*vectors of directions of scattering and incident light, respectively.  $r$  is the distance of detector from the particle, while  $\Delta S$  is the receiving surface of detector aligned normal to and centered on the straight line extending from the particle in the direction of the unit vector  $\hat{\mathbf{n}}^{\mathbf{inc}}$.  $O(r^{-2})$  in Eq. (4) represents that the forward scattering term decreases by the inversed square law.*"

15. L238: This paragraph doesn't make sense to me. Maybe a language issue? What's the role of the optical theorem here? "This is because" – what exactly is because of what?

   Response: We find references about optical theorem is not clear here. We modified the presentation to make the logic clear:

   *"Note that we can only apply the particle ensemble integration over the complex amplitude scattering matrix, S, in the forward scattering geometry. This is because integrating over extinction matrix elements is proportional to integrating corresponding forward amplitude scattering matrix elements."*

16. L243ff: I find conversion formulas S -> Z,K actually more relevant than providing radar variable formulas in terms of S.

   Response: (1). Conversion formulas S -> Z, K can be found in particle light scattering textbooks such as Mishchenko (2014). We have provided references and stated in the text that we keep the same convention with it. It seems unnecessary to copy so many equations in the paper.

   (2). We provide radar variable formulas in terms of Z and K, because they have the same dimensions ($L^2$), and we can directly perform ensemble averaging on them (not always the case with S).

   *"Therefore, we use Mueller and extinction matrices to represent radar variables because ensemble means can be performed directly on them, as is not the case for amplitude matrix. Also, they have a unified dimension of $L^2$."*

17. L265: What is the tilded rhohv? "is the magnitude of [...] whose amplitude is" – i might be off, but in my understanding the amplitude is the magnitude of a complex number. So why would they be two different variables (tilded rhohv and deltahv)?

   Response: (1). Tilded rhohv is the complex correlation coefficient.

   (2). It is a typo here. Fixed.

   Rhohv is the amplitude of complex correlation coefficient, while deltahv is the phase of the complex correlation coefficient.

18. L279: What's the difference between "reflectivity factor" and "reflectivity"? The path attenuation?

   Response: (1). There is no difference here. In this study, we distinguish whether path attenuation is applied by "measured / observed reflectivity" and "intrinsic reflectivity".

   (2). In many meteorology radar literatures, the "reflectivity factor" and "reflectivity" are used interchangeably, which both refer to horizontal polarization reflectivity $Z_H$ in dB-scale. But here we'd better call it "reflectivity" to avoid readers' confusion.

19. Tab1: Shape distribution information is missing here (in text, Wolfensberger and Berne (2018) is given, ie Garrett et al. (2015) can't contain this).

Response: Yes, Garrett et al. (2015) did the measurement work (field research), while Wolfensberger and Berne (2018) fitted the data as gamma distributions. Therefore, both references are added here.

20. L316: "in which y is the reciprocal of the aspect ratio" – Define what aspect ratio is for you. There are different definitions around. Compare, e.g., Ryzhkov's (minor axis / major axis, i.e. AR<=1) or Mishchenko (rotational axis / symmetry-plane axis, ie AR<1 for oblates, AR>1 for prolates).

Response: We use the Ryzhkov's definitions.

*", in which gamma is the reciprocal of the aspect ratio (minor axis / major axis, always less than unity)."*

21. L321: "The mass–diameter relationship is crucial in determining the PSD" – why?

Response: Because solving PSD parameters from NWP prognostic hydrometeor mass concentrations (for bulk microphysics) requires knowledge of mass of hydrometeor particles for particle of different sizes.

*"Since solving PSD parameters from NWP prognostic hydrometeor mass concentrations (for bulk microphysics) requires knowledge of the mass of various-sized particles, the mass–diameter relationship is crucial in determining the PSD (see the column of mass-diameter relationship in Table 1)."*

22. Sec2.3: For rain, is there also a shape distribution assumed?

Response: In ZJU-AERO, the aspect ratio of rain is a fixed value (i.e., a delta-function shape distribution) for a given diameter.

That is because the aspect ratio of rain has less natural variability than solid precipitation particles, for a given diameter.

23. L336: "it seems that a microphysics-consistent mass-diameter relation [...]Dmax³ for snow and graupel" – it seems??? how would Dmax³ be microphysics-consistent? What is rho_sp?

Response: (1). For many microphysics schemes, the solid precipitation particles are represented as an air-ice mixture spheres of different overall densities. The overall density of solid hydrometeor particle is "rho_sp"  (smaller than rho_ice, e.g. 100kg/m³ for snow and 500kg/m³ for graupel in WSM6). Therefore, the mass of solid particles of

maximum dimension $D_{max}$ can be calculated as $\bar{m}\left(D_{max}\right) = \frac{\pi}{6} \rho_{sp} D_{max}{}^3$.

Such a m-$D_{max}$ relationship is therefore consistent with the WSM6 microphysics.

(2). If RFOs such as ZJU-AERO use solid particle shape parameters from *in-situ* measurements (e.g. aspect ratio distribution measurements from Garrett et al. (2015)), but still use the above microphysics consistent m-$D_{max}$ relationships, then inconsistency arises:

The mass of solid hydrometeor particles is the mass of spheroid represented by RFO, not the sphere represented by NWP microphysics.

*"Again, there is a microphysics-consistent mass–diameter relationship*

$$\overline{m}\left(D_{\max}\right) = \frac{\pi}{6}\rho_{sp}D_{\max}{}^{3} \quad (\rho_{sp} \text{ is the overall density of the sphere solid precipitation}$$

*particle) for snow and graupel. Many microphysics schemes, such as WSM6, simply treated solid precipitation categories as spheres with different ice–air mixture ratios and hence different overall densities $\rho_{sp}$.*"

24. Eq22: That doesn't make sense to me. In this notation F(λ ) will be 0. What are F and F' anyways?

Response: Here we need to find the root of equation (for unknown parameter $\Lambda$):

$$Q_x = \sum_{D_i=D_l}^{D_i=D_u} \overline{m}\left(D_i\right)N_0 e^{-\Lambda D_i}\Delta D$$

There is no analytical way to solve this equation, so we use the Newton-iteration method. We define a function for Newton iteration method:

$$F(\Lambda) = \sum_{D_i=D_l}^{D_i=D_u} \overline{m}\left(D_i\right)N_0 e^{-\Lambda D_i}\Delta D - Q_x$$

$F(\Lambda)$ equals zero only when $\Lambda$ is the root of the equation. $F'(\Lambda)$ is the derivative of function $F(\Lambda)$.
We first have an initial guess of $\Lambda_0$. Then (n+1)-th guess $\Lambda_{n+1}$ can be derived from n-th guess $\Lambda_n$ by the following iteration equation:

$$\Lambda_{n+1} = \Lambda_n - \frac{F(\Lambda_n)}{F'(\Lambda_n)}$$

We added definitions of $F(\Lambda)$ and $F'(\Lambda)$ in the text, to make it clear for readers.

25. L355: What is V, where do you get that come from?

Response: V is the volume of occupied by irregular-shaped hydrometeors. It is calculated simply from geometries and mathematics formulas.

*"The mass $m(\gamma;D_i)$ for each morphology specifications can be calculated as the density of the pure ice $\rho_{ice}$ multiplied by the volume occupied by the non-spherical hydrometeor particle model $V(\gamma;D_i)$ (calculated from mathematics formulas of geometrical bodies)."*

26. Sec3: There is far too little information on how the database is constructed, e.g. how many/which grid points are used for each of the dimensions, what are the integration limits etc. And how that has been decided. And how much impact that has on the final radar variables.

Response: We have addressed this issue by adding details of database in Appendix B.

*"The grids of LUT dimensions are presented in Table B1. Only the Z and K matrix*

*elements that appear in the formulas of polarimetric radar variables in Eq. (7)-(14) are stored in the database. For temperatures, we set the minimum temperature for super-cooled liquid particles as 233 K, considering that homogeneous freezing starts at lower temperatures. For solid hydrometeors, the lowest temperature in the database is 203K, and lower environment temperatures encountered are taken as 203K. We provided LUTs for 6 bands that are widely used for ground-based and spaceborne weather radars. The ranges of diameters and aspect ratios for solid hydrometeors are suggested by observations of field-research (Garrett et al., 2015), while the range of diameters and aspect ratios of liquid hydrometeor have already been discussed in Section 3.3. The number of diameter bins is sufficient to produce a reasonable size spectrum of hydrometeors for various hydrometeor mass concentrations, as shown in Figure 11. The range of mass concentration grids for BSP LUTs is the same with that of Geer et al. (2021). Other external LUTs, such as Eriksson et al. (2018), need to undergo format conversions and interpolations (e.g., for different grids of diameter) before they can be applied in ZJU-AERO.*

**Table B1: Grid specifications for Look-Up Table (LUT) dimensions in ZJU-AERO. The format "[START:A, END:B, STEP:C]" represents a sequence of numbers ranging from A to B with an increment of C. The format "LIN[MIN:A, MAX:B, NUMBER:C]" denotes a sequence of evenly-spaced numbers on a linear scale ranging from A to B with a total number of C values, while "LOG[MIN:A, MAX:B, NUMBER:C]" denotes a sequence of evenly-spaced numbers on a log scale.**

| Dimensions | Grids |
|---|---|
| Matrix Elements | Mueller Matrix $\mathbf{Z}$ : $Z_{11}$, $Z_{12}$, $Z_{21}$, $Z_{22}$, $Z_{33}$, $Z_{34}$, $Z_{43}$, $Z_{44}$ |
| | Extinction Matrix $\mathbf{K}$ : $K_{11}$, $K_{12}$, $K_{34}$ |
| Temperatures [K] | Solid Hydrometeor (Snow, Graupel): 203, 213, 223, 233, 243, 253, 263, 273 |
| | Liquid Hydrometeor (Rain): 233, 238, 243, 248, 253, 258, 263, 268, 273, 278, 283, 288, 293, 298, 303, 308, 313, 318 |
| Frequencies [GHz] | 2.7 [S], 5.6 [C], 9.41 [X], 13.6 [Ku], 35.6 [Ka], 94.1 [W] |
| Mass Concentrations [kg·m-3] | LOG[MIN: $10^{-6}$, MAX: $10^{-2}$, NUMBER: 161] |
| Diameters [mm] | Snow ($D_{max}$): LIN[MIN: 0.2, MAX: 20, NUMBER: 128] |
| | Graupel ($D_{max}$): LIN[MIN: 0.2, MAX: 15, NUMBER: 128] |
| | Rain ($D_{eq}$): LIN[MIN: 0.1, MAX: 9, NUMBER: 128] |

| Elevations [°] | [START: -90, END: 90, STEP: 1] | | |
|---|---|---|---|
| Beta Angles [°] | [START: 0, END: 180, STEP: 1] | | |
| Reciprocal of Aspect Ratio [-] | Snow: [START: 1.1, END: 5.1, STEP: 0.2] | | |
| | Graupel: [START: 1.1, END: 3.1, STEP: 0.1] | | |
| | Rain: Single Value (see Figure 6) | | |

*To alleviate the truncation and quadrature error in particle size integration, a renormalization technique is applied after the particle number in each size bin is calculated. We first calculate the renormalization factor $\kappa$ :*

$$\kappa = \frac{\sum_{i=1}^{nbins} m(D_i)N(D_i)\Delta D}{Q_m} ,$$
(B1)

*Here, $m(D_i)$ is the mass of hydrometeor particle with a diameter of $D_i$, $N(D_i)$ is the particle number in that size-bin, while $\Delta D$ is the diameter step of the numerical integration. Then ($\kappa$ < 1) gives the ratio between the mass concentration presented by our PSD and the mass concentration of hydrometeors given by NWP model $Q_m$. Then we can calculate the corrected particle number distribution $N^{corr}(D_i)$:*

$$N^{corr}(D_i) = \frac{N(D_i)}{\kappa} ,$$

*(B2)*
*"*

27. L366: "This shape differs from the traditional spheroid shape commonly used in other radar observation operators. Therefore, in this section, we will delve into the design of the database" – Why is that relevant for the DB design, i.e., how does that affect the DB design or is a good example for it? Moreover, where design is discussed (subsecs 3.1 and 3.2), it's neither relevant whether it's rain or any other hydrometeor category nor what the shape model is. On the other hand, subsections 3.3-3.5 do not present or discuss DB design, but evaluate the contents of the DB.

Response: We acknowledge that the shape of particles does not affect the database design in ZJU-AERO. We reformulated the introduction of Section 3.

*"In Section 2, we provide a comprehensive description of the design of ZJU-AERO. Specifically, we emphasize that the hydrometeor optical properties database includes the elements of Z and K in units of mm² for single-scattering properties and those of <Z> and <K> in units of mm²·m⁻³ for bulk-scattering properties, both in the FSA convention. In the first two subsections, we will delve into the design of the database*

*(LUT) in ZJU-AERO with more details.*

*Furthermore, ZJU-AERO currently encompasses three types of hydrometeors: (1) rain, (2) snow, and (3) graupel. Among these hydrometeors, the rain category offers a non-spherical shape parameter option, known as the Chebyshev shape. This shape differs from the traditional spheroid shape commonly used in other radar observation operators. Therefore, we will evaluate the database contents using the Chebyshev raindrop as an example in the last three subsections."*

28. L386: "of the lookup table (stored in netCDF4 format)" – Are all DB Levels stored in one table/file?

    Response: No. Data of different DB levels are stored in separate files. Each DB file saves the scattering properties of one DB level, one frequency, and one hydrometeor category.

    *"(stored in netCDF4 format files separately for different database levels)"*

29. Tab2: "that a single database for one hydrometeor and once frequency occupies" – what is a "single database"? As opposed to a (single) lookup table (file)? Does "one hydrometeor" refer to one hydrometeor class (i.e. all sizes) or a single particle size?

    Response: (1). "Single database" refers to a single database lookup table file (in netCDF4 format).
    (2). One hydrometeor refers to one hydrometeor class.

    *""Volume" gives an estimation of the external storage that a single database lookup table file for one hydrometeor class and one frequency occupies."*

30. Tab2: Angle brackets typically indicate integrated variables. Note, that even RSSP and ASSP Z and K in your DB are already integrated (over azimuthal orientation and shape/canting distributions, respectively), i.e. they should technically have angle brackets, too.

    Response: Angle brackets are used only indicate integration over the particle size distribution in this study. Integration over azimuthal orientation and shape/canting distributions is indicated by overlines. A footnote has been added to clarify this.

    *"Angle brackets are only used to indicate integration over particle size distribution in this study. Integration over azimuthal orientation and shape distributions is indicated by overlines."*

31. Tab2: Are the full 16 elements of both Z and K kept (since only the diagonal 2x2 blocks of Z and only entries of K are used)?

    Response: (1) For the Z-matrix, only the following elements are stored: Z11, Z12, Z21, Z22, Z33, Z34, Z43, Z44
    (2) For the K-matrix, only the following elements are stored: K11, K12, K34.
    The elements that are stored are specified in Appendix B.

32. Tab2: How are the different PSD handled in the DB? Separate tables (files) (or DBs?)?

    Response: Different PSDs are handled by saving them in separate files. This is indicated in the dimension column as:
    *"(Different PSD schemes in separate files)"*

33. L396: This whole paragraph seems rather basic. Or standard. A textbook reference, eg to

Mishchenko, would suffice in my view (but references are urgently needed in any case!). It stops to elaborate, however, where it would get more interesting, e.g. how the reference orientation is set for prolate or for irregularly shaped particles.

Response: (1). We considered removing this paragraph, but many symbols will be undefined in the proceeding paragraphs and sections. So, we think it is better to preserved it, clearly specifying the notations of orientation preferences, as it will be very important in a radar forward operator to be featured with arbitrary shape particles.

(2). We have not implemented particles of prolate or irregularly shaped particles currently (only a coated spheroid model is under development for melting particles, whose reference orientation is set similarly with spheroid one). The reference orientation of irregularly shaped particle (such as aggregates snow) should be aligned with its Minimum / Maximum dimension to allow for a convenient representation of its orientation preferences.

34. L404f: "ZYZ convention" is at least ambiguous since there are two coordinate systems with unidentical axes, i.e. with respect to which of these axes (L- or P-system) are the rotations performed?

Response: It is with respect to the axes of P-system.

*"Using the ZYZ convention of the Euler angles specified by α, β, and  γ  (rotations were performed with respect to the $OZ_P$, $OY_P$, and $OZ_P$ axis, respectively)"*

35. L408: What is meant/referred to by "arbitrary orientation preferences"? Do you mean arbitrary orientations (but that wouldn't cover numbered item 4)? or actually orientation preferences, but numbered item 4 is neither arbitrary (random in alpha and gamma is already quite specific).

Response: Yes, we separate the introduction into two parts:

*"With a specified scattering geometry, we can now outline the procedures for computing scattering properties of particles with various orientations (steps 1-3) and then integrate over them for scattering properties with specific orientation preference (step 4): "*

36. Eq(24): Standalone equation without any referring text??? Why X has one and two overline(s) here, but Z and K don't have in Tab2 RSSP and ASSP entries, respectively?

Response: (1). We moved the equation to the middle of text:

*"Therefore, we performed internal averaging over α and  γ  for elements of Z and K in the scattering computation code before generating the RSSP – Level A database:*

$$\overline{\mathbf{X}}(e,\beta) = \frac{1}{4\pi^2} \int\limits_0^{2\pi} \mathrm{d}\alpha \int\limits_0^{2\pi} \mathrm{d}\gamma \mathbf{X}(e,\alpha,\beta,\gamma), \mathbf{X} = \mathbf{Z}, \mathbf{K}$$

, *(29)*

*Hence, the Level A database only has two residual scattering geometry dimensions: (1) e and (2) β, allowing for a reasonable volume for archiving."*

(2). We added footnotes to clarify that we have omitted the overlines for brevity elsewhere in the paper, for database elements, where no ambiguity could arise:

*"Overlines over elements of Z and K are omitted for brevity for level A database elements elsewhere in this paper."*

*"Two overlines over elements of $\mathbf{Z}, \mathbf{K}, \langle \mathbf{Z} \rangle,$ and $\langle \mathbf{K} \rangle$ are omitted for brevity for level B/C database elements elsewhere in this paper, and in formulas of polarimetric radar variables."*

37. L432: "Eq. (25) gives [...] Level A to Level B database conversion tool" – p(beta) given here is not general, but specifies a Gaussian distribution in polar angle with standard deviation sig_beta.

    Response: We separated the equation. We declare that the p(beta) is just a parameterization explicitly in the text:

    *"The integration in the Level A to Level B database conversion tool can be formalized as integration over canting angle β as follows:*

    $$\overline{\overline{\mathbf{X}}}(e) = \frac{1}{\pi} \int_0^\pi \mathrm{d}\beta \, \mathrm{p}(\beta) \overline{\mathbf{X}}(e, \beta), \mathbf{X} = \mathbf{Z}, \mathbf{K} \qquad (30)$$

    *Here* $\mathrm{p}(\beta)$ *is the probability distribution of canting angle β. A Gaussian distribution in polar angle is often used to approximate the probability distribution of canting angle:*

    $$\mathrm{p}(\beta) = \sin(\beta) \exp\left(-\frac{\beta}{2\sigma_\beta{}^2}\right), \qquad (31)"$$

38. L434: "It is important to consider particle symmetry" – Not that important, in my view. One can save a bit of computational resources, but results will still be correct when symmetries are ignored. However, it is far more important – and relevant for ensuring correct results – to not accidentally implicitly assume any symmetries in arbitrary particle shapes (considering or neglecting flipped and/or mirror-symmetric counterparts).

    Response: The computational resources required for microwave spectrum scattering are not so demanding. But considering the n-fold azimuthal symmetry can save a lot of time when generating level-A database for some shapes. We only have to modify the integration lower and upper bounds of Euler angles (alpha and gamma). It will not affect the size and interface of LUT. I think considering some of the symmetry worth its efforts.

    Nevertheless, we remove the words saying it is important.

    *"Particle symmetry can be considered in the orientation averaging of Eqs. (29) and (30) to avoid redundant evaluation"*

39. L439f: "a more accurate [...] commonly used spheroid model" – add reference for this statement (Ekelund20 rather concluded that Chuang and Beard model is no significant improvement at least regarding radar reflectivity compared to the spheroid model)

    Response: This sentence does not state that significant improvements in radar variables simulation can be found if Chebyshev-shaped raindrop model is introduced. It just states that the Chebyshev-shaped raindrop is more realistic than spheroid in geometric sense, as observed

by camera of wind tunnel experiment (Pruppacher et al., 1998).

*"By incorporating these factors, a more accurate representation of raindrop model* *can be achieved compared to the most commonly used spheroid model, as proved by* *photographs of water drop falling at the terminal velocity in air (Pruppacher et al.,* *1998)."*

40. L442: "The obtained results were then fitted [...]" – by whom? Are you still summarizing Chuang and Beard (1990)? Or is this your own work? Reference properly – this applies to the whole paragraph, actually (if it's all Chuang and Beard, reduce to only repeat the crucial info and refer to Chuang and Beard for the rest – e.g. why are the truncation and the Deq range and spacing in the coefficient calcs relevant here?)

    Response: Yes. It is the work of Chuang and Beard.
    We removed unnecessary information and referred to Chuang and Beard for the rest.

41. L452: "Comparing [...] with significantly different aspect ratios is meaningless" – Why is that? Predicted radar variables (for a given D or PSD) are the relevant parameters to be compared, regardless of the underlying aspect ratio, or the maximum dimension, or any other possible affecting particle property. It is interesting, though, to understand which and how impact each of the properties contributes.

    Response: Yes. The expression is modified.
    *"Comparing the optical properties of two shapes with significantly different aspect* *ratios cannot reveal the effects resulting from introducing Chebyshev shapes."*

42. L458: "Therefore, we can confidently assert" – I do not fully agree as this does not provide info yet on how much difference in SSP results from such seemingly small diffs in gamma. Only comparing SSP of particles with equivalent gamma and Deq could provide that confirmation.

    Response: Yes. We compared the SSP results of spheroids and Chebyshev shape raindrops with identical Deq=7mm and 1/gamma=1.7 to prove this:

[Figure]

Figure 1: The SSP factor of radar variable horizontal reflectivity (a) and horizontal attenuation (b) at temperature=283K, beta=0 deg., Dmax=7mm, and with identical reciprocal of aspect ratio = 1.7 for both Chebyshev and Spheroid raindrops.

Now it is safe to draw a conclusion that the differences in SSP results arise from the higher order shape specifications rather than the first-order aspect ratio parameter.

*"Therefore, we can infer that the differences in optical properties between the spheroid and Chebyshev models arise from higher-order shape specifications rather than the first-order aspect ratio parameter (also proved by comparisons between two shapes with identical aspect ratio, figure not shown)."*

43. L460: Does equal Deq imply equal Dmax (i.e. a=a')? Hence do diffs in gamma exclusively result from differences of minor axes b and b'?

Response: (1). Equal Deq cannot guarantee equal Dmax. Equal Deq only imply the volume occupied by two raindrop shapes are equal.
(2). The tiny differences in gamma can result both from major axes 'a' and minor axes 'b'.
We have stated in the footnote that we use Deq to measure the size of raindrops, while Dmax is used to measure the size of solid hydrometeor particles.

44. L467: "The initial examination [...] by Ekelund et al. (2020)" – Reformulate. This seems to imply Ekelund20 were the first to study rain drop SSP (I'm not even fully sure, they were the first using the Chuang and Beard and/or the Chebyshev model – actually, Aydin gives a short summary of Chuang and Beard effects in their Chapter on cm- and mm-wave scattering from hydrometeors in the Mishchenko00 book).

Response: Yes. We reformulate this sentence and remove the statement that this reference is the initial examination.

*"A detailed examination of the optical properties of Chebyshev model rain droplets and their sensitivity against traditional spheroid model were conducted by Ekelund et al. (2020)."*

45. L468: "they used a modified version of the EBCM code" – modified in what way? Ekelund et al. themselves state they used "the Fortran T-matrix code developed by by Mishchenko (2000)" and don't mention modifications.

Response: "modified" should be "extended."

*"they used an extended precision version of the EBCM T-matrix code (Mishchenko and Travis, 1994)"*

46. L471: "To ensure [...] a user-friendly radar operator interface, this study presents an optical property database [...] using the IITM code" – Check language: is it really that presenting the study ensures those things? What does user-friendliness have to do with shape model and T-Matrix code applied?

Response: (1). Ekelund20 stated in his paper that "It was unfortunately difficult to reach convergence for all sizes and frequencies, specifically at the temperature 310K where the imaginary refractive index is exceptionally high. As an example, the imaginary part of the refractive index reaches as high as 2.77 at 40GHz. However, it was found that convergence could be reached if the number of Chebyshev coefficients was reduced" and "The size grid is limited by the numerical instability of the EBCM method for particles that are big or have high aspect ratios."

However, IITM code has no limitations on that. It can provide a database with perfect integrity and accuracy, covering the giant drop size range. (2). User-friendliness only concerns the RFO interface of the DB, not the DB itself, and which T-matrix code we used to calculate it:

*"To ensure integrity and accuracy, this study presents an optical property database (with a user-friendly radar operator interface) for the Chebyshev raindrop model at the Ku- and Ka-bands (13.6 and 35.5 GHz, respectively) using the IITM code (Bi et al., 2013)."*

47. L479: "raindrops with their Deq larger than 8mm disperse easily" – Provide a reference for this statement. Ekelund20 argue with 5mm, supported by two references.

Response: Yes. Ekelund20 states that drops larger than 5mm tend to become unstable and break up. However, according to his reference Kobayashi2001, drops larger than 5mm (8mm giant drop) can appear during the onset of heavy rain.

The process of collision–coalescence and break-up in precipitation system is dynamic. That implies that giant drops (Deq=5mm~8mm) can be occasionally found in nature, though it is unstable. It can exist for a while but disperse during its falling stage. Additionally, Table 2.1 in Guifu Zhang's textbook about polarimetric weather radar states that the typical size range of raindrop is 0.1-8mm.

We modified our expression here:

*"Since raindrops with their Deq larger than 8 mm are beyond the typical size range (Zhang, 2016) and rarely found in nature (Kobayashi and Adachi, 2001)"*

48. L482: "we need to introduce intermediate quantities called the SSP factors" – Why do you NEED to introduce them? You just chose to (hopefully because they are illustrative and facilitate understanding). However, what do you mean by "SSP factors", what are they? Explain!

Response: (1) Yes. We choose to introduce those SSP factors.
(2). We explain what is "SSP factors of radar variables" as follows:

*"Additionally, we choose to introduce intermediate quantities called the "SSP factors for radar variables", which are illustrative and facilitate our understanding of how SSP of a given-sized particle affects radar variables."*

49. L483: "The SSP factor [zhv] for the unpolarized reflectivity" – Why using unpolarized reflectivity when radars ALWAYS measure a specific polarization? Using hv as subscripts in z seems a bad choice since zhv usually refers to horizontally transmitted/vertically received reflectivity occuring as the nominator in the definition of linear depolarization ration, LDR.

Response: We chose the unpolarized quantity for analysis, not for radar variables calculations. Yes, we switched from unpolarized quantities to horizontal-polarized quantities to avoid confusion, throughout the paper.

50. L486: "enable us to assess the contribution of particles with a diameter Deq to the radar reflectivity" – How? Provide a proper explanation of what the SSP factors are. And isn't that, i.e. "contribution of particles with Deq to reflectivity", what backscatagrand later will provide?

Response: (1). The expressions of "SSP factor of radar variables" are irrelevant of the PSD of hydrometeor, it facilitates our understanding on how the SSP of given-sized particle affects the radar variables:

$$\frac{\lambda^4}{\pi^5 |K_w|^2} \int \sigma_{bsca,h}(D)N(D)\mathrm{d}D$$

(2). When the PSD of hydrometeor is fixed, then we can calculate something like backscatagrand, which is a measure of the specific contribution of given-sized particles to the radar variables:

$$\frac{\lambda^4}{\pi^5 |K_w|^2} \int \sigma_{bsca,h}(D)N(D)\mathrm{d}D$$

In summary, SSP factors of radar variables describe how SSP affects radar variables. But only when PSD is determined, we can use quantities such as backscatagrand to measure the specific contribution.

*"These factors ($[z_h]$ and $[a_h]$) describe how a particle with a diameter Deq affects the radar reflectivity and specific attenuation, respectively."*

51. Fig8/Fig9: Looks fancy, but would a simple cartesian plot not suffice, be easier to digest and be less prone to biased interpretation? Particularly since the center of b) panels is not 0, but 60 (i.e. deviations are even smaller than they appear here). Radial axis of a) panels is unclear – it's neither linear nor true logarithmic, isn't it? (also, the middle value, I guess, should rather be $5.10^2$ instead of $5.10^3$). Displaying z-equivalents in log scale (ie dB) seems more suitable than linear scale (ie $L^2$ units) to judge relevance/significance of differences in radar applications. Why are you showing the elevation range, apart from -30 to 30° masked sector, twice? That's not symmetry, but identical parameters elevations!

Response: Thanks for suggestions, we switched to a simple cartesian plot, and make both y-axes as log-scale.

[Figure]

[Figure]

SSPs of Raindrop @ D = 7.00 mm, T = 283.0 K, Ka-band (35.50 GHz)

52. Fig8: Why is T-variation relevant to show? (Is variation with T entirely due to temperature dependence of refractive index, or are there any other T-dependent parameters?)

Response: (1). Temperature is one of the DB dimensions. We want to see how the shape effects compared with the temperature effects, since the dielectric constants of water change rapidly for temperature ranges from 263K – 313K at Ka-band.

(2). The variation with T entirely due to temperature dependence of refractive index. There are no other T-dependent parameters since they are quantities derived from SSP, not BSP.

*"The variations with respect to temperature entirely originate from temperature dependence of water dielectric constants."*

53. L517ff/Fig8: "we found that the [zhv] factor for ground-based observation geometries (e=0~20°) was significantly weaker" – First, differences at e=0° seem (close to) 0 to me. Second, what differences do you consider "significant"? I find them anyways hard to judge in linear units, this would be much easier in dB units.

Response: (1). We modified the axis to log-scale.
(2). It is not so significant for [Zh], actually.

*"Additionally, we found that the $[z_h]$ factor for ground-based scattering geometries*

*(e = 0~20°) was close to the values obtained for the spheroid model."*

54. L524ff: "demonstrate the stability of the deviations [...] in terms of the temperature [...] However, the finding is different for [zhv] against the orientation preference" – Yes, of course. Varying orientation present a significantly different (geometrical) cross-section to the observer, while varying T (apart from phase changes) essentially keeps the same "view" of the roughly same particle.

Response: Yes. We add your interpretation to the text.
*"It can be interpreted that varying orientation presents a significantly different "view" of raindrop to the observer, while varying temperature essentially keeps the same "view" of the raindrop with different dielectric constants."*

55. L528f: "Fortunately, the column 'orientation preference' in Table 1 shows that the standard deviation of beta=7° did not exceed this threshold for raindrops." – Fortunately? First, consider language: fortunate might be that rain drop tumbling is typically small, but it

hasn't to do with fortune that your Table shows the value it shows (I hope at least; or did you throw dices?). Second, why would higher sigma be "unfortunate"? And what would you do in such an "unfortunate" case, like e.g. in strong turbulence cases?

Response: Thanks. It might be inappropriate to use such words here.

*"Since the column "orientation preference" in Table 1 shows that the standard deviation of β = 7° for raindrops in normal turbulence conditions did not exceed this threshold for raindrops, we can infer that the CmS deviations found for β = 0° only show a minor decrease if orientation averaging is performed."*

56. Fig10: That is quite a rich plot. Considering that, there is very little presentation of its content and discussion/interpretation of it in the text. Particularly, there is practically none at all on the polarimetric variables. Without that, skip the respective plots/panels.

Response: We add analysis to polarimetric variables in the text. That explains why the CmS effects of BSP for polarimetric variables is not examined later in this study, which are mostly not significant in SSP DB.

*"As for other SSP factors of polarimetric radar variables, the CmS deviations are either not stable against size spectra (Figure 9h&i) or too small in terms of its base magnitude (Figure 9l), making them not significant for a particle group. However, the CmS effects of SSP factor for differential attenuation ($[a_h]$- $[a_v]$) in Figure 9k is significant at low absolute elevation angles, indicating that the SSP factor of vertical polarization attenuation $[a_v]$ is significantly stronger for Chebyshev model (considering $[a_h]$ is close for two shapes at low absolute elevation angles in Figure 9j)."*

57. L540f: "focus on describing the PSD options [...] (namely, ASSP to BSP conversion" – This subsection is supposed to be about DB Level B, i.e. according to Fig.2 the orientation and shape distribution averaging. Hence unclear why PSD appears here already.

Response: Yes. It is better to state orientation averaging and Level-B LUT of raindrops do not worth further analysis at the end of Section 3.3, and change the title of Section 3.4 to make it clear.

"*The canting angle of raindrops is generally small, so the CmS deviations of SSP factors found in Figure 7 and Figure 9 only show a minor decrease in Level B – ASSP database when compared to the curves of β = 0° in Level A – RSSP database. Hence, we omit further analysis of orientation averaging of raindrops and SSP factors of Level B database here.*

**3.4 PSD Options and Analysis of Level-B to Level-C LUT Conversion**

*In this subsection, we will focus on describing the PSD options for raindrops and the methods used to analyze their impacts on the conversion of SSP factors of radar variables to intrinsic radar variables (namely, ASSP to BSP conversion).*"

58. L542: "a total of six options for PSD schemes for raindrops [...] listed in Table 3" – Inconsistency? Tab.1 only lists 4 options?!

Response: (1). Yes, we omitted user option C5 (Walters et al., 2011) in Table 1. We added that option in Table 1.
(2). The "ThompsonTuned" PSD scheme is not a user option. It is only for experiment now. We declare this point in the footnote of Table 3.

*"1 The ThompsonTuned PSD scheme is for numerical testing only, therefore it is not*

*a user option in Table 1."*

59. Tab3: Meaning of x1 and x2 parameters to be introduced/shown by an equation. What is the relation of the six schemes listed here to the Cx schemes listed in Tab1?

Response: (1). We have added reference to equation (22) in the caption of Table3:

*"(1) MP1948, (2) AB2012, (3) Walters2011, and (4) Wang2016 can be generalized as a power-law $N_0$-$\Lambda$ relationship (see Eq. (22)) as stated in Section 2.3"*

(2). The 4 schemes in Group A and the Thompson2008 in Group B are the 5 user options in Table 1. The "ThompsonTuned" PSD scheme is not a user option. It is only for numerical testing now.

60. L572: "Figure 11 demonstrates that the uncertainty of intercept parameter No [...]" – How? Explain better. Do you really mean uncertainty? Or rather spread among PSD schemes?

Response: Yes, I mean the variability (spread) among PSD schemes. We add specifications on how to see that variability:

*"Figure 10 demonstrates the large variability of intercept parameter $N_0$ for a specified rain water content (RWC) among different PSD schemes. That variability can be shown by measuring the vertical coordinate difference of the intersection points between a given iso-RWC line and different $N_0$-$\Lambda$ lines of PSD schemes."*

61. L572: "Thompson scheme [...] priorities smaller drops" – prioritizes. Can this be seen in Fig11? How?

Response: For PSD schemes expressed by exponential distribution, larger intercept parameter $N_0$ means more smaller drops when the RWC is fixed. Intersection points between $N_0$-$\Lambda$ curves of PSD schemes and iso-RWC lines can be found to verify this point.

*"For PSD schemes expressed by exponential distribution, larger intercept parameter $N_0$ means more smaller drops when RWC is fixed. Therefore, the Thompson scheme (Thompson et al., 2008) priorities smaller drops, while the Wang scheme (Wang et al., 2016) places more emphasis on larger drops."*

62. L575ff: "For example, stratocumulus [...] production of unusual PSDs" – I do not understand what you are trying to say and what this has to do with PSD in RFO or even with DB Level B/RSSP.

Response: We removed those sentences and stated the DSD variabilities using terms of "maritime precipitation" and "continental precipitation", to make the logic clear:

*"According to global aircraft in-situ measurements carried out by Abel and Boutle (2012), the six single-moment (SM) microphysics assumptions in Table 3 cover the natural variability of raindrop PSDs ranging from continental precipitations to maritime precipitations. Typical continental PSD with intensive large drops such as Wang et al. (2016) and typical maritime PSD such as Thompson et al. (2008) with a large population of drizzle-sized raindrops are both present. That is why we have chosen the six SM microphysics schemes for the PSD sensitivity test."*

63. L579: "we introduced the concept of 'backscatagrand'" – Language/writing style: You didn't introduce anything yet, you are going to do that now. Don't announce that, but do it. Not just by naming it and already pointing to results, but EXPLAIN it. The current

attempt on explanation (L583ff) is too confused, too little to the point.

Response: Yes. We reformulate our representation:

We first give the definition of "backscatagrand" and then explain why we introduce it and what it helps us identify particles of which size-range dominate the energy of backscattering.

We review the definitions of "extagrand" and the reason why Geer at al. (2021) have introduced the "extagrand": that is out of the insight that particles in a narrow range of size contribute to the bulk scattering properties in radiative transfer simulations of passive sensors.

Then we propose our parallel definition of "extagrand" for active sensors (weather radars).

*"To explore the effects of different PSD schemes on the radar variables, a new intermediate quantity called "backscatagrand" is analyzed in Figure 11, which is a spectra of particle size defined as the product of horizontal backscattering cross section $\sigma_{bsca,h}(D)$ and size distribution function N(D). It helps us identify particles of which size-range dominate the energy of backscattering."*

64. Fig12: Include Thompson tuned here as it is used as the basic PSD in Sec4 (there even before discussing the need for and approach of tuning).

Response: Done.

[Figure]

65. L604: "It can be concluded from Figure 12" – Before making conclusions, present the results and discuss them.

Response: Yes. It is not a conclusion, but one of the results.

*"In Figure 11, we can find that the mass distributions and backscattering contribution spectrum (i.e., the so-called "backscatagrand") exhibit large uncertainties due to natural variability of PSD schemes."*

66. L609: "unique to modern PSD schemes" – Is it? Why? What defines a "modern" PSD scheme? Is T08 one or not? Besides, though not exactly coinciding peak & dip for MP48, backscatagrand still exhibits a kind of dip (or hiatus in its decrease).

Response: (1). No exactly, it is unique to continental PSD schemes that emphasize large drops.

(2). In this study, we refer to every PSD scheme except MP1948 (which

uses a constant intercept parameter $N_0$) as modern PSD scheme.

(3). Yes, there is a weak dip of backscatagrand for MP1948. But what we concern is the loss of reflectivity due to Mie oscillation effects: only when a large portion of hydrometeor mass get distributed within particle size interval of mass backscattering efficiency dip (caused by Mie oscillation effects), the loss of reflectivity will be prominent. MP1948 is not the case, its mass distribution is concentrated at around Deq=2mm (see Figure11d).

*"This phenomenon is unique to PSD schemes that contain more large drops (such as Wang2016 and AB2012)"*

67. L610: "leading to a loss of bulk reflectivity" and L615f: "T08 scheme must be much smaller" – Would ease discussion (by reducing unnecessary speculation), if you not just state, but show that. E.g. indicate resulting bulk zhv for each PSD and QR in the Figure 12 panels (lower row), e.g. by a symbol at placed at end of x-axis.

Response: We considered this suggestion, but there is not enough space in this Figure to hold so many results of bulk zhv for each PSD and each QR.

On the other hand, those results are shown in the figure of next section. The discussions and hypotheses will be verified in Section 3.5, which is part of the Level C LUT results.

68. L617: "relative importance [...] can also be diagnosed with [...] backscatagrand" – that's its ultimate purpose!

Response: Yes.

*"The relative importance of particles in the entire spectrum can be diagnosed with the curves of backscatagrand, which is the ultimate goal to introduce this concept."*

69. L625: "we made speculations" -> "we hypothesized". However, it would make for a smoother read if you avoid the need for hypothesizing/speculation by already presenting the resulting equivalent bulk properties belonging to the Figure 12 curves.

Response: (1). Done.

(2). We considered this suggestion, but the analysis of backscatagrand will be entangled with the analysis and discussions of level C LUT if we do this.

70. L627ff: Which of these conclusions actually require/are affected by the chosen shape model? How do your findings on Chebyshev shape model compare to Ekelund20's findings?

Response: (1). The conclusions 3 and 4 are talking about the CmS (Chebyshev minus Spheroid) effects, which are affected by the shape model.

(2). The conclusions on CmS deviations focusing on Ka-band (35.6GHz) are a little weaker than those at W-band (94.1GHz) shown by Ekelund et al. (2020). However, the patterns confirm well.

*"Those conclusions focusing on Ka-band (35.6 GHz) generally confirm well with what were found by Ekelund et al. (2020)."*

71. L642f: "sensitivities of polarimetric intrinsic radar variables [...] were not examined in this

paper" – Then there is no reason to present them at all!

Response: Yes. I have removed those panels of polarimetric radar variable in Figure 12.

*"The sensitivities of polarimetric intrinsic radar variables with respect to CmS deviations for ground-based radar bands (S/C/X-band) and viewing geometries were found to be negligible. Therefore, they are not shown here."*

[Figure]

72. Fig13: Is presenting polarimetric variables for close-to-nadir angles useful, or aren't they more interesting in ground-based measurements? Why only 3 PSD schemes included here? It might be more useful to present ZH and ZV separately (though not necessarily for close-to-nadir observations).

Response: (1). Presenting polarimetric variables for close-to-nadir angels are not so useful, so we have removed them (see above).

(2). We have examined the shape effects at Ka-band on ground-based measurements, it was found that the effect is only prominent for $Z_{DR}$ and $\alpha_H$-$\alpha_V$. The effect is even weaker for frequencies (S/C/X-band) where ground-based weather surveillance radar works at. Therefore, we do not show them in the paper.

[Figure]

Figure 2: The BSP of raindrop for different PSD schemes and shape models at temperature=283K, elevation=1 deg., and Ka-band (35. 5GHz).

(3). We add explanations on why only show results on 3 PSD schemes in the caption of Figure 12.

*"Here, only the results of PSD schemes of MP1948, Thompson2008 and Wang2016 are displayed, because MP1948 is a benchmark traditional PSD scheme, while Thompson2008 and Wang2016 are representative of typical maritime and continental precipitation PSD, respectively."*

(4). I think it is better to present ZH and ZDR, rather than ZH and ZV, to observe the difference between ZH and ZV.

(5). Yes, ZH and ZV are extremely close for close-to-nadir observations, almost undetectable.

73. Sec 3.3-3.5: What do these subsections provide beyond what Ekelund20 already did? That is, what justifies this lengthy presentation (it's 10 pages, ie ¼ of the whole manuscript). Particularly since Ekelund20 concluded differences in reflectivity from Chebyshev and spheroid shapes are negligible for ground-based weather radars and still pretty small in nadir-viewing geometries.

Response: (1). There are several points distinguish our work from Ekelund20:

a. Ekelund20 present the results of BSP directly, with little discussions and analysis on the mechanisms about how raindrops of different sizes affect the radar variables. However, we present the SSP of giant raindrops, and introduced an intermediate quantity called "backscatagrand" to facilitate understanding on how raindrops of different sizes contribute to the radar variables.

b. Ekelund20 used the code of EBCM T-matrix. Due to numerical stability issues, the code he used is unable to calculate the SSP of giant drops (Deq=7mm). The maximum diameter of raindrop in his database stops at Deq=5.75mm. However, this study uses IITM T-matrix code, we calculated the exact SSP results of giant drops (Deq=7mm) and present them in the paper. Additionally, results

calculated by IITM provide a cross-validation for the results of EBCM.
c. This paper focuses the results of Ka- and Ku-band, primarily for applications of spaceborne precipitation radar.

Moreover, those contents (Section 3.3-3.5) play an important role in this paper: It provides a sample of LUTs in ZJU-AERO. It also illustrates how we interpret and understand the results of scattering computation results for DBs of this RFO.

(2). Chebyshev raindrop model shows noticeable differences of bulk scattering properties (compared to the spheroid model) for electromagnetic waves of millimeter-wavelength at zenith and nadir observation geometries. This difference is more prominent for continental DSD models (e.g., Wang2016) with larger drops. The reflectivity contrast between zenith and nadir viewing angles can be as high as 4-5 dB, as illustrated in Figure 14. Such a difference is non-existent for spheroid raindrops with reflectance symmetry. Given the lower uncertainties in simulating reflectivity of liquid phase compared to solid and melting phases, such a difference deserves attention in specific applications such as comparing the data from ground-based and spaceborne radar observations:

*Warren, Robert A., et al. "Calibrating ground-based radars against TRMM and GPM." Journal of Atmospheric and Oceanic Technology 35.2 (2018): 323-346.*

[Figure]

Figure 3: The BSP of raindrop for Wang2016 PSD scheme temperature=283K, elevation=1 deg., and Ka-band (35. 5GHz) for spheroid and Chebyshev shape raindrop at nadir (elevation=-90 deg.) at zenith (elevation=90 deg.)

74. L668: "has switched its scan pattern" – How was it before? And/or how is that relevant here?

Response: (1). Before its scan pattern change, only a swath width of 125km (25 beams) can form a matched scan (MS) of Ku- and Ka-band. After the scan pattern change, the whole swath width of 245km (49 beams) can form a MS of Ku- and Ka-band. The picture from (Iguchi et al., 2010) vividly show this:

[Figure]

Figure 0.2 DPR's scan pattern before May 21 2018.
(a) and after May 21 2018 (b). KaHS beams scan in the inner swath before May 21 2018, but now they scan in the outer swath and match with KuPR's beams. Numbers in color indicate angle bin numbers for KuPR (blue), KaMS (yellow), and KaHS (red).

Figure 4: The scan pattern change (courtesy to Iguchi et al. (2010)).

(2) It is relevant because now we can get a full MS observation of Ku- and Ka-band with a swath width of 245km for comparison and evaluation of our RFO.

*"However, since May 21, 2018, the GPM-DPR has switched its scan pattern, so that now a full swath can be considered as the matched beam in the evaluation of forward radar operator."*

75. L678: "the structures [...] are described as contiguous and vague" – rather "appear more contiguous and vague"? Is this from visual impression or any quantitative measure?

Response: (1). Yes. Done.

(2). It is from visual impression. Such a property is something more like the texture of images, which is hard to be quantified as numbers (maybe AI can) but can be easily noticed by human eyes.

76. L681: "simulated radar reflectivity should be unbiased [...] we assume [...] was roughly unbiased" – Are the reflectivites unbiased? Analyze and show! On what grounds do you assume the model to be unbiased?

Response: (1). Here we just state that the reflectivity should have no significant systematic bias against observation if both the RFO and NWP model has no significant bias. Detailed examination on whether the reflectivity is unbiased is given in the PDF analysis in Section 4.2.

(2). NWP model precipitation is roughly assumed as having no significant bias because the precipitation forecast of regional NWP model can be frequently calibrated against the rain gauge network:

*Bárdossy, A., & Das, T. (2008). Influence of rainfall observation network on model*

*calibration and application. Hydrology and earth system sciences, 12(1), 77-89.*

*Cattoën, C., Robertson, D. E., Bennett, J. C., Wang, Q. J., & Carey-Smith, T. K. (2020). Calibrating hourly precipitation forecasts with daily observations. Journal of Hydrometeorology, 21(7), 1655-1673.*

As for the spin-up issues in GRAPES-NWP model, the forecast lead time is about 9 hours in this case. A 9 hours lead time can ensure that the RWC in the model has spined-up.

*Ma, Zhanshan, et al. "Spin-up characteristics with three types of initial fields and the restart effects on forecast accuracy in the GRAPES global forecast system." Geoscientific Model Development 14.1 (2021): 205-221.*

*"Although the NWP model could not accurately predict the cloud and precipitation timing and position, the probability distribution of simulated radar reflectivity should be unbiased when compared with observations (detailed examinations on that will be conducted in Section 4.2); otherwise, a systematic bias in the NWP model or the radar operator might be identified. Since (a) the precipitation forecast of CMA-MESO model can be frequently calibrated against rain gauge network (Bárdossy et al., 2008; Cattoën et al., 2020) and (b) a forecast lead time of 9 hours is beyond the 6-hour spin-up time of rain water content in CMA-NWP (Ma et al., 2021), we assume that the NWP model CMA-MESO has no significant bias in this case."*

77. L685f: "observations revealed a prominent bright band" – Really? Where? I can't identify that. Prominent BBs aren't particularly common in inhomogeneous atmosphere states.

Response: Yes, not so prominent here.

*"Actually, the GPM-DPR observations revealed a weak bright band (BB) signature below the freezing level, which can be recognized at Ku-band (with an average reflectivity enhancement of about 3 dBZ, see Figure 14a) but not so clear at Ka-band (see Figure 14b)."*

78. L689f: "it was expected [...] have a positive OmB signature" – Why? Due to model lacking melting scheme? Be clear.

Response: Yes.

*"Based on the observations and simulations, it was expected that the melting layer would exhibit a positive OmB signature due to lack of melting hydrometeors in ZJU-AERO."*

79. Fig14: What RFO setup is used here? Describe in text, not in figure caption!

Response: Done. Also for the figure in Section 4.2.

*"The radar operator applied the ThompsonTuned PSD option (for developers only) and the Chebyshev morphology options A2 (see Table 3) for the category rain in the simulations. For snow, we use default option A2. For graupel, we use default option A2 and B1."*

80. L698: "altitude levels (namely, 0, 3, 5, and 8km)" – Isn't 0km in the blind zone?

Response: Yes, 0km is the blind zone. Actually we plotted the lowest clutter-free gate. It is better to mark it as "NearSurface".

*"namely, NearSurface (the first clutter-free gate), 3, 5, and 8 km"*

81. L700: The post-processing information should go into the text and be described more

clearly.

Response: Done.

*"The reflectivities in Figure13a–d are masked by a flag called "flagPrecip" (available at the ground) offered in the L2A swath data of GPM-DPR, while the simulation of reflectivity in Figure13e–h is masked by the sensitivity threshold of 12 dBZ to keep it the same with sensitivity of GPM/DPR observation. As for the calculation of OmB reflectivity, the radar gates for which the reflectivity is undetectable (below the instrument sensitivity) either in simulation or observation are filled with a "background" reflectivity of 0 dBZ to generate the OmB reflectivity."*

82. L713: "a minor negative OmB bias was observed" – Visually or using any quantitative means?

Response: We can notice there are many extreme large DFR values in Figure14(f) (>30dB), while it cannot be found in Figure14(c). That is why I say there is a minor negative OmB bias of DFR.

However, when I tried to quantify this bias using the mean profile of reflectivities and DFR, I find there is actually no prominent bias of DFR at liquid phase layer (0-4km) for the ThompsonTuned PSD scheme. Such a phenomenon is caused by that the DFR of observation and simulation still have discrepancies in their PDFs.

[Figure]

Figure 5: The mean bias profile of (a) Ku-band reflectivity, (b) Ka-band reflectivity, and (c) DFR for the case of Typhoon Haishen in the Section 4 of the manuscript.

Considering the above findings, we modified the expressions here:

*"Moreover, extremely large DFR values (>30 dB) were found in simulations (Figure 14f) but not in observations (Figure 14c) in the 0 to 2 km layer."*

83. L713f: "which could be attributed to the over-attenuation" – What is over-attenuation? Why does it occur in Ka, not Ku? And only in simulations, not observations? Actually, it hasn't been made clear whether attenuation was considered in the simulations and whether the GPM observations used here were corrected for attenuation affects or not.

Response: (1). By "over-attenuation", I mean the RFO may have

overestimated the attenuation effects at Ka-band.

(2). Attenuation is considered in the simulation of RFO. We use the "zFactorMeasured" quantity in the product of GPM/DPR, which is not applied the attenuation correction. We added statement in the text:

*"We applied the attenuation in simulations, while using the "zFactorMeasured" product of GPM/DPR (no attenuation correction applied)."*

(3). It probably because the RFO tends to over-estimate the Ka-band attenuations under heavy precipitation conditions, or other unknown deficiencies of the RFO. We modified the expression here:

*"Moreover, extremely large DFR values (>30 dB) were found in simulations (Figure 14f) but not in observations (Figure 14c) in the 0 to 2 km layer, which could be attributed to an over-estimation of attenuation in the Ka-band simulation for heavy precipitations (Figure 14e)."*

84. L718: "The OmB plots [...] useful tools for verifying and calibrating new observation operators and identifying their deficiencies" – How do you separate observation operator deficiencies from NWP model deficiencies?

Response: Yes, of course NWP model deficiencies cannot be entirely disentangled with RFO deficiencies. But since the precipitation forecast of NWP calibration against rain gauge is practical, we often assume NWP model is unbiased when tuning RFO.

For more discussion on this issue, please refer to our response to specific remark 76.

85. L719: "the bias of distributions is reasonable and demonstrates the capabilities" – You did not really present any distributions (i.e. statistics)? Just scene snapshots.

Response: Yes. We modified the presentation.

*"Overall, the results are generally reasonable and demonstrates the capability of ZJU-AERO in simulating the reflectivity of dual-frequency spaceborne radar such as GPM-DPR. More analysis on bias of probability distribution of reflectivities / DFR will be presented in the next subsection."*

86. Fig16: Why not binning simulation data equivalently to observations? Why showing 4-6 and 6-8km layers when not relevant for comparing rain PSD (and shape model) effects?

Response: (1). Actually, the simulation data and observation data are binned equivalently, though they are plotted by bars and lines, respectively. We add statements in the caption to avoid confusion.

(2). Yes. We have removed the results of those layers.

[Figure]

*Figure 15: The observation and simulation distributions of radar reflectivity at the Ka band (the first row), Ku band (the second row) bands, and the dual frequency ratio (DFR) (the third row). The last row shows the log-scale histogram of rain water content ($Q_R$) predicted by CMA-MESO. The data of distributions were gathered from the reflectivity of all the radar gates in the four altitude layers (namely, the 0–2 km layer, 2–4 km layer as the tags on the top of the columns suggest). The first two layers (0–2 and 2–4 km) were primarily composed of liquid hydrometeors, while the layers above 4km contain melting and solid particles (results not shown here). The observation distributions of reflectivity are shown by grey bars in the panels, while the simulation distributions are indicated by curves with different colors and styles. The data of observations and simulations are binned equivalently between 12dBZ to 50dBZ with a bin-size of 2dBZ.*

87. L734: "Based on statistical analysis" – why not showing the PDF of QR?

Response: Done.

We add PDF of QR for Figure 15 (see response to remark 86).

*"Based on statistical analysis (see Figure 15g and h), we found that the mass concentration of rain in the CMA-MESO at the storm rain bands was primarily dominated by moderately heavy rain"*

88. L738f: "As a result, applying the Thompson2008 PSD could lead [...] (Figure 13)" – Rather "According to Sec.../Fig.13, ...". Could? Under which conditions? Or does it?

Response: Done.

*"According to Section 3.5/Figure 12, applying the Thompson2008 PSD leads to a significant reduction in the simulated reflectivity under conditions of moderately heavy rain ($Q_R \sim 10^{-3}$ kg·m$^{-3}$), which is consistent with our findings in Figure 15(a), (b), (c), and (d) for the Ku- and Ka-bands of liquid layers."*

89. L745: "Fortunately, we identified an optimal point" – By luck? Describe the tuning procedure.

Response: We added our tuning procedure in the text:

*"We use a scoring method to quantify the match between observation and simulation histograms, as proposed by Geer and Baordo (2014):*

$$s = \sum_{j=1}^{nbands} \sum_{i=1}^{nbins} \left| \log\left( \frac{N_{sim}^{j}(i)}{N_{obs}^{j}(i)} \right) \right|, \tag{36}$$

*Here, s represents the final score, with $N_{sim}$ and $N_{obs}$ indicating the counts in the bins of the simulation and observation histograms, respectively. The superscript 'j' denotes j-th band. To prevent infinite values in summation, we assume a count of 0.1 in bins with empty values in either the simulation or observation histograms.*

*Next, we establish a grid of free parameters for optimization, which includes three parameters ($N_1$, $N_2$, and $Q_{R0}$). Due to the broad range of these free parameters, the grid is set up in a quasi-logarithmic scale.*

*Then, simulations are conducted and the score from Eq. (36) is evaluated for each grid point. By identifying an optimal region in the parameter space with the best score, we refine the grid in that region to pinpoint a more precise parameter subregion. This iterative process is carried out to fine-tune the parameters.*

In practice, we constructed a 4X4X4 grid with $N_1$=[3E4, 1E5, 3E5, 1E6]; $N_2$=[2E3, 3E3, 5E4, 8E3]; $Q_{R0}$=[1E-4, 3E-4, 1E-3, 3E-3], encompassing the range of PSD schemes from Thompson 2008 to MP1948. The initial results are generally satisfactory, although further refinement through a second run could potentially improve the outcomes.

[Figure]

Figure 6: The scores for three 2D parameters meshes ($N_1$=3E3, $N_2$=1E5, and $Q_{R0}$=3E-4).

90. L754: "However, we still believe that the CmS effects could be" – Reformulate. Believing is a bad argument in science. What supports this believe? References cited by Ekelund20, however, suggest that drops do not get larger in heavier storms, but break up instead and form more smaller ones.

Response: We reformulate the expression:
According to the references cited by Ekelund20, the giant drop (~7mm) can exist during the onset of heavy storms. They form by collision and coalescing and can easily disperse, existing dynamically in particle ensembles. They are still within the typical range of raindrop size (see discussions in response to remark 47).

*"However, the CmS effects could be significant in cases of extremely heavy rain in the supercell storms according to our observation system simulation experiments (OSSE). An OSSE of GPM-DPR overpassing a meso-scale convective system (recorded with extreme heavy precipitation events) reported more than 1 dB CmS decrease effects at Ku-band and more than 2 dB at Ka-band (figures now shown)."*

The results and discussions of the OSSE are shown in the response to the Reviewer #1 / Major Comment 4.

91. L760: "demonstrated the data analysis tools in each layer" – Which layer? Which tools? You presented some analysis of rain drop DB data, but not any tools.

Response: We reformulate the expression:

*"LUT visualizations and analysis are conducted for database of different layers: Level-A (raw single scattering properties database), and Level-C (bulk scattering property database)."*

92. L765ff: Instead of a lengthy and fairly uncertain to-be-done wishlist, I'd prefer more summary of main findings of the analyses. Like a statement whether considering more sophisticated shape models than spheroids is actually really relevant.

Response: Yes, we have added those statements to the summary section.

*"We concluded that the Chebyshev-shape raindrop model shows noticeable differences of bulk scattering properties (compared to spheroid model)* only *at zenith and nadir viewing geometries and for milimeter-wavelength radar bands. This difference is more prominent for continental DSD models (e.g., Wang2016) with larger drops. These deviations can reach up to 2-3 dB at Ka-band for spaceborne radars in heavy continental precipitation regions where large drops dominate. Given the lower uncertainties in simulating reflectivity of liquid phase compared to solid and melting phases, such a difference deserves attention in specific applications such as comparing data from ground-based and spaceborne radar observations (Warren et al., 2018)."*

93. L765: What is "geometric characterization"? Before dealing with melting particles, presenting frozen hydrometeor modelling seems more appropriate.

Response: (1). We mean geometric model of melting hydrometeors here.

*"2. Improve the* modelling *of melting particles (including melting snow, melting graupel and melting hail) by using layered inhomogeneous modelling rather than an effective medium approximation for the mixture matrix."*

(2). Yes. we added the frozen hydrometeor modelling ahead.

*"1. improve frozen hydrometeors modelling by introducing irregular-shaped aggregates and riming snow."*

94. L777: "modules to interface with the data" – How does that fit (or where are these indicated) in Fig2 Flow chart?

Response: Actually, it is an external I/O module to interface with the format of spaceborne radar data format. It is difficult to fit it into Fig2 Flow chart.

*"We have already implemented an* external I/O module *to interface with the data format of FY3-RM/PMR in ZJU-AERO"*

**Spelling & other minor things:**

1. L48: Remove second "described".

Response: removed

2. L78: Add reference for PMR and DPR properties (a more current one than Iguchi03, covering the instruments that are actually flying).

Response: OK, we have added the following reference Iguchi2020:

*Iguchi, T.: Dual-frequency precipitation radar (DPR) on the global precipitation*

*measurement (GPM) mission's core observatory, Satellite Precipitation Measurement: Volume 1, 183-192, 2020.*

3. L79ff: Add reference for PR2 and RM constellation (what is RM, by the way?).

   Response: We added one reference of FY3G-PMR just accepted and published (in December 2023):

   *Zhang, P., Gu, S., Chen, L., Shang, J., Lin, M., Zhu, A., Yin, H., Wu, Q., Shou, Y., and Sun, F.: FY-3G satellite instruments and precipitation products: first report of China's Fengyun rainfall mission in-orbit, Journal of Remote Sensing, 3, 0097, 2023.*

   However, the Precipitation Radar 2 (PR2) is only in concept now (I know it from a report of Chinese domestic conference). We have not found a published paper in English.

   The "RM constellation" is referring to the Chinese rain measurement constellation: there will be more than one rain measurement satellites such as FY-3G working on the orbit in the future.

   However, there is no appropriate reference for those instrument and constellation, so we decided to not mention them in this paper. This paragraph is deleted.

4. L83: "This study is organized into …" – rather: "This paper is organized as follows."

   Response: Thanks for suggestion. Done.

5. L110: "rarely" rather "weakly" or "negligible"

   Response: Thanks for suggestion. Done.

6. L128: "This step facilitates" – In the meaning of "prepares for" (things done in module (C))?

   Response: Yes. We modified the presentation to make it clearer:

   *"This step prepares for a quick and convenient second-round interpolation from the regular model grid to the radar beam trajectories (in step 3). "*

7. L147: "specifies the settings of hydrometeor" – rather "properties"? Hydrometeors.

   Response: Thanks for suggestion. Done.

   *"The hydrometeor submodule (D) specifies the properties of hydrometeors in each radar gate along each trajectory. "*

8. L167: "would be performed" – is? The whole "If" seems unnecessary – if N=M=1, then it can simply be seen to be performed over 1x1 pts.

   Response: It means the antenna pattern weighted averaging will be carried out if there are more than one sampling sub-trajectories. Yes, N=M=1 is just a special case of the general treatment of sub-beams sampling method. We removed the "if" clause:

   *"... Finally, antenna pattern integration (weighted averaging for sub-trajectories) is carried out. ..."*

9. L197: "At this point, we should" – This sentence should, if there at all, be at beginning of Sec2.2. Further, why "should" you?

   Response: Thanks for suggestion. This sentence has been moved to the beginning of Section 2.2.

   Yes, "we can specify …" is better than "we should specify …" here, considering that there are enough contents introducing the general flow-

10. L244: "Svv/hh" – wrong order of subscripts? It's not zh that is a function of Svv, but zv, is it? L249 and elsewhere: "without dimension" -> "dimensionless"

Response: Yes. We have separated the equation of Zh and Zv.
"without dimension" has been substituted by "(dimensionless)" for ZDR and RHOHV.

11. L256f: Either use two separate equations for ah and av or use ± and ∓.

Response: We have separated the equations of ah and av.

12. L262: In my understanding, that angle symbol typically rather indicates phase, not amplitude (see e.g. Wikipedia on complex numbers). Why not use the explicit calc formula as in zh and rhohv calcs (ie |x|)?

Response: Yes. It is a typo. It should be phase rather than amplitude in the text. In terms of explicit calc formula of the phase, amplitude notation is not enough, functions in the form of arctan2(y, x) is needed. We think the phase expression here is clearer.

*"... The notation '∠' in Eq. (13) indicates the phase of the complex value. ..."*

13. L264: "radii to deg. Again." – typo?

Response: Yes, not "again", the last unit transform is from radii·km-1 to deg.·km-1, but this time it transforms from radii to deg. We removed the "Again" here.

14. L316 and elsewhere: "possibility distribution" -> "probability distribution"

Response: Done for 5 places in the paper.

15. L322: "the hydrometeor category follows" – which hydrometeor category?

Response: Actually, all the PSDs of the hydrometeor categories implemented currently (rain, snow, graupel) can be described by the gamma distribution. So, we removed the "For instance" here.

*"In this study, all the hydrometeor categories follow the gamma distribution ..."*

16. L326: "is a prescribed constant in the microphysics package" – which microphysics package? of ZJU-AERO? the CMA-NWP?

Response: Actually, this sentence is indicating that mu is always a prescribed constant rather than prognostic parameters in almost all the bulk microphysics parameterizations (single moment / double moment).
For the ZJU-AERO, all the PSD distributions listed in Tables 1 assume mu as a constant (either microphysics-consistent or observation-operator-forced).
For CMA-NWP, the single-moment microphysics scheme used also assume mu as a constant.

*"... As is often the case (for all PSD options in ZJU-AERO, see Table 1), $\mu$ is a prescribed constant in the microphysics package. ..."*

17. L330: "solve the unknown parameter λ" – provide λ formula here, not 5 lines later (equivalently Eq(19) already belongs at "given the mass concentration" just before).

Response: Yes. we put the sentence to 5 lines later rather than move the formula here. We find it also make sense, equivalently.

*"If a hydrometeor category is of single-moment microphysics scheme, given the mass*

*concentration of arbitrary hydrometeor category "x" $Q_x$ in units of $kg \cdot m^{-3}$:*

**[Equation of $Q_x$]**

*…… Then we can solve the unknown parameter $\lambda$ and determine all relevant PSD information that pertains to that radar gate analytically by plugging Eq. (21) into Eq. (23):*

**[Equation of $\lambda$]"**

18. L360: what are BSP and SSP? First occurence, so spell out.

    Response: Done. It is bulk scattering properties and single scattering properties, respectively.

19. L361, 365: "now": what is that supposed to express?

    Response: The first "now" is redundant. It is removed.

    The second "now" means that "ZJU-AERO currently encompasses three types of hydrometeors", implying that there will be more types of hydrometeors in the future (e.g., melting particles, hails, cloud ice, etc.). It is now substituted by "currently".

20. L405: "we could uniquely determine" -> "any arbitrary orientation of the particle can uniquely be determined"

    Response: Good suggestion! Done.

21. L407: "With a specified orientation and observation geometries" – Check grammar. Also, terminology is uncommon ("observation geometry" involves an observer in my interpretation, here it's rather the scattering geometry).

    Response: Done. All the "orientation and observation geometries" have been modified as "scattering geometry" in this paper.

22. L409: "We used the T-matrix code" – I assume, here not the (or any specific) code is meant, but the T-Matrix approach or method.

    Response: Yes, "we used the T-matrix method" is better here.

23. L427: "the particle with original orientation" – original? Do you mean the reference orientation (alpha=beta=gamma=0°), ie where L- and P-system are oriented identically?

    Response: Yes. It is the reference orientation.

    *"Panel (a) shows the laboratory coordinate system $OX_LY_LZ_L$ and the particle with reference orientation ($\alpha=\beta=\gamma=0$)."*

24. L442f: "as shown in Eq(26)" – put Eq(26) here and integrate into sentence.

    Response: Done

25. L451: "more prone to aerodynamical effects" – meaning? why relevant here?

    Response: Yes, this sentence is incomplete. It means that large drops are more prone to aerodynamic effects, therefore having a flatter base compared with the reference spheroid (see Fig 6c).

    *"It can be noted from Figure 6(c) that larger raindrops are more prone to aerodynamic effects and therefore having a flatter base compared with the reference "spheroid"."*

26. L481: "it is necessary to visualize" – To analyze, I agree. But that does not necessarily require visualization.

    Response: Yes, Done.

27. L485: "often referred to as [...]" – Provide reference.

Response: Yes, we provide a text book of polarimetric weather radar as reference:
*"Zhang, G.: Weather radar polarimetry, Crc Press2016."*

28. Eq(27): Check notation in equation. Really zhv == <[zhv]>, ie the SSP factor for the unpolarized reflectivity is equal the unpolarized reflectivity itself?

Response:

The SSP factor of horizontal reflectivity is defined by the first formula:

$$[z_h] = \frac{(Z_{11} - Z_{12} - Z_{21} + Z_{22}) \cdot (2\lambda^4)}{\pi^4 |K_w|^2}$$

Then we can plug in the SSP factor definition to the particle ensemble integration signature '<>' and prove that zhv == <[zhv]> establishes:

$$\langle [z_h] \rangle = \left\langle \frac{(Z_{11} - Z_{12} - Z_{21} + Z_{22}) \cdot (2\lambda^4)}{\pi^4 |K_w|^2} \right\rangle$$

$$= \frac{(2\lambda^4)}{\pi^4 |K_w|^2} \langle Z_{11} - Z_{12} - Z_{21} + Z_{22} \rangle$$

$$= \frac{(2\lambda^4)}{\pi^4 |K_w|^2} \left( \langle Z_{11} \rangle - \langle Z_{12} \rangle - \langle Z_{21} \rangle + \langle Z_{22} \rangle \right)$$

$$= z_h$$

This above equation does not mean that the SSP factor for the horizontal reflectivity is equal to the horizontal reflectivity itself. Instead, it means that the particle group ensemble mean of SSP factor of horizontal reflectivity is horizontal reflectivity.

$[z_h]$ is expressed in terms of the SSP (i.e., elements of $\mathbf{Z}$), while $z_h$ is expressed in terms of the BSP (i.e., elements of $\langle \mathbf{Z} \rangle$). Basically, The SSP factor of a radar variable is defined as one variable whose particle group ensemble mean is proportional to (or equal to) that radar variable. We reformulated the expression here:

*"The SSP factor "$[z_h]$" for the horizontal reflectivity "$z_h$" is defined as:*

$$[z_h] = \frac{(Z_{11} - Z_{12} - Z_{21} + Z_{22}) \cdot (2\lambda^4)}{\pi^4 |K_w|^2} \tag{33}$$

*Here, the quantity enclosed in square bracket "$[z_h]$" indicates the SSP factor of the radar variable "$z_h$". we can perform particle ensemble mean on it, which is often indicated by angle brackets (Zhang, 2016):*

$$\langle [z_h] \rangle = \frac{2\lambda^4}{\pi^4 |K_w|^2} \left( \langle Z_{11} \rangle - \langle Z_{12} \rangle - \langle Z_{21} \rangle + \langle Z_{22} \rangle \right) = z_h \tag{34}$$

*Here, the last equal is the definition of horizontal reflectivity in Eq. (7). Similarly, the SSP factor "$[a_h]$" for horizontal attenuation coefficient "$a_h$" is defined as:*

$$[a_h] = K_{11} - K_{12} = \sigma_{ext,h} \tag{35}$$

*If we perform particle ensemble mean on it, we find it is proportional to the definition of horizontal attenuation coefficient in Eq. (11):*

$$\langle[a_h]\rangle = (\langle K_{11}\rangle - \langle K_{12}\rangle) = 10^3 \cdot a_h,$$

(36)

*As shown by Eqs. (34) and (35), these factors ([$z_h$] and [$a_h$]) describe how a particle with a diameter Deq affects the radar reflectivity and specific attenuation, respectively."*

29. L495: "The beta dimension in the database are fixed at 0°" – Rather, the database entries for beta=0° are shown, while beta dimensions has more entries as suggested by Fig9.

Response: Yes. We modified the caption:

*"The database entries for β = 0 ˚ are shown."*

30. L500: As L495, T was not fixed in the database, I assume, but the results for T=10°C shown here.

Response: Yes. We modified the caption:

*"The database entries for temperature=10 ˚ C (283 K) are shown."*

31. Fig10: In caption, explicitly point out which SSP factors are shown in each panel (particularly, info on what is shown in row 1 vs row 3 is missing).

Response: Done. We pointed out the SSP factor shown in each panel in the caption:

*"The first column shows horizontal polarization SSP factors ([$z_h$] and [$a_h$]) mentioned in Figure 8 and Figure 9, the second column shows SSP factors contributing to observed differential reflectivity ([$z_{DR}$] and [$a_h$]- [$a_V$]), and the third column shows the SSP factors contributing to the observed total phase shift ([ $\delta_{hv}$] and [$K_{DP}$])."*

32. L516: "This phenomenon [...], which is likely attributed" – Ekelund20 attribute it to that.

Response: Yes, we changed the way of presentation.

*"This phenomenon was also found for the Chebyshev model at 94.1 GHz (Ekelund et al., 2020), which is attributed to its flatter bottom surface and sharper top surface geometries in Ekelund et al. (2020), as depicted in Figure 5."*

33. L539: "show a minor decay": Minor deviations/differences compared to beta=0° results? Or what is meant by decay?

Response: Yes, it means that the CmS deviations only decrease a little when beta < 10 deg. compared to beta = 0 deg. results, as shown in Figure 9. Maybe "show a minor decrease" would be better.

34. L550: "Among the six schemes, group A is characterized" – First introduce that you distinguish two groups before characterizing each.

Response: Good suggestion! Done.

*"We classified the six PSD schemes into two groups by how the intercept parameter $N_0$ is determined. Group A is characterized by …"*

35. L564: "by assuming q_air=[...] of standard atmosphere" – At pressure of...?

Response: Yes, At sea level (1013.25hPa).

*"by assuming $\rho_{air}$=1.225 kg • m$^{-3}$ of standard atmosphere at sea level (1013.25hPa)."*

36. Fig11: As RWC lines are only background information, present them less prominently (thinner lines, uncolored lines? or maybe as weakly coloured filled contours?). Could PSD

scheme curve style be selected to make groups A and B easily distinguishable?

Response: OK. We present the RWC lines with less prominent styles (thinner and monochromatic lines). We try to distinguish PSD schemes using curve series with different colors (red and black).

[Figure]

37. Fig12: Indicate more clearly in the upper-row panels that black & red curves are m(D)N(D) measuring on lefthand y-axis, while blue is sig_bsc/M(D) measuring on the righthand y-axis (e.g add blue in legend. and/or color the sig_bsc/m(D) formula in the top right of the panels blue). Axis labels and legend text should be increased to a readable font size.

Response:

(1). Done. We indicate that the black & red curves are $Q_R$-normalized m(D)N(D) of different PSD schemes, while the blue curve is sig_bsc/M(D) by adding legends.

(2). Done. We have increased the labels and legend text to readable font size.

Backscatagrand for Raindrop @ T = 283.0 K, Elc = -80.0°, Ka-band (35.50 GHz)

38. Fig12: Use identical Deq ranges on x-axes over all panel for better comparability. Maybe scale lefthand y-axes (upper and lower panel) by QR to make shape of PSD better comparable.

    Response: Done (see the above figure). We have normalized the distributions by Q$_R$.

39. L598f: "while the single-particle unpolarized backscattering per unit particle mass sig_bsc,hv(D)/m(D) is indicated" – Use the previously introduced (L592) terminology 'mass backscattering efficiency', otherwise you add confusion.

    Response: Good suggestion! Done.

40. L602f: "Readers can verify that the" – Remove that unnecessary and odd intro (reader anyways cannot; they can at max roughly qualitatively guess), simply leave the following statement.

    Response: Done.

41. L605: "exhibit large uncertainties" – Are these indeed uncertainties? Or, simpler, differences?

    Response: The six PSD schemes introduced in Table 3 roughly covers the nature variability of PSDs over different regions (ocean and continent). Therefore, we think the differences in the results between those PSD schemes exhibit the natural variability (uncertainties) over different conditions.

    *"It can be concluded from Figure 12 that the mass distributions and backscattering contribution spectrum (i.e., the so-called "backscatagrand") exhibit large uncertainties due to natural variability of PSD schemes."*

42. L612: "Under moderately heavy precipitation conditions [...] outlier" – Under all but the heaviest conditions, actually.

    Response: Yes. Done.

    *"Under all but extremely heavy precipitation conditions (Q$_R$<10$^{-2}$kg·m$^{-3}$, see Figure12 c&g)"*

43. L614: "even though" -> "while"(?)

Response: Yes. It should be "while".

44. L663: "GPM-DPR's Ku-band radar is similar to" – meaning what?

Response: It means that GPM-DPR's Ku-band radar basically follows the instrument characteristics of its predecessor on TRMM (of course there are some improvements such as instrument sensitivities, swath width, etc.).

*"According to Iguchi (2020), the GPM-DPR's Ku-band radar basically follows the instrument characteristics of the Tropical Rainfall Measurement Mission (TRMM) Precipitation Radar (PR), while the new Ka-band radar is sensitive to light rain and snow."*

45. L665: "accurate estimates of DSD can be obtained" – rather "more accurate" as they still have uncertainties. Reference missing.

Response: Done.

We have added two references on DSD retrievals using DPR dual-frequency reflectivity:

*Rose, C. and Chandrasekar, V.: A GPM dual-frequency retrieval algorithm: DSD profile-optimization method, Journal of Atmospheric and Oceanic Technology, 23, 1372-1383, 2006.*

*Liao, L., Meneghini, R., and Tokay, A.: Uncertainties of GPM DPR rain estimates caused by DSD parameterizations, Journal of Applied Meteorology and Climatology, 53, 2524-2537, 2014.*

46. L670: "defined as the difference" – of log space reflectivities, isn't it?

Response: Yes.

*"The dual frequency ratio (DFR) is defined as the difference between the log-space measured reflectivity at two channels"*

47. L684: "freezing level [...] was found within" – in obs or model?

Response: It is the freezing level of model. We visualize the freezing level of NWP model using red contour lines in the updated figure.

*"Notably, the freezing level in this case was found within the altitudes ranging from 4 to 5 km (see the 0 °C NWP model background isothermal line in Figure 15)"*

48. L686f: "It's worth noting" – Why? How is this relevant here, telling me what?

Response: The long-term observation of melting layer (Fabry and Zawadzki, 1995) found that the melting layer signature is less significant in convective precipitations. That may explain why the melting layer is not so significant in this typhoon case.

*"As reported by long-term radar observation statistics of melting layers documented in Fabry and Zawadzki (1995), the magnitude of BB signature in melting layer is significantly weaker in deep-convection regions. Therefore, the weak BB signatures in melting layer of this tropical cyclone case can probably be attributed to the large riming rates (Zawadzki et al., 2005) in convective precipitations of tropical cyclone eyewall and rain bands."*

49. L694: "We plan to report [...] in upcoming publications" – Rather "Implementation of [...] will be subject to upcoming publications"

Response: Done.

*"Implementation of melting particle models and their relevant validations will be subject to upcoming publications."*

50. Fig14: Color scale with quasi-white region (around 18dBZ) seems not the best in combination with a white background.

    Response: OK. We switched to another colormap without quasi-white region for Fig14 and Fig15:

[Figure]

[Figure]

GPM-DPR Swath Cross Section Reflectivity between A and B (Observation and Simulation)

51. L711: "The radar operator [...]" – Remove; it doesn't belong in caption and is redundant anyways.

52. L752: "Not surprisingly" -> rather "As suggested by results in Sec3"
Response: Done

53. L762: "assessments of PSDs and morphology options" – Add: for rain.
Response: Done.
*"We also conducted sensitivity assessments of PSDs and morphology options for rain in ZJU-AERO"*

54. L835: Geer et al. has long ago been published as proper GMD paper. Update.
Response: Done.

**References:**

1. [Eriksson18] Eriksson, P., R. Ekelund, J. Mendrok, M. Brath, O. Lemke, and S. A. Buehler: A general database of hydrometeor single scattering properties at microwave and sub-millimetre wavelengths. Earth Syst. Sci. Data, 10, 1301-1326, doi: 10.5194/essd-10-1301-2018, 2018.

2. [Shrestha22] Shrestha, P., Mendrok, J., and Brunner, D.: Aerosol characteristics and polarimetric signatures for a deep convective storm over the northwestern part of Europe – modeling and observations. Atmos. Chem. Phys., 22, 14095-14117, doi: 10.5194/acp-22-14095-2022, 2022.

3. [Trömel21] Trömel S, Simmer C, Blahak U, et al.: Overview: fusion of radar polarimetry and numerical atmospheric modelling towards an improved understanding of cloud and precipitation processes. Atmos Chem Phys., 21, 17291-17314, doi: 10.5194/acp-21-17291-2021, 2021.

4. [Zhang21] Zhang, G., J. Gao, and M. Du: Parameterized Forward Operators for Simulation and Assimilation of Polarimetric Radar Data with Numerical Weather Predictions. Atmos Sci., 38(5), 737-754, doi:10.1007/s00376-021-0289-6, 2021.

Response: References were added in the revised manuscript.

---

## Referee Report (RR1)

**Revision-Review Xie et al. „ZJU-AERO V0.5: An Accurate and Efficient Radar Operator Designed for CMA-GFS / MESO with Capability of Simulating Nonspherical Hydrometeors "**

**General remarks**

The revision of the manuscript includes a number of clarifications that I appreciate. In my opinion, the ratio of presented content (or amount of text and figures) to scientifically interesting and novel content is quite high making it fairly tedious to read. Readability and tangibility, hence quality, of the paper could significantly improve in my opinion if the manuscript were purged from not-too-relevant or basic details (that were much better placed in the user guide) as already pointed out in the initial review.

The amount of present, though hidden-by-clutter, "core content" is sufficient for publication in my view, so rejection is uncalled for. And while I'd personally strongly recommend the authors to make the manuscript more concise, I leave it to the editor to decide how strongly to request that.

**Specific remarks**

L20: "database [...] with a multi-layered architecture" – Here, it remains unclear what that might refer to, hence meaningless. I suggest to spend a few more words on this.

L39f: "non-linear [...] nature of the observation operator" – The challenge is not the operator nonlinearity, but the nonlinearity of the processes the operator describes.

L48ff: I suggest to cover all mentioned RFOs in roughly the same detail (eg Zeng16 also provides a melting/mixed-phase particle scheme, makes use of Mie/T-Matrix-based bulk scattering lookup tables for speedy calculations and is online-coupled to the COSMO and ICON models).

L70: The meaning of both "accurate" and "multi-layered" in this context is not clear.

L84: "The formulas [...] are briefly presented" – Remove brief here; it's fairly detailed.

Fig1: This figure is only shortly mentioned in text, as far as I see without an actual clear purpose ("certain factors, such as beam-broadening, beam-bending, and beam-blocking among others (as shown in Figure 1)"). Remove it unless you give it a proper description/explanation and purpose.

L130: "can also interface with the output of WRF NWP model" – what is the difference/add-on to the general NWP variable reading from external storage files from the 1st sentence in this module description?

L131ff: This bullet point first suggests that trajectory are solved online (L131), then mentioning choice of (the same or a different?) online solver and an offline solver. Confusing.

L137: "radar variables are averaged" – More precisely, that should be an integration.

L167 & L173: Is the averaging (or integration) really done over the observable dual-pol parameters, specifically ZDR and rho_hv? This seems wrong since these variables are not additive.

L175: "procedures of steps 1-5 (excluding [...]) are independent" – doesn't that imply, it's steps 2-5 only?

L266: What are the $S^{fwd}$?

L358: "possibility-weighted averaging" – probability-weighted

L430: "various orientations" – Mishchenko refers to them (steps 1-3) as "fixed orientation" in contrast to orientation-averaged ones (step 4). Might be helpful to make use of this established terminology.

L435: "while for particles without axial symmetry or those that are inhomogeneous, we applied the invariant-imbedding T-matrix (IITM) code" – To which particles does this refer in the current state? According to later parts of the manuscript, IITM has been applied for the Chebyshev-shaped rain drops (which are actually both axial symmetric and homogeneous). For any others yet?

L439f: Add reference to conversion formulas.

L451ff: The sin(beta) does not really belong to the Gaussian probability distribution, but to the integration over polar angle (differently formulated: the sin(beta) has to appear in any integration over polar angle regardless of the purpose of that integration (compare eg phase function integration formulas) and of the probability distribution applied (compare eg formula for total random orientation)). Moreover, Eq31 specifically gives a Gaussian around beta=0°.

L467: "To improve the accuracy of raindrop modeling" – Since your paper is on radar forward operator, this is misleading. Make clear that this only refers to the geometric model of the raindrop, not to any radar modeling. Also, formulation seems to (wrongly) imply that you will do/develop these improvements.

L468f: "By incorporating these factors, a more accurate representation of raindrop model" – more accurate geometric(!) representation of raindrop shapes (of real, observed shapes, not of a model!).

L524: Why inventing a new name for this, when it is nothing more than the attenuation cross section?

L528f: "describe how a particle with a diameter Deq affects the radar reflectivity" – It's not the effect (ie a contribution) that is shown, particularly not when it later regards [zDR], but the equivalent radar parameter of a single, monoddisperse particle of that size.

FigA1: Most of the caption does not describe what is seen in the figure, but explanation of theory or approach, hence belong in the manuscript text.

---

## Author Response (AR2)

**Reviewer # 1**

**Summary:**

The authors have made major modifications to the manuscript and have done a good job in addressing most of the points I raised in my review of the original submission. I have only a few comments that I think should be revised before it is ready for publication.

Response: Thank you for the positive feedback on our previous revision. We have carefully considered your additional comments and made further improvements to the manuscript accordingly. Please find our responses below:

**Comments:**

1.  Regarding my original major comment #3, the authors conducted the simulation of polarimetric radar variables for a case of the mesoscale convective system (MCS). It is difficult to evaluate the ability of this radar forward operator to reproduce polarimetric radar signatures since only one moment of PPI snapshot is shown. What I cannot understand is why the authors did not apply the forward operator to the Typhoon case presented in the manuscript. I think the simulation results shown in the manuscript for the Typhoon case should at least include the simulated polarimetric radar variables if the authors claim this operator is a new forward polarimetric radar operator.

    Response: (1). Typhoon Haishen did not make landfall in China (specifically, it did not cross over the longitude of 130E). Consequently, there are no available ground-based radar observations from the Chinese radar network for this typhoon. This is the reason we did not conduct the simulation of polarimetric radar variables for the Typhoon Haishen.

    (2). To compensate for this, we have conducted a simulation case of Typhoon Doksuri, which made landfall in Fujian Province of China, at approximately 02:00 UTC on July 28, 2023. This simulation was based on the operational forecast provided by the CMA-MESO, initialized at 18 UTC on July 27, 2023. The CAM-MESO run was with a horizontal resolution of 3km and a vertical resolution of 51 layers, using the single moment microphysics scheme of WSM6. We conducted volume PPI simulations at the beginning of four consecutive hours prior to the landfall event, specifically at 23UTC on July 27 and 00UTC, 01UTC, and 02 UTC on July 28 (presented in Figure 1, Figure 2, Figure 3, and Figure 4, respectively). The observations for these simulations were obtained from the S-band polarimetric radar located in Xiamen (Z9592).

    The ZJU-AERO used its default settings, which are compatible with the WSM6, with the number of horizontal sub-beam quadrature points set to 3 and the number of vertical quadrature points set to 5.

The simulations presented in Figure 1-4 for multiple moments and multiple PPI sweeps, along with the previously included case of MCS, effectively demonstrate the capability of ZJU-AERO in reproducing polarimetric radar variables for the NWP model employing single-moment microphysics schemes. The case data has been included in the updated version of the demonstration sample data, which has been uploaded to Zenodo. Additionally, the results have been incorporated into the amended version of the user guide.

[Figure]

Figure 1: Comparison of observed radar variables on two PPI sweeps with their simulated counterparts using the ZJU-AERO. Columns 1 and 3 show the observed radar variables for PPI sweeps with fixed elevation angles of 0.48° and 1.49°, respectively. Columns 2 and 4 present the model equivalents, simulating the same radar variables by ZJU-AERO. The four rows show the following radar variables; horizontal reflectivity ($Z_H$), differential reflectivity ($Z_{DR}$), specific differential phase shift ($K_{DP}$), and co-polar correlation coefficients ($\rho_{hv}$). The simulations were conducted at 23 UTC on July 27, 2023.

There are several important notes for simulations in Figure 1-4:
(1). For the observed PPI sweep at the lowest fixed elevation angle ($\gamma$ =0.48°),

the sector of azimuth angles ranging from 315° to 360° experiences notable partial clutter beam-shielding effects. While the simulations are able to capture this phenomenon, the partial shielding effect is less pronounced compared to the observations. This discrepancy might be caused by the deficiency of the 3-km resolution topology map used in the simulations (the same topology map used by the CMA-MESO model).

(2). The simulations yield higher values for $Z_H$ and $Z_{DR}$ compared to the observations, while the simulations of $K_{DP}$ are generally in good agreement. This indicates that the WSM6 single moment microphysics scheme has probably overestimated the effective radius of raindrops for tropical cyclones, which is also illustrated in Section 4.2 of the manuscript.

(3). The distinct melting layer signature observed in $\rho_{hv}$ cannot be reproduced by simulations. This discrepancy is attributed to the lack of mixed-phased particles in the ZJU-AERO.

[Figure]

Figure 2: As Figure 1, but shows the observation and simulations at approximately 00 UTC on July 28, 2023.

[Figure]

Figure 3: As Figure 1, but shows the observation and simulations at approximately 01 UTC on July 28, 2023.

[Figure]

Figure 4: As Figure 1, but shows the observation and simulations at approximately 02 UTC on July 28, 2023.

2. For the response to my original minor comment #5, I still have no idea how the authors calculate the Z11, Z12, Z21, and Z22. This is also a common problem that occurs repeatedly in the 2.2 section. Authors always try to explain the details in the forward operator using more complex definitions from other people's articles. It is easy to get lost in some of the basic concepts and not get the specific approach of the forward operator. I suggested the authors should remove some nonsignificant definitions and formulas and reorganize the 2.2 section to give the readers a clear idea of what key variables need to be calculated and how to calculate when reproducing this operator.

Response: Thank you for your suggestions.

(1). We have reorganized Section 2.2, placing the basic details of FSA and BSA to Appendix C, and the equations of polarimetric radar variables to Appendix D.

(2). We have added specifications on how to calculate Z11, Z12, Z21, and Z22. Also, we have provided formulas for the extinction matrix elements (K11, K12, and K34), which are used in this study.

**Reviewer # 2**

**General remarks:**

The revision of the manuscript includes a number of clarifications that I appreciate. In my opinion, the ratio of presented content (or amount of text and figures) to scientifically interesting and novel content is quite high making it fairly tedious to read. Readability and tangibility, hence quality, of the paper could significantly improve in my opinion if the manuscript were purged from not-too-relevant or basic details (that were much better placed in the user guide) as already pointed out in the initial review.

The amount of present, though hidden-by-clutter, "core content" is sufficient for publication in my view, so rejection is uncalled for. And while I'd personally strongly recommend the authors to make the manuscript more concise, I leave it to the editor to decide how strongly to request that.

Response: Thanks for the suggestions. We understand the reviewer's concern about the writing style. To address the reviewer's concern (hopefully), we have reorganized Section 2.2, placing the not-so-relevant details of FSA and BSA to Appendix C, the equations of polarimetric radar variables to Appendix D. We hope those modifications can improve the readability of the paper.

**Specific remarks:**

1. L20: "database […] with a multi-layered architecture" – Here, it remains unclear what that might refer to, hence meaningless. I suggest to spend a few more words on this.

   Response: Good suggestion. We included the following words:

   *"Specifically, three levels of databases are created that store the single scattering properties for different shapes at discrete sizes for various fixed orientations, integrated single scattering properties over shapes and orientations, and bulk scattering properties incorporating the size average, respectively.*

2. L39f: "non-linear […] nature of the observation operator" – The challenge is not the operator nonlinearity, but the nonlinearity of the processes the operator describes.

   Response: Yes. We have reformulated the expression.

   *"In essence, using radar data in the observation space is preferred over the model space due to the highly non-linear and non-unique nature of the processes that observational operator of polarimetric radar describes."*

3. L48ff: I suggest to cover all mentioned RFOs in roughly the same detail (e.g. Zeng16 also provides a melting/mixed-phase particle scheme, makes use of Mie/T-Matrix-based bulk scattering lookup tables for speedy calculations and is online-coupled to the COSMO and ICON models).

   Response: Good suggestion, we have included those details.

   *"Zeng et al. (2016) described an efficient radar operator that is online-coupled to the Consortium for Small-scale Modelling (COSMO) and Icosahedral Nonhydrostatic Weather and Climate Model (ICON) model, making use of Mie / T-matrix scattering look-up table of solid, liquid and melting (mixed-phased) hydrometeors, named as efficient modular volume-scanning radar forward operator (EMVORADO). Although the early versions of*

*EMVORADO focus on non-polarimetric radar variables, later developments on EMVORADO have enabled its capability on simulating dual-polarization variables and conducted sufficient evaluations (Trömel et al., 2021; Shrestha et al., 2022)."*

4. L70: The meaning of both "accurate" and "multi-layered" in this context is not clear.

Response: We have added a few explanations on "accurate" and "multi-layered":

*"We utilize a semi-analytical scattering computation approach of T-matrix to ensure accuracy and features a multi-layered optical database that includes single scattering properties at discrete sizes and orientations, integrated single scattering properties over shapes and orientations, and bulk scattering properties incorporating the size average. Additionally, ZJU-AERO allows for flexibility in particle orientation and shape probability distribution tuning."*

5. L84: "The formulas […] are briefly presented" – Remove brief here; it's fairly detailed.

Response: Removed.

6. Fig1: This figure is only shortly mentioned in text, as far as I see without an actual clear purpose ("certain factors, such as beam-broadening, beam-bending, and beam-blocking among others (as shown in Figure 1)"). Remove it unless you give it a proper description/explanation and purpose.

Response: We have added proper description / explanation to Figure 1 in the text of the manuscript.

*"The ZJU-AERO was developed to simulate the observable polarimetric radar variables for radar systems aboard on various platforms, including ground-based, space-borne, and potentially airborne radars in the future. The conceptual graph of the multi-platform radar operator is shown in Figure 1. While the physical principles of the weather radar detection process are universal, certain factors, such as beam-broadening, beam-bending, and beam-blocking among others (as indicated along the beam trajectory in Figure 1), which are critical to ground-based radars, are not equally important across platforms. For example, the beam-bending effect is typically negligible for space-borne radar due to large absolute elevation angles (usually 70~90°, as illustrated in Figure 1 for spaceborne radar) and shorter beam trajectories (usually < 20 km) below the model top, as compared to the ground-based radar (the trajectory can reach up to about 250 km). However, the simulations of space-borne and ground-based radars share consistent hydrometeor setting entries for snow/graupel, melting snow/graupel, and rain, such as the dielectric model, particle morphology (distribution), size distribution, orientation preference, and others."*

7. L130: "can also interface with the output of WRF NWP model" – what is the difference/add-on to the general NWP variable reading from external storage files from the 1st sentence in this module description?

Response: Actually, there are only technical issues about adapting the forward radar operator to interface with the external grid data of another NWP model. For example, the projections of model horizontal mesh, the staggered vertical grid, the model variable mapping table, etc.
We have added a sentence at the end of this module description to improve clarity:

*"Enabling ZJU-AERO to interface with grid data from another NWP model involves only*

*technical adjustments, requiring basic information about that NWP model's horizontal mesh (projections), vertical grid, and variable mapping table."*

8. L131ff: This bullet point first suggests that trajectory are solved online (L131), then mentioning choice of (the same or a different?) online solver and an offline solver. Confusing.

Response: Sorry for the confusion arises from mistaken formulation. We have fixed this inconsistency:

*"For ground-based radar (B1), users have the option of using an online trajectory solver that uses the temperature and humidity profiles above the radar site. Specifically, the atmosphere refractive index Na are determined from atmosphere temperature T, pressure P and water vapor mixing ratio Q. The trajectory is determined by a ray-tracing ordinary differential equation (ODE) solver (Zeng et al., 2014). Additionally,, ZJU-AERO offers an alternative option of using an offline 4/3R_E solver for ground-based radar in ZJU-AERO."*

9. L137: "radar variables are averaged" – More precisely, that should be an integration.

Response: Yes, we have reformulated the expression here:

*"The observable radar variables are calculated by integrating bulk scattering properties over the antenna patterns (as described in step 5) to obtain the final results (beam-broadening and beam-blocking are considered in this way)."*

10. L167 & L173: Is the averaging (or integration) really done over the observable dual-pol parameters, specifically ZDR and rho_hv? This seems wrong since these variables are not additive.

Response: Yes, the antenna pattern integrations are performed based on bulk scattering properties (<Z> and <K>) rather than dual-pol parameters. The procedure we described contains some mistakes. We have amended them:

*"5.2 Once the bulk scattering properties on each sub-trajectory gridpoint within each radar gate are available, the antenna pattern integration involves integrating the bulk scattering properties within the scanning volume of each radar gate.*

*5.3 Then, the core module calculates the intrinsic polarimetric radar variables on each radar gate, based on the bulk scattering properties incorporating the antenna pattern integration presented in step 5.2. These radar variables include single-polarization reflectivities ($Z_H$ for horizontal reflectivities), dual-polarization variables ($Z_{DR}$, $K_{DP}$, $\delta_{hv}$, $\Phi_{DP}$ and $\rho_{hv}$ for differential reflectivity, differential phase shift, backscatter differential phase and co-polar correlation coefficient, respectively), and attenuation variables (aH and aV for horizontal and vertical attenuation coefficient, respectively). The definitions of these variables can be found in Appendix D.*

*5.4 For a detailed explanation of the intrinsic radar variables, please refer to Zhang (2016). In the final step of core module, the observable radar variables are obtained by taking into account the attenuation and phase shift accumulated along the beam trajectories."*

11. L175: "procedures of steps 1-5 (excluding […]) are independent" – doesn't that imply, it's steps 2-5 only?

Response: Yes. Done.

*"The above procedures of steps 2-5 are independent for each single beam."*

12. L266: What are the S^fwd?

    Response: We have added explanations to the superscripts of "fwd" here:
    *"Here, the superscripts of "fwd" over matrix elements indicate that these are elements of the forward scattering matrix."*

13. L358: "possibility-weighted averaging" – probability-weighted

    Response: Done.

14. L430: "various orientations" – Mishchenko refers to them (steps 1-3) as "fixed orientation" in contrast to orientation-averaged ones (step 4). Might be helpful to make use of this established terminology.

    Response: Good suggestion. We have substituted the invented terminology of "various orientations" as the established terminology "fixed orientation".

15. L435: "while for particles without axial symmetry or those that are inhomogeneous, we applied the invariant-imbedding T-matrix (IITM) code" – To which particles does this refer in the current state? According to later parts of the manuscript, IITM has been applied for the Chebyshev-shaped raindrops (which are actually both axial symmetric and homogeneous). For any others yet?

    Response: (1). Basically, those are the guidelines for selecting scattering computation approaches. However, there are exceptions when the EBCM approach suffers from numerical stability issues computing scattering properties of Chebyshev raindrops, for which we use the IITM approach instead. (2). There is no particle geometric model that is not axial symmetric or homogenous when the manuscript was finished. However, we have developed a melting particle model with inhomogeneous geometric specification in another paper for ZJU-AERO (under review), for which the guidelines of using IITM takes effect.

    (3). We have amended the formulation here:

    *"The guidelines for selecting scattering computation approach are as follows: for both axially symmetric (i.e., the dielectric constant distribution of electromagnetic medium in spherical coordinates $\varepsilon(r, \theta, \varphi)$ is irrelevant with azimuth angle $\varphi$) and homogenous particles, we used the EBCM T-matrix code (Mishchenko and Travis, 1994), while for particles without axial symmetry or those that are inhomogeneous (or shapes that the EBCM approach suffer from numerical stability issues), we applied the invariant-imbedding T-matrix (IITM) code (Bi et al., 2013; Bi and Yang, 2014; Bi et al., 2022; Wang et al., 2023)"*

16. L439f: Add reference to conversion formulas.

    Response: Done.

    *"3. The forward and backward amplitude scattering matrices $\mathbf{S_{FSA}}$ were then converted into the backscattering Mueller and extinction matrices, $\mathbf{Z}$ and $\mathbf{K}$, respectively (Mishchenko, 2014)."*

17. L451ff: The sin(beta) does not really belong to the Gaussian probability distribution, but to the integration over polar angle (differently formulated: the sin(beta) has to appear in any integration over polar angle regardless of the purpose of that integration (compare e.g. phase function integration formulas) and of the probability distribution applied (compare e.g. formula

for total random orientation)). Moreover, Eq31 specifically gives a Gaussian around beta=0°

Response: Yes, sin(beta) should belong to the general form of the integration over probability distribution, regardless of the specific form of the probability distribution. We have amended the formulas:

*"The integration in the Level A to Level B database conversion tool can be formalized as integration over canting angle β as follows:*

$$\overline{\overline{\mathbf{X}}}(e) = \frac{1}{\pi} \int_0^\pi \sin(\beta) \mathrm{d}\beta \, \mathrm{p}(\beta) \overline{\overline{\mathbf{X}}}(e, \beta), \mathbf{X} = \mathbf{Z}, \mathbf{K} \qquad (17)$$

*Here* $\mathrm{p}(\beta)$ *is the probability distribution of canting angle β. A Gaussian distribution in polar angle is often used to approximate the probability distribution of canting angle:*

$$\mathrm{p}(\beta) = \exp\left(-\frac{\beta}{2\sigma_\beta{}^2}\right), \qquad (18)"$$

18. L467: "To improve the accuracy of raindrop modeling" – Since your paper is on radar forward operator, this is misleading. Make clear that this only refers to the geometric model of the raindrop, not to any radar modeling. Also, formulation seems to (wrongly) imply that you will do/develop these improvements.

Response: Thanks for the suggestion, we have made it clear that we only improved the raindrop shape representation in the forward radar operator ZJU-AERO by using the established Chebyshev raindrop model, not any other modeling aspects, not developing the geometric model on our own.

*"To improve the accuracy of raindrop shape representation in the forward radar operator ZJU-AERO, it is essential to apply an established raindrop model that accounts for various effects such as the surface tension, hydrostatic and aerodynamic pressures, and static electric forces."*

19. L468f: "By incorporating these factors, a more accurate representation of raindrop model" – more accurate geometric(!) representation of raindrop shapes (of real, observed shapes, not of a model!).

Response: Thanks for the suggestion, it has been reformulated.

*"By incorporating these factors, a more accurate geometric representation of observed raindrop shapes can be achieved compared to the most commonly used spheroid model."*

20. L524: Why inventing a new name for this, when it is nothing more than the attenuation cross section?

Response: That is because "the SSP factor of $a_h$" is the name regarding the function of this variable, while in practice, it can be defined just as the extinction cross section of this particle. It seems be better to provide a consistent naming rule for those SSP factors (i.e., [$z_h$], [$a_h$]…) for clarity.
We add a note that [$a_h$] is exactly the extinction cross section in the text to remind readers of this point.

*"Similarly, the SSP factor "[$a_h$]" for horizontal attenuation coefficient "$a_h$" is defined (the extinction cross section by essence)"*

21. L528f: "describe how a particle with a diameter Deq affects the radar reflectivity" – It's not the

effect (i.e. a contribution) that is shown, particularly not when it later regards [$Z_{DR}$], but the equivalent radar parameter of a single, monodisperse particle of that size.

Response: Yes. We have amended the interpretation of those factors here: *"As shown by Eqs. (21) and (23), these factors ([$z_h$] and [$a_h$]) are equivalent radar parameters of radar reflectivity $z_h$ and specific attenuation ah, respectively, for a single particle of size $D_{eq}$."*

22. FigA1: Most of the caption does not describe what is seen in the figure, but explanation of theory or approach, hence belong in the manuscript text.

Response: Done. We have moved the sentences describing the theory or approach to the manuscript of this appendix:

*"For space-borne radars onboard rain measurement satellite platforms, such as the Tropical Rainfall Measuring Mission/Global Precipitation Measurement/Fengyun3-Rain Measurement (TRMM/GPM/FY3RM), the trajectory of the radar beam can be treated without considering the beam-bending effects while still maintaining precision. The WGS84 coordinates of satellites, denoted as A (scLat, scLon, scAlt), in addition to the centre of foot-print, A1 (Lat, Lon), can be obtained through the satellite radar L1/L2 products thereafter referred to as swath files. These coordinates can then be used to calculate the local elevation angle of a given radar gate, C, using the knowledge of trigonometry in Figure A1(b). The length of segments H and RE can be computed by converting the (latitude, longitude, altitude) WGS84 coordinates to the Earth-Centre–Earth-Fixed (ECEF) coordinates. The range of radar gate AC is provided by the space-borne radar observation system in the L1 product of GPM/DPR known as "scRangeEllipsoid" (Iguchi et al., 2010). When neglecting the beam-bending phenomenon in the space-borne radar detection, the local elevation angle, e', can be expressed as e' = e-α in which e is the elevation angle on the satellite. The angle α could be determined using trigonometry, given that AC represents the range of the radar gate C."*